# Sequence and chemical specificity define the functional landscape of intrinsically disordered regions

Iris Langstein-Skora[1,13], Andrea Schmid[1,13], Frauke Huth [2,13], Drin Shabani [1], Lorenz Spechtenhauser[1], Mariia Likhodeeva [3], Franziska Kunert[3], Felix J. Metzner[3], Ryan J. Emenecker [4,5], Mary O. Richardson[5,8], Wasim Aftab[6], Maximilian J. Götz[1,9], Sarah K. Payer[1,10], Niccoló Pietrantoni[1,11], Valentina Valka[1,12], Sakthi K. Ravichandran[1], Till Bartke[7], Karl-Peter Hopfner [3], Ulrich Gerland [2], Philipp Korber [1]✉ & Alex S. Holehouse [4,5]✉

Intrinsically disordered regions (IDRs) pervasively engage in essential molecular functions, yet they are often poorly conserved as assessed by sequence alignment. To explore the seeming paradox of how sequence variability is compatible with persistent function, we examined the functional determinants for a poorly conserved but essential IDR. We show that IDR function depends on two distinct but related properties: sequence and chemical specificity. Whereas sequence specificity operates via binding motifs and depends on the precise order and identity of residues, chemical specificity reflects the sequence-encoded chemistry of multivalent interactions across an IDR and depends on local and global chemical properties. Unexpectedly, a binding motif essential in the wild-type IDR can be removed when compensatory changes to the sequence chemistry are introduced, highlighting the orthogonality and interoperability of these properties, and expanding the sequence space compatible with function. Our results provide a general framework for the functional constraints on IDR evolution.

Intrinsically disordered proteins and protein regions (collectively referred to as intrinsically disordered regions (IDRs)) play important and often essential roles in many biological processes across the kingdoms of life, often in the context of molecular recognition[1–3]. Despite their importance, IDRs are often poorly conserved as assessed by alignment-based methods[4–8]. One exception here is short linear motifs (SLiMs, referred to here as motifs): 5- to 15-residue regions that often contain multiple conserved positions in a consensus pattern[9–11]. Motifs can engage in sequence-specific binding (also referred to as site-specific binding[12]), where the precise order and identity of amino acids is critical[11,13]. Importantly, motifs are modular: they can mediate molecular recognition when inserted into otherwise non-binding contexts[9,13].

IDRs can also mediate molecular interactions in the absence of motifs via distributed multivalent interactions driven by sequence-encoded chemistry, an idea referred to as chemical specificity[3,8,14,15]. Such interactions can mediate stoichiometric protein–protein interactions or the formation of biomolecular condensates[12,16–20]. For distributed multivalent interactions, the precise order or even identity of specific amino acids can be unimportant; what matters is the presentation of certain chemical moieties.

Motif-based (sequence-specific) interaction and distributed multivalent (chemically specific) interaction are generally thought to drive orthogonal types of molecular recognition (that is, loss of a motif cannot be compensated for by changing the distal chemical context)[16,21].

**Fig. 1 | IDRs are poorly conserved, as assessed by linear alignment, and Abf1 is an essential protein with a poorly conserved yet essential IDR. a**, Schematic showing conservation and disorder across a hypothetical protein. **b**, Histogram of per-protein correlations ($r$) between per-residue conservation and disorder scores. The inset shows an example of a single protein, with each marker representing a single amino acid. The histogram reports on $r$ values for the entire yeast proteome. **c**, Percentage of the sequence defined as IDRs for essential ($n = 1,182$) versus non-essential ($n = 3,989$) *S. cerevisiae* proteins. The box plot reports on the distribution of values across the yeast proteome, with the median value shown as a central line, the box capturing the interquartile range and whiskers representing values within 1.5× the interquartile range. A Mann–

Whitney *U*-test to assess whether the two distributions differ revealed a *P* value of 0.69, indicating that they are not significantly different. **d**, Sequence analysis of Abf1 with per-residue conservation. The horizontal grey line corresponds to the conservation score expected for a randomly shuffled sequence (Null). **e**, Schematic of the plasmid shuffle assay. The measured variants were expressed from a plasmid. **f**, Spot dilution assays for semi-quantitative assessment of sequence-dependent growth rates, scoring each viable construct between +4 and +1 (Supplementary Fig. 2). **g**, Domain diagrams for Abf1 variants encoded from truncation mutations, with their viability shown to the right-hand side. **h**, Amino acid sequence of Abf1 IDR2[449–662]—the focus of this study. NLS, SV40 NLS; WT, wild type.

This orthogonality may also reflect the degree of order in the bound state[19]. Accordingly, our current working model for IDR evolution posits that substantial sequence variation is tolerated so long as essential conserved positions in motifs are retained and/or the bulk amino acid composition and patterning are maintained[5,22–24].

In this Article, using *Saccharomyces cerevisiae* as a model, we investigate the interplay between these two modes of interaction in the context of conservation and function in an essential IDR. Our results suggest that, rather than two distinct modes of interaction, IDR-mediated molecular recognition should be considered as a two-dimensional (2D) landscape, in which motifs can cooperate with—and even be entirely replaced by—distributed multivalent interactions (that is, chemical specificity). The insertion of short sequence modules can impart function not as bona fide motifs, but simply by altering the sequence chemistry. Our results imply that sequence context and motifs can compensate for and buffer one another, providing a key missing piece in our understanding of the sequence constraints on IDR evolution.

## Results

### Proteome-wide analyses reveal weak alignment-based conservation for yeast IDRs

IDRs often undergo more substantial sequence variation than folded domains[4,6]. To quantify this difference (Fig. 1a), we performed a systematic analysis of sequence conservation and predicted disorder across

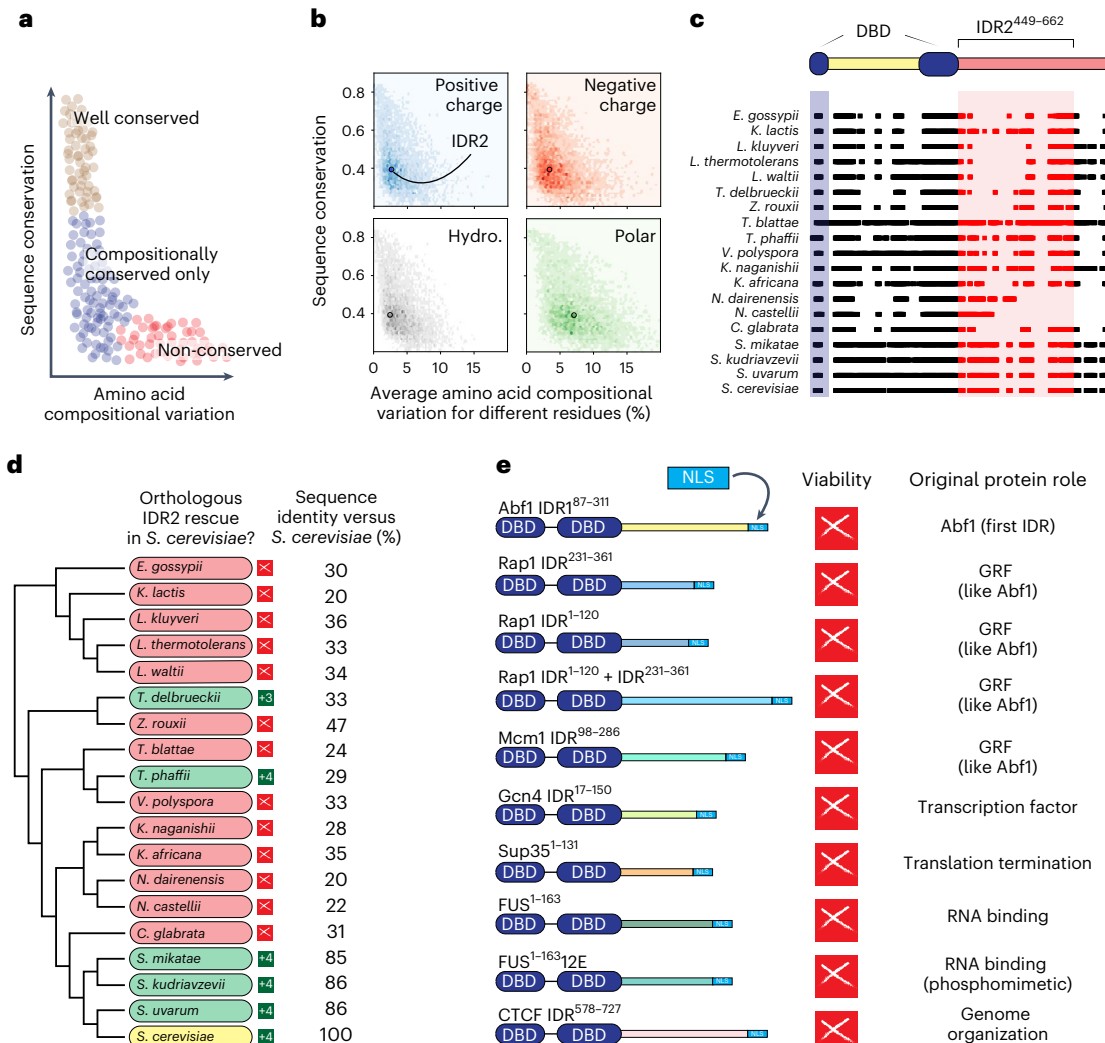

**Fig. 2 | IDR2 shows relatively conserved composition, yet most orthologous sequences cannot rescue function. a**, Schematic of compositional and sequence conservation. **b**, An analysis of all *S. cerevisiae* IDRs revealed that Abf1 IDR2 (coloured dot) was relatively conserved in charge and hydrophobicity (Hydro.) composition, but less so for polar composition. See also Extended Data Fig. 2. **c**, Schematic of orthologous IDR2s shown across full sequence alignment. **d**, Phylogenetic tree (for the whole organism) versus IDR2 rescue ability. Red

and green shading represent inviability and viability, respectively, of the IDR2 orthologue in *S. cerevisiae* (shaded yellow). Most orthologues did not support viability. Sequence identity versus *S. cerevisiae* IDR2 is shown to the right. See also Extended Data Fig. 3. **e**, Alternative IDRs tested, with their original functions noted to the right. In all cases, IDRs are fused C-terminal of the bipartite Abf1 DBD. See also Supplementary Fig. 4.

the *S. cerevisiae* proteome (Methods)[25,26]. This analysis revealed pervasive disorder and confirmed an expected anti-correlation between conservation and disorder (Fig. 1b and Extended Data Fig. 1a–f). Interestingly, essential proteins in *S. cerevisiae* were, on average, as disordered as non-essential proteins (Fig. 1c)[27]. We therefore wondered whether seemingly poorly conserved IDRs in essential proteins could be functionally conserved despite substantial sequence variation.

### Abf1 is an essential chromatin protein with an essential IDR
We sought a model protein to explore the conservation of function without sequence conservation. Because long IDRs are abundant in chromatin-associated proteins (Extended Data Fig. 1g), we turned to yeast general regulatory factors (GRFs). GRFs typically harbour sequence-specific DNA-binding domains (DBDs) and long IDRs (Supplementary Fig. 3)[28], are abundant and are mostly essential for viability. Although often referred to as transcription factors, their IDRs mostly do not contain transactivation domains[29], and they modulate genomic processes by organizing or partitioning chromatin[30–38], akin to mammalian architectural factors, such as CTCF[39].

Abf1 is an essential *S. cerevisiae* GRF[40] that modulates DNA accessibility via nucleosome positioning[31,33,35]. As an insulator, it uncouples expression levels at bidirectional promoters[30,41,42] and participates in roadblock termination of transcription[43–45]. With a bipartite DBD and two poorly conserved IDRs (IDR1 and IDR2), Abf1 was well suited for our study (Fig. 1d). Although the DBD mediates sequence-specific DNA recognition, the function of the IDRs is unclear, although all evidence suggests they mediate molecular interactions with partners[46–50].

Our interest in Abf1 was buoyed by previous observations—reproduced by us (Supplementary Figs. 1 and 2)—that the full-length *Kluyveromyces lactis* orthologous Abf1 could complement function in *S. cerevisiae*, despite 150+ million years of evolution[51]. That orthologous IDRs offer the same functional conservation is often assumed but rarely tested. To investigate this, we began by asking which wild-type (*S. cerevisiae*) IDRs from Abf1 were necessary and sufficient for function.

Previous truncation experiments suggested important carboxy (C)-terminal sequences (CS1/2) in IDR2 (refs. 52,53). We extended this truncation approach using a classical plasmid shuffling assay, in which the genomic copy of *ABF1* was deleted and the sole copy of wild-type

*ABF1* was provided on a plasmid carrying a *URA3* marker (Fig. 1e)[54]. Transformation with a plasmid bearing a mutated *abf1* gene and selection on 5-fluoroorotic acid (5-FOA) plates for loss of the wild-type *ABF1* gene plasmid (5-FOA is toxic in the presence of *URA3*) enabled us to assess whether the mutated *abf1* gene supported viability (Methods, Supplementary Fig. 1 and Supplementary Tables 1–8).

While *abf1* deletions are inviable, how Abf1 variants alter growth rate provides one route to investigate Abf1's essential (but so far ill-defined) function. We rated the growth rates of viable mutant constructs semi-quantitatively, from +4 (similar to that of the wild type) to +1 (very slow growth) in serial-dilution spotting assays (Fig. 1f and Supplementary Fig. 2). For the inviable constructs, we further confirmed that they: (1) were expressed; (2) entered the nucleus; and (3) bound specifically to selected Abf1 DNA-binding sites by anti-Flag chromatin immunoprecipitation (ChIP) targeting the Flag-tagged Abf1 variant in contrast with untagged wild-type Abf1 (Methods and Extended Data Fig. 2a).

Our truncation approach revealed that IDR2 was essential, but not IDR1 (Fig. 1g). The maximal C-terminal IDR2 truncation reported previously to be viable in the presence of IDR1 (that is, IDR2$^{449-662}$)[53] was also viable at a wild type-like growth rate in the absence of IDR1 (Fig. 1g,h). We used IDR2$^{449-662}$ as our wild-type IDR reference and refer to it as IDR2 for the remainder of this work.

IDR2 is poorly conserved across all orthologues (Fig. 1d), yet essential for viability, at least in *S. cerevisiae*. The CS1/2 region—the conservation peak in Fig. 1d at around residue 650—is a nuclear localization signal (NLS)[49] and was unnecessary if a heterologous SV40 NLS was included, as was the case for all mutant constructs herein (Fig. 1g; IDR2$^{449-623}$).

### Abf1 IDR2 is poorly conserved by alignment but modestly conserved by chemistry

Although conservation is often assessed by linear sequence (for example, via alignment), previous work established that IDRs can be conserved in terms of sequence composition[5,8,16]. We reasoned that protein sequences could be analysed in terms of compositional conservation and linear sequence conservation (Fig. 2a). A comprehensive analysis across *S. cerevisiae* IDRs and folded domains found that many IDRs were well conserved compositionally, despite poor linear sequence conservation (Fig. 2b and Extended Data Fig. 2b–g). Importantly, this analysis revealed that IDR2 is more conserved in terms of charged and hydrophobic residues than most IDRs with a similar degree of sequence conservation (Fig. 2b and Extended Data Fig. 2c–f).

### IDR2s from orthologous Abf1 proteins largely fail to rescue function in *S. cerevisiae*

We expected that compositional conservation in IDR2 would explain functional conservation, such that orthologous IDRs with similar composition would support viability in *S. cerevisiae*. To test this, we fused orthologous versions of IDR2$^{449-662}$ from 18 different yeast Abf1s to the *S. cerevisiae* Abf1 DBD (Fig. 2c). These orthologous IDR2$^{449-662}$ regions were identified using the highly conserved DBD and NLSs, which defined the start and end of IDR2$^{449-662}$ orthologues (Fig. 1d).

We tested our evolutionary chimeric proteins (*S. cerevisiae* Abf1 DBD and orthologous Abf1 IDR2) in our plasmid shuffle assay (Fig. 2d). Unexpectedly, outside of the sensu stricto *S. cerevisiae* complex (the bottom four species in Fig. 2d), only two of the 15 chimeras were viable. There was no obvious relationship between viability and sequence composition, sequence identity or sequence length (Extended Data Fig. 3a,b), falsifying our expectation that compositionally conserved IDRs would be functionally conserved.

Next, we tested several IDRs from proteins with similar functions and similar amino acid compositions, including IDRs from Abf1 (IDR1), other GRFs (Rap1 and Mcm1), a yeast transcription factor (Gcn4) and a human insulator (CTCF) (Fig. 2e, Extended Data Fig. 3c

and Supplementary Fig. 4). We also tested unrelated but compositionally similar low-complexity IDRs from the human RNA-binding protein FUS and the yeast translation termination factor Sup35 (Fig. 2e and Extended Data Fig. 3c). These IDRs failed to confer viability. However, to our surprise, some compositionally similar IDRs taken from other *S. cerevisiae* DNA-binding proteins (the transcription factors Gal4 and Pho4, as well as another GRF, Reb1) conferred viability in place of IDR2 (Fig. 3a and Supplementary Fig. 5).

Collectively, these results suggested that: (1) not any IDR could functionally replace Abf1 IDR2; (2) similarity in IDR composition is insufficient for IDR2 function; (3) not all yeast GRFs function in the same way via their IDRs; and (4) there may be functional overlap between the GRF Abf1 and some transactivating transcription factors (Gal4 and Pho4), but not others (Gcn4).

### Amino acid composition in IDR2 is insufficient to explain function

IDR2 appears to provide molecular recognition similar to that of Gal4, Pho4 and Reb1. These factors can also mediate chromatin remodelling[31–36,55–57]. We therefore wondered whether these IDRs share a common motif for recruiting the requisite machinery. Given that motifs depend on their linear sequence, shuffling a motif should disrupt its function. As an initial test, we designed three globally shuffled variants of IDR2 in which ~65% of the residues were shuffled across the sequence (Fig. 3b). These shuffles retained the same composition and wild type-like levels of patterning for different chemical groups (Extended Data Fig. 3d–f). These shuffles were inviable, demonstrating that IDR2-like composition alone is insufficient for viability and suggesting the presence of one or more motifs.

### IDR2 possesses an essential SLiM

To identify motif(s) in IDR2, we developed an unbiased approach termed sequential sequence shuffling (Fig. 3c). IDR2 was subdivided into abutting 30-residue windows and the sequence in each window was locally shuffled. This revealed two central windows that were intolerant to shuffling, which we confirmed by shuffling everything except the central 60-residue sub-sequence (Fig. 3c). We then repeated the procedure using ten-residue windows within the 60-residue sub-sequence. We identified a 20-residue sub-sequence (the essential motif) that could not be shuffled and was essential for IDR2 function (Fig. 3c). Despite being essential, this region was unremarkable with respect to conservation and other sequence properties (Fig. 3d).

We questioned whether similar motifs might exist in the other functional IDRs identified in Fig. 3a. Sequence alignment between IDR2 and Gal4$^{768-881}$ was relatively poor (Extended Data Fig. 3g), identifying just one loosely homologous region (named Abf1$^{G4}$ in Abf1 and Gal4$^{G4}$ in Gal4) (Fig. 3e). Intriguingly, shuffles of IDR2 in which Abf1$^{G4}$ was shuffled (LS-13 and LS-15) showed a growth defect (Fig. 3c), hinting that Abf1$^{G4}$ may contribute to function as an additional, albeit non-essential, motif.

Finally, we investigated the molecular basis for essential motif function. Structural bioinformatics predicted that this region forms a transient helix (Fig. 3f)—a feature that is frequently associated with IDR-mediated interactions[21]. To establish the importance of helicity, we introduced helix-disrupting point mutations in the DNA encoding the essential motif, which abrogated viability (Fig. 3g). These results are at least consistent with a model in which sequence-specific binding relies on a helical bound state.

### IDR context is a critical functional determinant

Motifs should—in principle—confer function when inserted into a non-functional context[10]. Context here refers to the IDR sequence surrounding the motif (Fig. 3h). We tested whether the essential motif met this definition. As a non-functional context, we selected the phosphomimetic variant of the low-complexity IDR from the human RNA-binding protein FUS (FUS$^{1-163}$12E)[58] (Supplementary Fig. 6). FUS$^{1-163}$12E is a

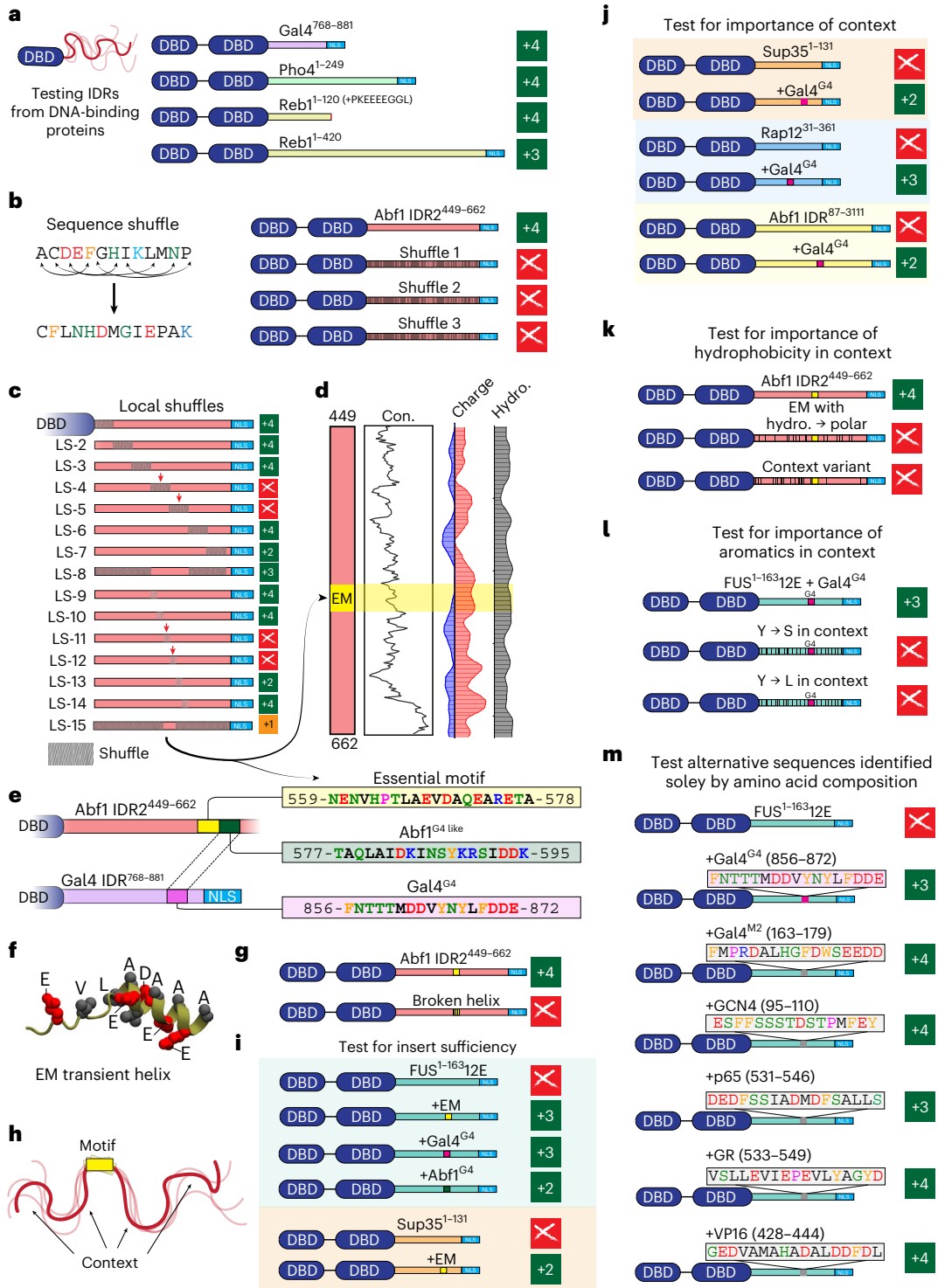

**Fig. 3 | Sequence motifs play a key role in IDR2 function. a**, IDRs from other (compare with Fig. 2e) yeast transactivators (Gal4 and Pho4) and from the GRF Reb1 conferred viability. **b**, Global shuffles were inviable (that is, IDR2 composition alone was insufficient for viability). **c**, Sequential sequence shuffling pinpointed an essential motif (EM) in the centre of IDR2. Note that LS-12 is also referred to as EM shuffle in Figs. 4c and 6. **d**, The EM was not conserved across orthologues or enriched for charge or hydrophobicity compared with the rest of IDR2. **e**, Three putative motifs are shown in the context of their IDRs. Abf1^G4 and Gal4^G4 were identified by sequence alignment of IDR2 and Gal4^768–881 (Extended Data Fig. 3g). **f**, Structural bioinformatics implicated the EM in forming a transient helix. **g**, Mutations leading to helix disruption

(those encoding substitutions of glycine for other polar residues) abrogated viability. **h**, Schematic showing motif and context. **i**, Insertion of the EM into the (non-functional) FUS^1–163 12E (shaded blue) and Sup35 (shaded orange) contexts conferred viability. Similarly, insertion of Gal4^G4 or Abf1^G4 into FUS^1–163 12E conferred viability. **j**, Gal4^G4 rescued viability in several unrelated non-functional IDR contexts. **k**, IDR2 context mutations that reduced hydrophobicity but preserved the EM were inviable. **l**, FUS^1–163 12E context mutations that reduced aromaticity but preserved Gal4^G4 were inviable. **m**, Sub-sequences compositionally matched to Gal4^G4 and taken from a range of transcription factors also provided viability when inserted into FUS^1–163 12E. Con., conservation.

compositionally uniform low-complexity disordered region that lacks secondary structure or known binding motifs[58,59]. However, FUS[1–163]12E contains uniformly spaced hydrophobic (aromatic) and acidic residues, conferring sequence properties similar to those of IDR2. Although FUS[1–163]12E alone was inviable (Fig. 2e), insertion of the essential motif into the FUS[1–163]12E context rescued viability (Fig. 3i). Similarly, inserting the essential motif into another inviable low-complexity sequence (Sup35[1–131]) also conferred viability (Fig. 3i). These results were consistent with the essential motif being a bona fide motif: it conferred viability as a modular entity. Interestingly, insertion of Gal4[G4] and Abf1[G4] into the FUS[1–163]12E context also rescued viability (Fig. 3i), consistent with (but not demonstrating) a model where Gal4[G4] and Abf1[G4] are also motifs.

In addition to the importance of inserted motifs, we tested whether viability depended on properties of the host IDR sequence context. First, the essential motif alone fused to the Abf1 DBD was inviable, demonstrating the need for IDR context (Supplementary Fig. 1). Next, we inserted Gal4[G4] in three more otherwise inviable IDR contexts (Sup35[1–131], Rap1[231–361] and Abf1 IDR1[87–311]) (Fig. 3j). In all of these, Gal4[G4] conferred viability, confirming its general ability to rescue function. Although these constructs were viable, they showed distinct growth rates; the Abf1 IDR1 and Sup35 contexts conferred slower growth than the Rap1 and FUS[1–163]12E contexts. Apparently, viability could be modulated not only through motifs, but also by altering the context.

Motivated by our conservation analysis, we tested the importance of hydrophobicity outside the essential motif (Fig. 2b). In stark contrast with our shuffle variants (Fig. 3c), two variants that preserved the essential motif but reduced hydrophobicity in distinct ways were inviable (Fig. 3k). Next, we designed variants of FUS[1–163]12E + Gal4[G4] in which the contextual aromatic residues were converted to serine or leucine (Fig. 3l), yielding inviable constructs (Fig. 3l). Furthermore, if Gal4[G4] was inserted into a glutamine-rich IDR from the yeast transcriptional co-repressor Ssn6, this variant was also inviable (Supplementary Fig. 1). This indicates that IDR context hydrophobicity is a key determinant of IDR function.

## Compositionally selected sub-sequences can rescue Abf1 function

Although IDR context depends on sequence chemistry, motifs should depend on sequence order and amino acid identity. To confirm that Abf1[G4] and Gal4[G4] were bona fide motifs, we intended to demonstrate that unrelated but length-matched sub-sequences with amino acid composition matching Abf1[G4] or Gal4[G4] were inviable. To our surprise, five randomly selected sub-sequences from yeast and non-yeast transcription factor IDRs with an amino acid composition similar to Gal4[G4] (Supplementary Fig. 7) were viable if inserted in the FUS[1–163]12E context (Fig. 3m). This extremely surprising result forced us to step back and revisit how IDR2 could be interacting with its partners.

## IDR-mediated function is driven by sequence and chemical specificity

Conventionally speaking, the ability of a short (<20-residue) sequence to confer function when inserted into a non-functional context is taken to demonstrate a bona fide motif (for example, a SLiM)[9,60]. Given that SLiMs can interact without acquiring a defined 3D structure, structural characterization is sufficient but not necessary to nominate a region as a SLiM[61].

Sequence-specific motifs are often defined by three characteristics: an inability to tolerate shuffling (Fig. 3c); sensitivity to point mutations (Fig. 3g); and autonomous modular activity (Fig. 3i). As such, our results confirmed that the essential motif is a bona fide motif. However, the discovery that compositionally similar but unrelated sub-sequences were also functional implied that either we had a remarkable ability to identify motifs by eye or something else was at play.

Thus far, we identified two functional determinants: (1) the presence of a motif (Fig. 3g,i,j); and (2) the presence of an IDR context that

we interpreted to mediate distributed multivalent interactions, as hydrophobic residues were critical (Fig. 3k,l). These two functional determinants can be considered in terms of sequence specificity (dependence on a precise amino acid sequence) and chemical specificity (dependence on sequence-encoded complementary chemistry, without the requirement for an exact amino acid order). Generally, these two modes are discussed separately; motifs are considered in the context-specific stoichiometric interactions, whereas distributed multivalent interactions are mainly associated with biomolecular condensates[13,16,20,21]. However, we now wondered whether these two interaction modes might instead exist on a combined 2D landscape (Fig. 4a).

To guide our intuition, we performed simple, coarse-grained simulations to quantify 1:1 binding between an IDR and a partner as a function of motif and context binding strength (Fig. 4b and Extended Data Fig. 4a–d). These simulations revealed a 2D landscape in which a bound state could be achieved through many combinations of motif and context binding strengths. Sequence changes could alter the context (Fig. 4b; from 1→2), the motif (Fig. 4b; 1→3) or both. Accordingly, we tested this conceptual framework via further rational sequence design.

Motifs are, by definition, sequence specific; they depend on their precise linear order of amino acids. It should not be possible to shuffle or distribute the residues of a motif and retain function. In support of this, variants in which the essential motif was either locally shuffled or had its residues distributed across the FUS[1–163]12E context were inviable (Fig. 4c). Essential motif distribution was also not tolerated in a Sup35 context (Fig. 4c), confirming that this result was not context dependent.

Next we asked whether motif shuffling or redistribution could be interpreted in the context of our 2D landscape. Motif redistribution simultaneously disrupted the motif and altered the context chemistry (Fig. 4d; 2→3). Motif shuffling (or point mutations in the DNA sequence encoding the motif) disrupted the motif without impacting the context chemistry (Fig. 4d; 4→6). Finally, variants resulting from mutations introduced to the DNA sequence encoding amino acid residues outside the motif disrupted the context chemistry (Fig. 4d; 4→5). This conceptual framework allowed us to re-interpret our additional variants in a new light.

## The identification of motifs requires motif shuffling or redistribution as a control

Our binding landscape model raised the intriguing possibility that Gal4[G4] and other sequences identified in Fig. 3m were not bona fide motifs, but instead altered the IDR sequence chemistry (that is, context), albeit locally. We tested this by distributing the amino acids of these sequences across the IDR context. In all cases, including multiple independent shuffles of the same sequence and across numerous distinct contexts, these distributed variants were viable (Fig. 4e). This demonstrated that these subregions (that is, Gal4[G4], p65 and GR, as shown in Fig. 3m) were in fact not bona fide motifs, but instead simply altered the sequence chemistry without regard for precise amino acid order or identity. In contrast, we reiterate that the essential motif is a bona fide motif: it does not tolerate point mutations (Fig. 3g), shuffling (Fig. 3c) or distribution (Fig. 4c). However, the same degree of viability could be achieved either with a combination of context and motif or with a context-only IDR.

## Rational design reveals the molecular grammar of chemical specificity

The context-only IDRs (Fig. 4e) prompted us to generate rational designs based on chemical principles. Given the importance of hydrophobic residues (Fig. 3k,l) we tested the sufficiency of a hydrophobic context by designing a FUS[1–163]12E variant with additional, evenly distributed hydrophobic residues (+4 tyrosine (aromatic) and +3 methionine

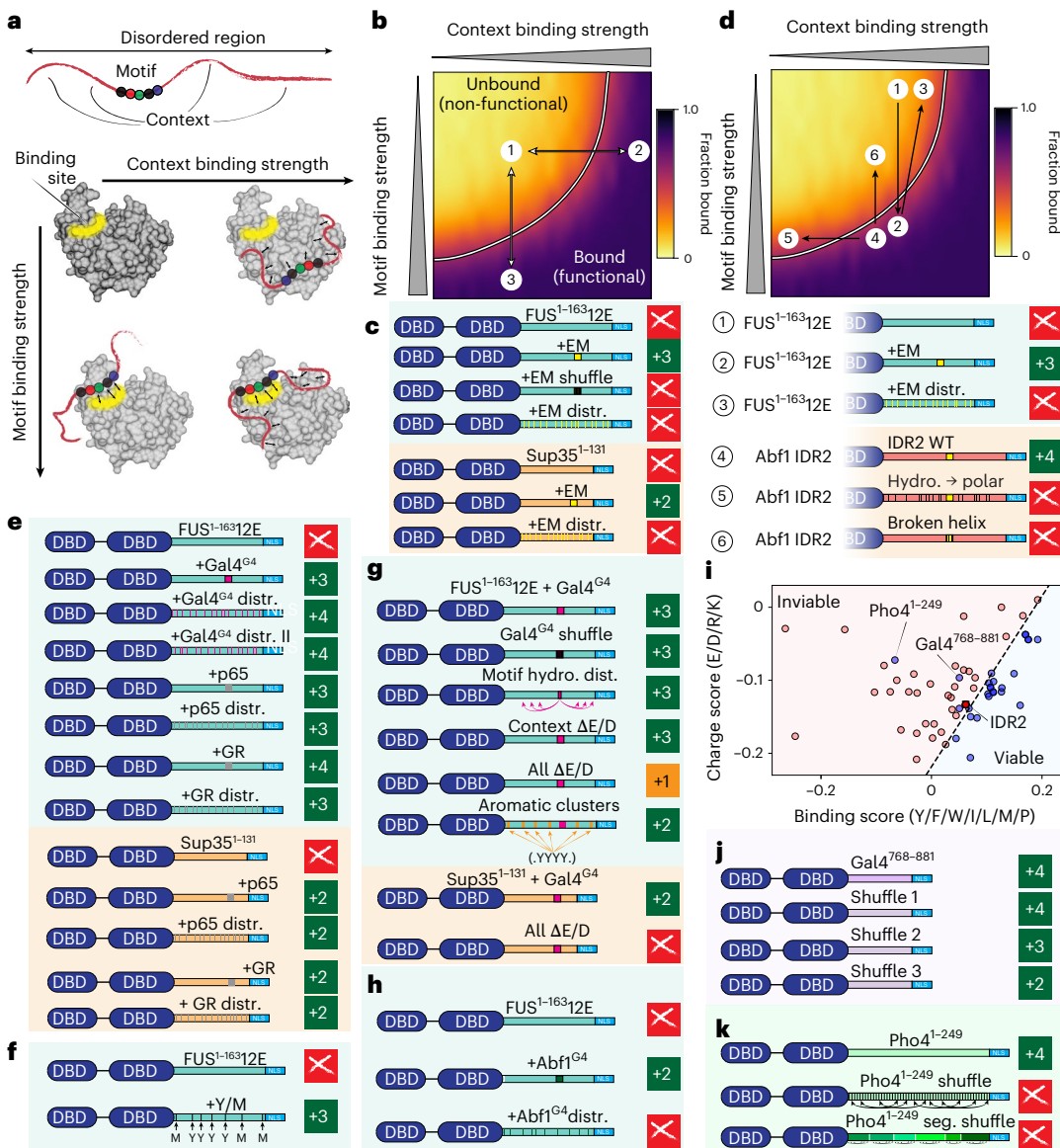

**Fig. 4 | IDR2-mediated interactions can be understood via a 2D binding landscape. a,** IDR-mediated interactions can be understood in terms of motif binding and context binding. **b,** Motif and context binding can be projected onto a simple 2D binding surface. Numbers represent identifiers to describe different types of mutations that modulate context (1→2) or motifs (1→3). **c,** The EM is a true motif; it cannot be shuffled or distributed (distr.). **d,** Variants can be interpreted through the 2D binding landscape. **e,** The sub-sequences identified in Fig. 3m conferred viability if distributed in FUS[1–163]12E and Sup35 contexts, with minimal change to growth phenotypes compared with the non-distributed motifs. Two independent Gal4[G4] distribution variants were tested to exclude the possibility that a new motif was generated by chance. **f,** Rational de novo design of a functional context-only IDR based on FUS[1–163]12E. **g,** Gal4[G4] rescued viability when added to

the FUS[1–163]12E context, and the residue order or relative positions of the Gal4[G4] residues did not affect this effect. The composition of acidic residues tuned the growth phenotypes in both a FUS[1–163]12E context (top, shaded blue) and a Sup35 context (bottom, shaded orange). The patterning of aromatic residues also tunes the growth phenotype in a FUS[1–163]12E context. **h,** Abf1[G4] rescued function in the FUS[1–163]12E context, but did not tolerate distribution, suggesting that this region is a motif. **i,** For the 68 sequences that lacked the EM, viable (blue) and inviable sequences (red) could be delineated based on the charge score and binding score. The numerical values are based on the weighted sequence composition. **j,** Global shuffles of the Gal4[768–881] construct revealed a variety of functional consequences. **k,** Although Pho4[1–249] supported viability, global and segmental (seg.) shuffle constructs were inviable, suggesting that Pho4[1–249] harbours a motif.

(aliphatic)) mirroring the chemical groups introduced by Gal4[G4]. This completely synthetic construct was fully viable with a near-wild-type growth phenotype (Fig. 4f). This result demonstrates that—at least in the context of our growth assay—an IDR with an evolved and essential bona fide motif can be replaced by a de novo-designed IDR lacking any kind of motif. We further established that locally shuffling or distributing Gal4[G4] in FUS[1–163]12E had no impact on growth (Fig. 4g; top), cementing the ability of a motif-free sequence to confer wild type-like viability.

We further explored the molecular grammar of chemical specificity using rational sequence design. Our bioinformatics analysis

implicated acidic residues as conserved (Fig. 2b). To test their importance, we removed all acidic residues from the context in the FUS[1–163]12E + Gal4[G4] sequence, with no impact on growth compared with FUS[1–163]12E + Gal4[G4] (Fig. 4g; bottom). However, removing all acidic residues from the sequence resulted in a viable but slow-growing phenotype (Fig. 4g). We interpret these variants as tuning chemical specificity (Fig. 4b). Similarly, the Sup35 + Gal4[G4] and Sup35 + EM constructs (where EM stands for essential motif) showed slower growth at baseline than the FUS[1–163]12E + Gal4[G4] and FUS[1–163]12E + EM constructs, respectively, and removing all acidic residues from the Sup35 context led to

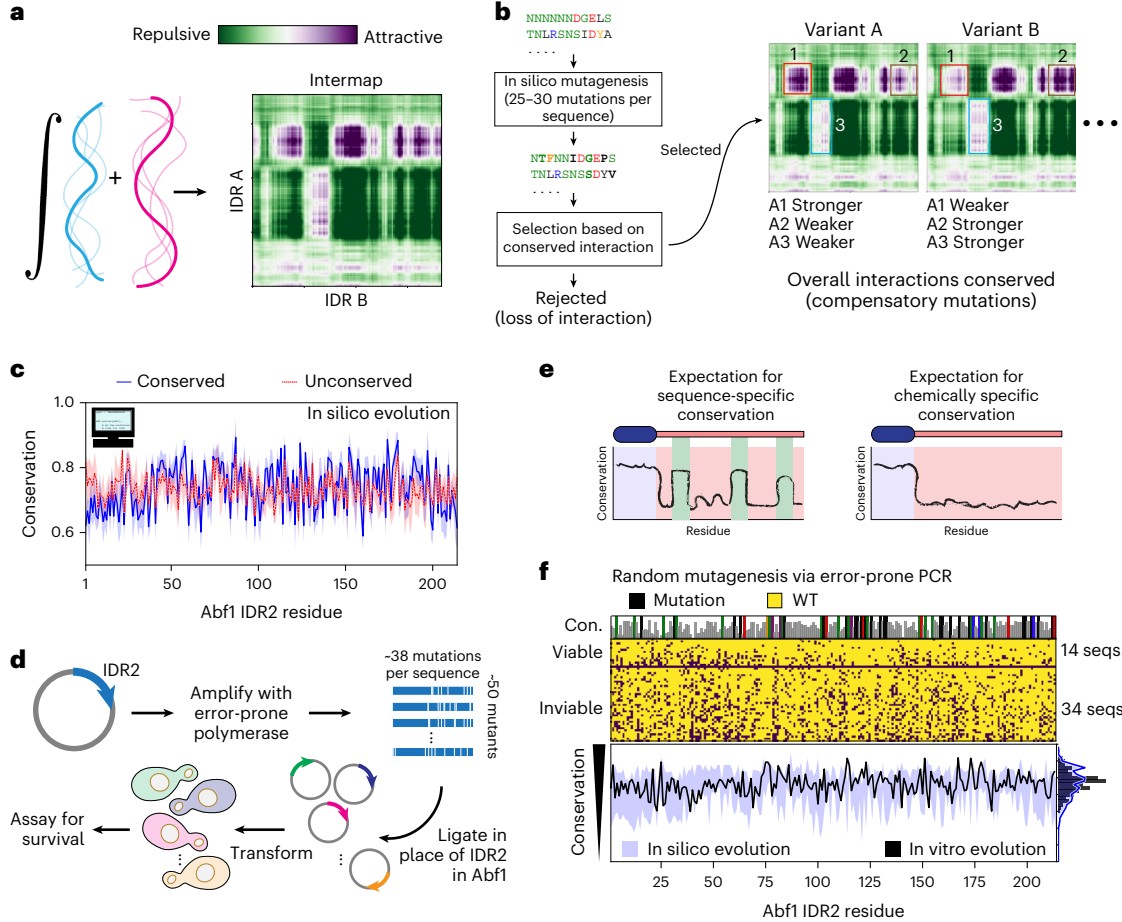

**Fig. 5 | Chemical specificity imparts cryptic conservation that is challenging to identify from multiple sequence alignments. a**, Chemical specificity between two IDRs can be predicted directly from sequence using FINCHES. **b**, In silico evolution scheme. Mutations were generated by converting the protein sequence to nucleic acids, stochastically mutating nucleotides using standard transition/transversion rates and then converting back to protein space. Selection was then performed by computing the predicted intermolecular interaction map between the IDR2 variant and the partner protein, requiring overall conservation of attractive interactions. **c**, Comparison of per-residue conservation for sets of orthologues evolved under selection (blue) versus no selection (dashed red). These two profiles were indistinguishable. **d**, Schematic of error-prone PCR used for in vitro evolution. **e**, Expected

conservation patterns for sequence (left) versus chemical (right) specificity. If sequence-specific conservation is at play, we expect to see local conservation peaks reflecting highly conserved regions. **f**, Results from in vitro and in silico evolution experiments. Top, in vitro evolution identified 14 viable and 34 inviable constructs. Viable constructs had mutations distributed throughout the sequence, including the region encoding the EM, without conserved peaks. Residues that were never altered in viable constructs are coloured by amino acid type (top bar plot). Bottom, comparison of per-residue conservation from 14 in vitro (black) and in silico (blue) evolution, where in silico experiments show the distribution that emerged when sets of 14 viable variants (that is, with conserved attractive interactions) were generated many different times with the same average number of mutations. Seqs., sequences.

an inviable construct (Fig. 4c,g). Together, these results suggested that Sup35 provided a weaker context than FUS[1–163]12E. The Sup35 context may be generally worse for binding than the FUS[1–163]12E context, or the two contexts may lead to distinct patterns of interaction with cellular components.

Patterning of aromatic residues has been shown to tune intermolecular interactions in various contexts[16,62,63]. To investigate this, we designed a FUS[1–163]12E + Gal4[G4] variant with aromatic residues clustered together (Extended Data Fig. 4e). Although this variant was viable—suggesting that evenly spaced aromatic residues were not a strict prerequisite for function—we observed a growth defect (Fig. 4g). This result suggests that changes in residue patterning can further tune chemical specificity.

Next we asked whether the essential motif is the only bona fide motif in IDR2. Given that shuffling Abf1[G4] in IDR2 compromised growth (Fig. 3c; LS-13 and LS-15), this hinted that IDR2 may contain another non-essential motif (i.e. Abf1[G4]) in addition to the already identified essential motif (EM). To test this, we designed a construct in which Abf1[G4] was distributed across FUS[1–163]12E (Fig. 4h). Whereas

FUS[1–163]12E + Abf1[G4] was viable, this distributed-Abf1[G4] construct was inviable, suggesting that Abf1[G4] is a bona fide motif.

Finally, we asked whether our motif-free constructs could reveal chemical principles to differentiate between viable and non-viable sequences. We divided our 67 viable and inviable sequences based on amino acid composition along two dimensions: charge (y axis; charge score) and the hydrophobicity of aliphatic and aromatic residues (x axis; binding score) (Fig. 4i). This enabled us to delineate between viable and inviable sequences (dashed line in Fig. 4i). In doing so, we noted two outliers—Gal4[768–881] and Pho4[1–249]—suggesting that these sequences may also possess bona fide motifs. We designed further variants to explicitly test this.

With Gal4[768–881], we generated three global shuffles (controlling for residue patterning; Extended Data Fig. 3e). Although each of these was viable, they showed variable growth phenotypes, ranging from wild type (+4) to relatively slow growing (+2) (Fig. 4j). Despite excluding the presence of a bona fide motif, these results illustrate how local sequence chemistry (beyond just composition) can influence function.

With Pho4$^{1–249}$, we generated two types of shuffle: (1) a global shuffle, where the entire sequence was shuffled; and (2) a segmental shuffle, in which subregions of ~40 amino acids were locally shuffled, preserving the overall chemical distribution (Fig. 4k). The segmental shuffle was designed explicitly to preserve local chemical composition and sequence patterning (Extended Data Fig. 3f). Unlike in Gal4$^{768–881}$, both of these shuffles were inviable, consistent with the presence of at least one bona fide motif. Although further investigation is needed to pinpoint the motif(s) (Discussion), this suggests that additional motifs beyond the essential motif could mediate Abf1 function.

In summary, although IDR-mediated interactions have historically been viewed through the lens of sequence-specific motif binding, here we show that a motif-free construct can fully rescue an IDR with an essential motif in the wild-type context if appropriate chemically specific interactions are provided. Furthermore, chemically specific interactions were sufficient and necessary for function, whereas sequence-specific interactions were essential only in a small window of chemical contexts and were otherwise insufficient. More generally, our results imply that chemical specificity could be conserved despite massive changes in linear sequence—an idea we sought to investigate further.

### Conservation of chemical specificity is invisible to alignment-based methods

Next we asked how conservation of chemical specificity might manifest across orthologues. To investigate this, we leveraged FINCHES, a recently developed approach to predict IDR-mediated chemical specificity from sequence[64]. FINCHES allowed us to take an IDR and a partner protein and predict which IDR regions or residues facilitate attractive and repulsive interactions (Fig. 5a). Using this, we evolved IDR2 in silico under the constraints of retaining chemically specific interactions with a putative partner.

Although the IDR2 binding partners essential for viability remain unknown, Abf1 was previously isolated in a stable trimeric complex with Rad7 and Rad16 (refs. 47,65). Moreover, the amino (N)-terminal IDR of Rad7 (Rad7$^{NTD}$) possesses several distinct regions that are predicted to interact with IDR2 (Extended Data Fig. 4f,g). We therefore performed in silico evolution of IDR2 under selection for chemically specific interactions with the Rad7$^{NTD}$ (Fig. 5b). Although Rad7$^{NTD}$ was used here, we emphasize that our conclusions do not depend on the specific partner (Extended Data Fig. 4h). As a control, we also evolved IDR2 without selection. In the limit of small sets of orthologues (20–30 sequences), we could not distinguish conserved versus unconserved sets of artificial orthologues via alignment-based methods, despite the conserved variants being evolved under strong selective pressure (Fig. 5c). This suggests that alignment-based approaches are unable to capture conservation of chemical specificity.

To test our computational results experimentally, we performed random mutagenesis to investigate the signatures of conservation in IDR2 (Fig. 5d). Using error-prone PCR, we generated a large number of length-matched randomly mutagenized versions of IDR2, tested their viability in vivo and then assessed per-residue conservation across viable sequences. High conservation of specific residues or regions would imply conservation of sequence specificity (Fig. 5e; top). Alternatively, a lack of regional conservation would be consistent with (although not unambiguously demonstrate) conservation of chemical specificity (Fig. 5e; bottom). This approach yielded 48 variants—14 viable and 34 inviable—with a distribution of point mutations (Fig. 5f; top). Across the 14 viable sequences generated by error-prone PCR, no region or residue was statistically enriched for conservation as assessed by alignment. Furthermore, a pool of mutationally matched sequences evolved in silico under conservation of chemical specificity produced data that were statistically indistinguishable from the experimental data (Fig. 5f; bottom). Although caveats remain (Discussion), these results further support the conclusion that alignment-based methods cannot capture conservation of chemical specificity.

Finally, across the yeast proteome, we identified ~450 IDRs from proteins that were poorly conserved as assessed by alignment, yet highly conserved in chemical specificity across various chemical interactions (Methods and Supplementary Table 9). These proteins are involved in essential cellular processes, including transcriptional regulation, DNA repair and cell signalling. We reiterate that conserved sequence features (for example, amino acid composition and patterning parameters) have been reported before[5,8,23,66]. Here we propose that conservation operates not at the level of sequence-intrinsic features but at the level of complementary chemical interfaces. Our work, and that of others, argues that conservation in yeast IDRs is widespread, albeit following different rules than conservation in folded domains.

### The IDR of Abf1 is not required for nucleosome barrier function in vitro

Next, we sought to determine which essential cellular function(s) IDR2 is responsible for. The best-known role of Abf1 is nucleosome barrier function. Abf1 binding to DNA generates nucleosome-free regions (NFRs) and an alignment point (barrier) against which regular nucleosome arrays become phased[36,67,68]. Such nucleosome positioning impacts transcription at promoters, both by NFR generation and by positioning the so-called +1 nucleosomes at transcription start sites[69]. The nucleosome barrier function of Abf1 relies on cooperation with ATP-dependent chromatin remodellers (for example, the yeast INO80 complex). The remodeller slides nucleosomes against DNA-bound Abf1 and spaces nucleosomes into regular phased arrays[67,70]. We hypothesized that this nucleosome barrier function involves the IDR of Abf1 (for example, by regulating or recruiting the remodeller).

We tested this hypothesis using in vitro genome-wide chromatin reconstitution, an approach previously used to demonstrate barrier function mechanism[67,70,71]. This assay employs purified factors and micrococcal nuclease digestion coupled with high-throughput sequencing (MNase-seq), allowing genome-wide mapping of nucleosome positions (Fig. 6a). To our surprise, neither an Abf1 variant with an inviable IDR (FUS$^{1–163}$12E) nor the lack of an IDR (ΔIDR1/2; that is, Abf1 DBD only) compromised the barrier function of Abf1 (Fig. 6b and Extended Data Fig. 5a–c). Furthermore, the IDR was not involved in INO80 remodeller recruitment in vitro (Extended Data Fig. 5b). In short, our results demonstrate that IDR2 is not required for barrier function, falsifying our first hypothesis.

### Inviable Abf1 variants mostly maintain NFRs and flanking nucleosomal arrays in vivo

Our next hypothesis was that inviable Abf1 IDR2 variants impair nucleosome organization in vivo, especially at essential genes. We turned to an ad hoc in vivo depletion approach using an Abf1 anchor-away system, which allows the testing of inviable Abf1 variants after depletion of wild-type Abf1 (Fig. 6c)[42,69,72]. Our ad hoc depletion system was validated by the recapitulation of previous observations. Upon Abf1 depletion, NFRs became partially filled and flanking nucleosomal arrays became disorganized (the peaks shifted towards the NFRs and peak-to-trough ratios decreased) at previously annotated Abf1 responder sites in promoters, in contrast with annotated non-responders (Extended Data Fig. 5d,e)[69]. Moreover, we reproduced gene expression changes (total RNA sequencing (RNA-seq)) for genes with responder Abf1 sites (defined by effects on nucleosome organization[69]) in their promoters[69] (Extended Data Fig. 5f).

Although we used MNase-seq to recapitulate previous work, we turned to next-generation DNA methylation footprinting coupled with nanopore sequencing (occupancy measurement by DNA methylation (ODM-seq)—also known as Fiber-seq; Fig. 6d)[71,73]. ODM-seq is a single-molecule technique that scores both occupied and non-occupied DNA (Fig. 6d), employs saturation of methylation, yields low variation among replicates and thereby reliably measures nucleosome positions and occupancy (Extended Data Fig. 6 and Supplementary Fig. 8).

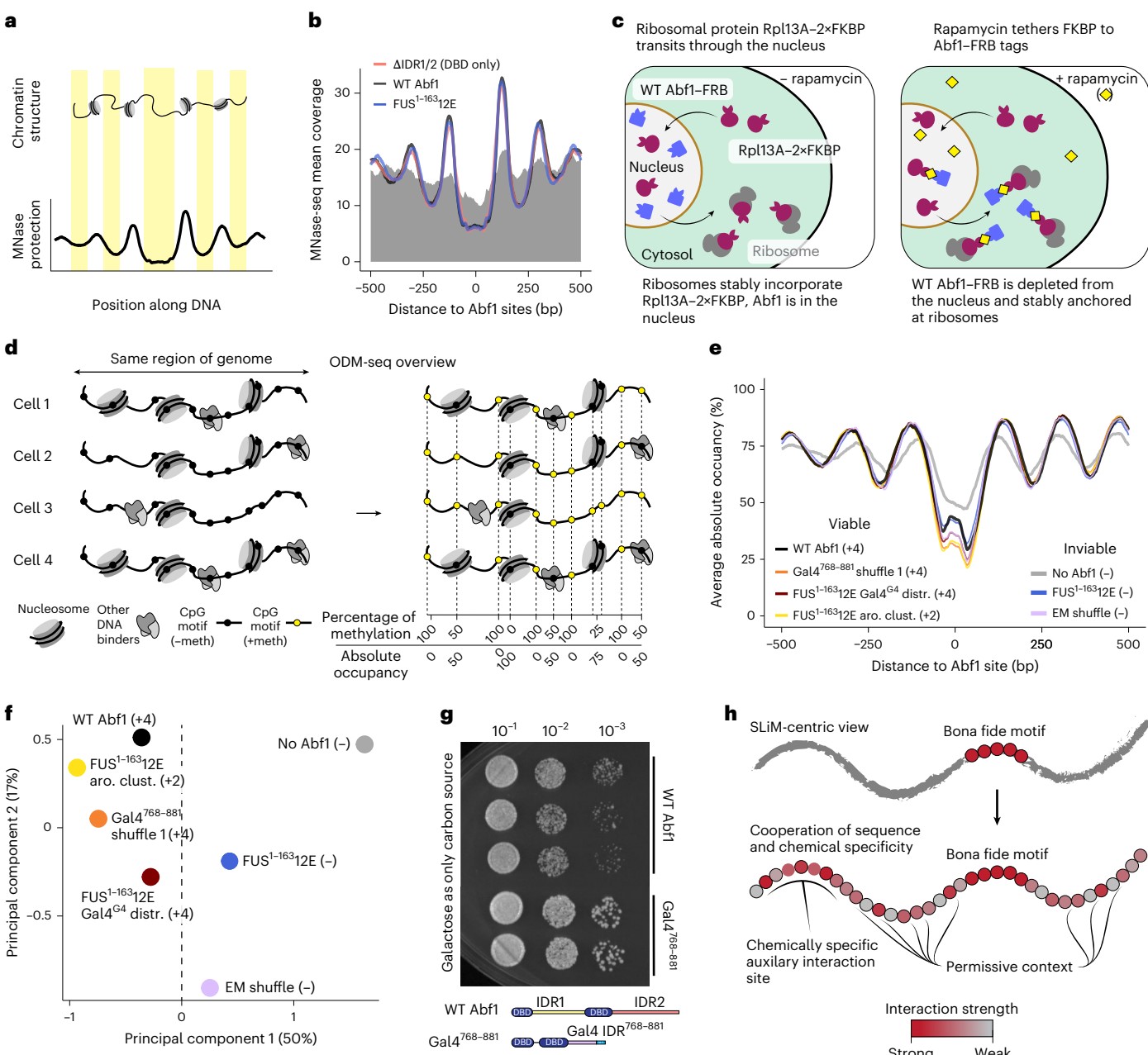

**Fig. 6 | Variant IDRs have diverse effects on Abf1 function. a**, Schematic of MNase-seq mapping. **b**, In vitro genome-wide nucleosome positioning assay at Abf1 sites (*n* = 379; class I + II sites[69]) revealed that Abf1 IDRs were not required for nucleosome positioning barrier function. The lines compare the indicated purified Abf1 variants in combination with ATP, reconstituted nucleosomes (salt gradient dialysis (SGD) chromatin) and the yeast chromatin remodeller INO80. The grey area reflects the experiment with wild-type Abf1, but in the absence of INO80. **c**, The anchor-away technique enables ad hoc depletion of wild-type Abf1 from the nucleus. The wild-type copy of Abf1 is genetically fused to FRB, whereas the ribosomal protein Rpl13A, which transits through the nucleus during ribosome assembly, is fused to the double FKBP12 tag (2×FKBP). After adding rapamycin, FRB and FKBP bind and nuclear Abf1 is sequestered at cytosolic ribosomes, leading to ad hoc nuclear depletion. **d**, Schematic of ODM-seq chromatin mapping. A change from black to yellow represents DNA methylation. **e**, Abf1 IDR variants did not increase occupancy in NFRs relative to wild-type Abf1 and did not grossly alter

nucleosome organization around Abf1 sites. The composite plots show ODM-seq data for the indicated Abf1 IDR variants aligned at *n* = 380 responder sites (classes I and II[69]; see also Extended Data Fig. 5d,e). **f**, Inviable Abf1 IDR variants or conditions lacking Abf1 affected +1 nucleosome positioning at essential genes differently than viable Abf1 IDR variants (growth rates in parentheses). PCA results are shown for all essential genes of ODM-seq data in the upstream flank of the in vivo +1 nucleosome (Extended Data Fig. 7g) averaged over two biological and two technical replicates (Extended Data Fig. 8a) each for the indicated Abf1 conditions. **g**, The Gal4 transactivation domain in the Gal4[768-881] construct allowed growth even under fully GAL-inducing conditions with galactose as the sole carbon source. Shown is a spot dilution assay after two days of incubation. **h**, Whereas a classical view might delineate IDRs into motifs and non-motifs, our work here—and that of others—supports an emerging model in which motif function is licensed by a permissive context and in which chemical specificity contributes additional interactions that can even compensate for the loss of a motif. aro. clust., aromatic clusters.

Combining the anchor-away technique with ODM-seq, we mapped genome-wide nucleosome organization for seven different conditions in the nucleus upon wild-type Abf1 depletion: wild-type Abf1 (positive control); no Abf1 (negative control); two inviable constructs (that is,

FUS[1-163]12E (Fig. 3i) and essential motif shuffle (LS-12; Fig. 3c)); and three viable constructs with either a wild type-like growth rate of +4 (FUS[1-163]12E + Gal4[G4] distr. (Fig. 4e) or Gal4[768-881] shuffle 1 (Fig. 4j)) or a compromised growth rate of +2 (FUS[1-163]12E aromatic clusters (Fig. 4g)).

Both viable and inviable Abf1 IDR variants generated nucleosome organizations at previously annotated promoter Abf1 responder sites that were very similar to wild-type Abf1 and different from those generated in the absence of Abf1 (Fig. 6e). This suggested that both NFR generation or maintenance and nucleosome positioning in vivo mainly relied on the Abf1 DBD and less on the IDR properties as seen in vitro (Fig. 6b). Nonetheless, we note that composite plots of many genes may obscure small effects and/or effects at few genes.

### Abf1 IDR is required for +1 nucleosome positioning at essential genes

As annotated Abf1 responder sites did not reveal gross differences in nucleosome organization between viable and inviable constructs, we wondered whether we were missing relevant effects by focusing only on these sites. A comprehensive comparison of annotated Abf1 sites in yeast—identified either via various in vivo mapping techniques or consensus DNA sequence motifs (position weight matrices (PWMs))—revealed a striking incongruency (Extended Data Fig. 7a,b). This analysis also recapitulated the previous conclusion that binding sites mapped in vivo need not contain a PWM (Extended Data Fig. 7c)[41,74]. Furthermore, recent work by Mahendrawada et al.[74] identified many genes whose expression changes in response to transcription factor ablation, even if those genes lack binding sites for the ablated transcription factor in their promoters. We plotted our ODM-seq data of wild-type versus no Abf1 at genes annotated by Mahendrawada et al.[74] as: (1) only having Abf1 binding sites in their promoters; (2) only responding to Abf1 ablation; or (3) both being responders and having Abf1 promoter sites. Indeed, transcription response, but not necessarily Abf1 sites, correlated with altered nucleosome organization (Extended Data Fig. 7d–f). This confirmed, on the previously unassessed level of nucleosome organization, that a response to Abf1 ablation need not be linked to an Abf1 site. It also confirmed that alterations in nucleosome organization at the NFR and +1 nucleosome correlate with gene expression changes[38,69,75,76]. Therefore, we searched in our ODM-seq data of Abf1 variants not for Abf1 sites around which nucleosome organization was affected, but for genes where nucleosome organization was affected at NFRs or +1 nucleosomes.

As gleaned from the composite plots comparing wild-type Abf1 and no Abf1, we looked for changes in: (1) the NFR region, which may reflect differential binding of nucleosomes or transcription machinery (for example, a preinitiation complex); (2) the upstream flank of the +1 nucleosome, which usually contains the transcription start site[41,77] and where even small shifts of nucleosome positions may impact transcription regulation[38,69,75,76,78]; and (3) the downstream flank of the +1 nucleosome, which also reflects shifted positions (Extended Data Fig. 7g). Given our focus on essential genes, we performed principal component analyses (PCAs) of ODM-seq data for wild-type Abf1, no Abf1 and Abf1 variants in these three regions of essential genes. Strikingly, PCA clearly separated inviable from viable conditions for the upstream flank of the +1 nucleosomes of essential genes (Fig. 6f) and the downstream flank, but not for the NFR region (Extended Data Fig. 8). We concluded that proper +1 nucleosome positioning—as a proxy for proper gene regulation—at essential genes depended on Abf1 IDR, in a manner incompatible with the inviable variants or the absence of Abf1.

### Abf1 can accommodate a strong and transcriptionally active activation domain

Abf1 lacks a transactivation domain[29] and participates in the insulation of transcription regulation[41,79]. It was therefore unclear a priori whether a GRF such as Abf1 could tolerate bearing a strong transactivation domain. We identified several viable Abf1 constructs with well-known transactivation domains, especially the Gal4[768–881] construct with the strong Gal4 transactivation domain (Fig. 3a). Nonetheless, it was possible that the Gal4 activation domain, including the short Gal4[G4] activation domain sequence present in several of our constructs, was

repressed by Gal80 in our glucose-containing media[80]. We confirmed that the Gal4[768–881] construct still supported viability with galactose as the sole carbon source (that is, with the GAL regulon fully induced; Fig. 6g). We concluded that Abf1's essential GRF function was compatible with a strong transactivation potential.

## Discussion

IDR-mediated interactions are often seen to be driven by sequence-specific binding motifs (for example, SLiMs) or by distributed multivalent interactions. These interaction modes are determined by sequence specificity and chemical specificity, respectively, and both are functionally important[17,21,81–83]. Surprisingly, we uncovered here that—at least in Abf1—these two modes were compensatory for one another. Using viability as a readout for function, a weak context could be compensated for by introducing a motif (Fig. 3) and, more surprisingly, the absence of a motif could be compensated for by the gain of context strength (Fig. 4). Importantly, we showed that testing motif shuffling or distribution was essential to identify bona fide motifs.

We interpreted our results using a 2D binding landscape, where the function of IDR2 reflects the interoperable combination of motif- and context-dependent binding (Fig. 4b). Based on our molecular understanding, we rationally designed new IDRs that were functional, albeit dramatically different from the wild-type sequence (Fig. 4). Beyond our work here, biophysical hints for a model in which sequence and chemical specificity cooperate in driving IDR-mediated binding have been found in in vitro reconstituted systems[18,19,84–88]. Finally, our work suggests plausible rules for evolutionary conservation in IDRs. Although our in silico and in vitro evolution experiments do not definitively prove one model over another, they are consistent with the conservation of chemical specificity shaping sequence variation in IDRs. Moreover, proteome-wide analyses identified hundreds of examples in which such conservation indicated conserved intermolecular interactions despite large changes in amino acid sequences (Supplementary Table 9).

We note several limitations of our work. First, we posit that if two IDRs offered equivalent growth phenotypes when provided in a protein where alteration or deletion of the same IDR rendered yeast inviable, these IDRs were functionally equivalent. This definition was empirically determined by our assay. However, different IDRs may confer viability in different ways by interacting with different partners. As the Abf1 interactome is revealed in future work, this question can be answered more directly. Second, we focus here on a minimal system lacking IDR1. However, assessing full-length IDRs (as opposed to just IDR2) of Abf1 orthologues may reveal motif migration within IDRs (see below). Third, IDR2 was important for +1 nucleosome positioning in vivo—especially at essential genes (Fig. 6f)—by a so far unknown mechanism beyond a pure nucleosome barrier function. In vivo +1 nucleosome positioning depends on combinations of barriers, such as Abf1, with remodellers, and possibly the transcription machinery[67,89,90]. We speculate that the interplay with transcriptional machinery may depend on IDR2, but we showed that the pure barrier function did not (Fig. 6b). Furthermore, +1 nucleosome positioning may involve a role of GRFs in torsional insulation[91] or 3D chromatin organization[92], which may also affect genes without Abf1 sites, for which we could not distinguish direct from indirect effects. The distinction between direct and indirect effects, while mechanistically important, does not matter for the distinction between viable and inviable outcomes.

Our ODM-seq data for Abf1 variants in vivo demonstrate that the function of GRFs such as Abf1 does not solely entail generating NFRs by binding to specific sites. Although NFR maintenance was grossly impaired upon Abf1 depletion, as described[69], even inviable Abf1 constructs maintained NFRs (Fig. 6e). Instead, the combination of NFRs and proper organization of flanking nucleosomes, especially +1 nucleosomes, appeared to constitute Abf1 function, especially at essential genes (Fig. 6f). This parallels with recent findings in the context of replication[93] and 3D genome organization[92], where not

just NFRs alone but the combination of NFRs and phased regular arrays was key.

Over the past decade, there has been a growing appreciation for the intersection of sequence and chemical specificity in the context of IDR function. The chemical determinants of IDR function have been explored in various contexts[5,8,29,63,64,66,94–101], yet how these properties influence the evolutionary landscape of disordered regions remains less clear. Our work suggests that compensatory changes, facilitated by chemical specificity, may buffer the loss of motifs. Our in silico and in vitro evolution experiments implied that the conservation of chemical specificity may be almost invisible in alignment-based analyses. This motivates novel interpretations of conservation in IDRs[5,22–24,66,102,103].

Functional molecular interactions generally involve specificity, facilitating interactions with relevant partners and preventing deleterious off-target binding. Shape complementarity and chemical compatibility typically mediate specificity in folded domains and, to a lesser extent, SLiMs[15,104]. Here we find that even for an IDR that depends on a bona fide motif, alternative sequences with rationally interpretable chemical features can uphold function without any motif. Whether motif-free variants interact with the same partners as motif-containing variants (our preferred interpretation) or with different partners without obvious phenotypic consequences (an alternative interpretation) remains to be seen. However, we speculate that this context-only mode, driven by just chemical specificity, offers an evolutionarily plastic mode of molecular recognition (Fig. 6h)[8,17,20]. Although work here does not strongly support a homotypic phase-separation-based model to underlie Abf1 function (Extended Data Fig. 9), it is consistent with a model in which Abf1 is recruited to specific loci in the nucleus through chemically specific interactions[105–107]. With this in mind, we propose that chemically specific recognition could tune or even determine recruitment or interaction with molecular assemblies without the need for strict stoichiometry (for example, biomolecular condensates)[108,109], providing an ideal paradigm for evolutionarily adaptable multicomponent cellular assemblies. Finally, our work demonstrates that nothing can be assumed when interrogating motifs; shuffling and/or distribution are required tests to confirm whether the effect of a putative motif is mediated by sequence or chemical specificity (Extended Data Fig. 9h,i).

The importance of context in relation to motif function has been seen in other systems[61,82,110]. If context can tune motif function, we might expect a rational evolution experiment to reveal compensatory mutations in action. To test this, we took a sequence generated by random mutagenesis (NCS-21) with 55 point mutations and a +3 growth phenotype. We reverted six mutations that had introduced de novo hydrophobic residues outside the essential motif to their original polar amino acids. Although this variant was more similar to the wild type in sequence, it showed severely impaired growth (+1 growth phenotype) (Supplementary Fig. 2). Therefore, NCS-21 provided a direct example of compensatory mutations offsetting otherwise deleterious mutations.

Whereas full-length Abf1 from *K. lactis* confers functionality in *S. cerevisiae*, the *K. lactis* IDR2 does not. More broadly, we found limited functional conservation of IDR2 across orthologues, despite conservation of amino acid composition and sequence features (Fig. 2d and Supplementary Fig. 9). There are several (non-mutually exclusive) explanations for this result. First, interaction networks co-evolve by coupled evolutionary changes, such that motifs in orthologues may be incompatible with partners in *S. cerevisiae*. Second, functionally important features can relocate across an IDR-containing protein. The specific location of a binding motif in the protein may be relatively unimportant, such that motifs can be lost from one region and emerge in another. An intriguing prediction from our work is that motifs are expected to appear and disappear within a given IDR, a prediction supported by extant work on ex nihilo motif evolution[111].

Although the essential motif in Abf1 appeared to be poorly conserved by alignment, we still expect evolutionarily conserved motifs to be important and ubiquitous across proteomes. Accordingly, we identified thousands of short, conserved hydrophobic sub-sequences within IDRs, with almost twice as many in essential versus non-essential proteins (Extended Data Fig. 10 and Supplementary Tables 10–14). We also identified many IDRs with low sequence conservation but extremely high chemical specificity (Supplementary Table 9). As a corollary, we wondered whether other non-conserved regions of transient structure may be found in the yeast proteome and, upon analysis, identified 963 short (<40-residue) subregions in IDRs with predicted transient helicity (Supplementary Table 17). This set of de novo predictions included the previously identified Pho4 activation domain (Pho4[69–94]) and four separate subregions in the N-terminal IDR of Reb1, both of which conferred viability (Fig. 3a). Moreover, global and segmental shuffles (in which local subregions were internally shuffled) of Pho4[1–249] were non-viable (Fig. 4k), suggesting a bona fide motif within Pho4[1–249]. Although many such regions may be inert, others may offer specific binding interfaces (as was the case for Abf1) or specific helical regions identified in transactivation domains[99,112].

Finally, we focused on the function measured by viability in *S. cerevisiae* growing under low-challenge laboratory conditions. This probably missed some facets of Abf1 function and, accordingly, the importance of other Abf1 regions and features. For example, ongoing work implied large-scale IDR-dependent remodelling of transcription during glucose starvation with IDRs as intrinsic sensors of intracellular state[113]. Tuning IDR chemistry (via post-translational modifications, changes in protonation state or changes in side chain solvation properties) offers an attractive route to modulate interaction specificity and thereby regulatory output.

## Online content

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

¹Biomedical Center, Division of Molecular Biology, Faculty of Medicine, LMU Munich, Martinsried, Germany. ²Physics of Complex Biosystems, Technical University of Munich, Garching, Germany. ³Gene Center, Department of Biochemistry, LMU Munich, Munich, Germany. ⁴Department of Biochemistry and Molecular Biophysics, Washington University School of Medicine, St. Louis, MO, USA. ⁵Center for Biomolecular Condensates, Washington University, St. Louis, MO, USA. ⁶Core Facility Bioinformatics, Biomedical Center, Faculty of Medicine, LMU Munich, Martinsried, Germany. ⁷Institute of Functional Epigenetics, Helmholtz Zentrum München, Neuherberg, Germany. ⁸Present address: Moderna, Cambridge, MA, USA. ⁹Present address: Gene Center, Department of Biochemistry, LMU Munich, Munich, Germany. ¹⁰Present address: Proteros Biostructures, Martinsried, Germany. ¹¹Present address: Scuola Superiore Sant'Anna, Pisa, Italy. ¹²Present address: Program of Applied Epigenetics, Josep Carreras Leukaemia Research Institute (IJC), Badalona, Spain. ¹³These authors contributed equally: Iris Langstein-Skora, Andrea Schmid, Frauke Huth. ✉e-mail: pkorber@lmu.de; alex.holehouse@wustl.edu

## Methods

### Plasmids and yeast strains for 5-FOA plasmid shuffling assay and growth rate scoring by spot assay

The plasmid pRS416-Abf1 contained the wild-type *ABF1* gene (a PCR fragment from 690 base pairs (bp) upstream and 170 bp downstream of the coding region, generated with primers *pRS416_Abf1_for* and *pRS416_Abf1_rev* (Supplementary Table 2) and genomic DNA of BY4741 as a template) inserted into the EcoRI and BamHI sites of plasmid pRS416 (ref. 114). Plasmid pRS315-Abf1 contained the equivalent insert to plasmid pRS416-Abf1 (a PCR fragment amplified with primers *pRS315_Abf1_for* and *pRS315_Abf1_rev* (Supplementary Table 2) and genomic DNA of BY4741 as a template), but it was inserted via BamHI and HindIII in plasmid pRS315 (ref. 114).

Rationally designed *abf1* mutant constructs were generated via Gibson cloning[115,116] using the PCR primers and templates in Supplementary Table 3 for insert and backbone fragments. The templates for inserts were mainly ordered by gene synthesis (Thermo Fisher Scientific; Supplementary Table 3) or amplified from genomic DNA of strain BY4741. All constructs in pRS315 contained an SV40 NLS (amino acid sequence: PKKKRKV) at the C terminus. All rationally designed constructs in pRS315, except ΔIDR1, contained the 3xFlag tag introduced via Gibson cloning using primers and template, as specified in Supplementary Tables 2 and 3.

For PCR, Phusion polymerase (New England Biolabs) and thermo cycling conditions were used according to the manufacturer's recommendations. A pool of inserts for randomly mutagenized *abf1* constructs was generated by error-prone PCR (dNTP-Mutagenesis Kit; PP-101; Jena Bioscience) with primers *N2-IDR2-N*dC* and *N3-IDR2-N*dC* and assembled via Gibson cloning into a backbone PCR fragment generated with primers *N1-IDR2-N*dC* and *N4-IDR2-N*dC* (Supplementary Table 4).

For both the mutagenized insert and the plasmid backbone PCR, pRS315-ΔIDR1&IDR2$^{449-662}$ was used as a template. After transformation of the Gibson assembly reaction into *Escherichia coli* strain DH5α and DNA isolation (NucleoSpin Plasmid (NoLid); 740499.250; Macherey-Nagel) from single bacterial colonies, the insert DNA sequences were confirmed by Sanger sequencing with the primers listed in Supplementary Table 5.

The parent diploid strain with heterozygous deletion of the *ABF1* gene (Y24962 (BY4743; MATa/MATα; *his3Δ1/his3Δ1*; *leu2Δ0/leu2Δ0*; *LYS2/lys2Δ0*; *met15Δ0/MET15*; *ura3Δ0/ura3Δ0*; *YKL112w/YKL112w::kanMX4*)) was obtained from EUROSCARF (www.euroscarf.de). Y24962 was transformed with plasmid pRS416-Abf1 and sporulated in liquid culture (1% potassium acetate, 5 μg ml$^{-1}$ histidine and 25 μg ml$^{-1}$ leucine). Sporulation and tetrad dissection were performed according to Dunham et al.[117]. Spores with both the *abf1::kanMX* deletion allele and plasmid pRS416-Abf1 were selected after tetrad dissection (MSM 400; Singer Instruments) on yeast nitrogen base (YNB) − uracil plates and sub-sequent growth on YPDA + kanamycin (0.2 mg ml$^{-1}$) selection medium.

Three strains originating from independent such spores were kept for further use: 7-A2 (MATa), 7-C5 (MATα) and 7-D4 (MATa). Their mating type was determined by colony PCR with primers as published[118]. These strains were transformed with plasmids (pRS315-[*abf1* construct name]) bearing *abf1* mutant constructs, as detailed in Supplementary Tables 1–4, and selected on YNB −uracil − leucine medium. The insert region of every plasmid batch was checked by Sanger sequencing before transformation into yeast. For each construct, independent single colonies after transformation (for which the numbers of replicates (always ≥3) are given in Supplementary Fig. 1 and Supplementary Table 1) were re-streaked for single colonies on YNB −uracil − leucine selection medium, then single colonies thereof were streaked out as patches on the same medium. From these patches, strains were streaked again as patches on 5-FOA − leucine plates (0.67% YNB, 0.2% Synthetic Complete Supplement Mixture − uracil − leucine

drop-out mix, 2% glucose, 50 μg ml$^{-1}$ uracil and 0.1% 5-FOA (Toronto Research Chemicals (F595000) or Diagnostic Chemicals Limited (1555)) and incubated for 3 days at 30 °C. Viability on 5-FOA plates was visually scored by comparison with known viable or inviable strains on the same plate.

Inviable strains were distinguished from viable strains because they showed a smeary appearance and, at most, sparse single colonies. In contrast, viable strains grew as one more or less dense patch. (Supplementary Fig. 1 and Supplementary Table 1). Sparse single colonies that came up on 5-FOA plates (Supplementary Fig. 1) were shown to bear a plasmid of the wrong size or sequence by colony PCR (Supplementary Table 8), usually reflecting recombination between pRS416-Abf1 and pRS315-[*abf1* construct name] so that at least parts of wild-type Abf1 IDR2 were gained that supported viability. Three independent single colonies of strains with constructs that scored as inviable on 5-FOA plates were stored as glycerol cultures from the patches on YNB − uracil − leucine medium.

Constructs viable on 5-FOA plates were tested for uracil auxotrophy on YNB − uracil medium and for bearing the correct plasmid by colony PCR using the primers given in Supplementary Table 8. The colony PCR product was often further validated by Sanger sequencing using the primers given in Supplementary Table 5. For each construct, three validated strains from independent single colonies that were viable on 5-FOA and inviable on −uracil medium were stored as glycerol culture.

Yeast cells grown from these three glycerol cultures of independent clones for each viable Abf1 construct were scored for growth rate by spotting serial dilutions in water (1:10, 1:100, 1:1,000 and 1:10,000) of overnight cultures adjusted with yeast peptone dextrose adenine (YPDA) media to an optical density measured at a wavelength of 600 nm (OD$_{600}$) of 0.5 (Genesys 20 photometer; Thermo Fisher Scientific) on YPDA plates, then incubated at 30 °C for 1–3 days (Supplementary Fig. 2). Growth rates were scored visually into four broad categories (+1 to +4) compared with the rate of two independent clones of strain 7-A2 bearing plasmid pRS315-Abf1 spotted on the same plates and defined to have a growth rate score of +4 (Supplementary Fig. 2 and Supplementary Tables 1 and 7).

### ChIP and quantitative real-time PCR

ChIP assays were performed for all inviable constructs bearing a Flag-tag and some viable constructs, as indicated by the respective ChIP values in Supplementary Table 1. Corresponding yeast strains were grown to logarithmic phase in YNB − uracil − leucine selection medium (inviable constructs or viable constructs together with the plasmid pRS416-Abf1) or in YNB − leucine selection medium (viable constructs) at 30 °C. One hundred OD$_{600}$ units (Zeiss PMQ II photometer) were crosslinked for 15 min at room temperature with a 1% final concentration of formaldehyde while shaking slowly, and crosslinking was quenched by adding a 250 mM final concentration of glycine for 10 min at room temperature while shaking slowly. Cells were harvested by centrifugation for 5 min at 4,000 r.p.m. (Heraeus Cryofuge 6000i) and 4 °C. Cell pellets were washed with ice-cold ST buffer (10 mM Tris-HCl (pH 7.5) and 150 mM NaCl) and washed twice in FA lysis buffer (50 mM HEPES-KOH (pH 7.5), 150 mM NaCl, 1 mM ethylenediaminetetraacetic acid (EDTA), 1% Triton X-100, 0.1% sodium deoxycholate and 0.1% sodium dodecyl sulfate (SDS)). Washed pellets were flash-frozen in liquid nitrogen and stored at −80 °C. For cell lysis, pellets of 50 OD$_{600}$ units were resuspended in 250 μl FA lysis buffer supplemented with protease inhibitors (1× cOmplete EDTA-free protease inhibitor cocktail (Roche) and 1 mM phenylmethylsulfonyl fluoride). Zirconia beads (350 μl 0.5 mm glass beads; 11079105; BioSpec Products) were added and cells were lysed in a bead beater (Precellys 24; Bertin Technologies). Lysis efficiency was controlled by comparing OD$_{600}$ values before and after bead beating and was 75–90%. Beads were removed by centrifugation, cell lysates were adjusted to 1.2 ml with FA lysis buffer, and chromatin was sheared by sonication at 4.5 °C (30 cycles; 30 s on

and 30 s off; high intensity; Bioruptor, Diagenode). Fragment lengths were usually around 1 kilobase as assessed in a Bioanalyzer after DNA purification (high-sensitivity DNA chip). If this fragmentation range was not met, the sample preparation was repeated, starting from a new culture. The chromatin solution was pre-cleared by incubation with 25 µl protein G beads for 1–2 h at 4 °C with soft rolling. For immunoprecipitation, 500 µl pre-cleared chromatin solution was incubated with or without (no antibody control) 2.5 µl anti-Flag M2 antibody (F3165; Sigma–Aldrich) overnight at 4 °C with soft rolling and for four further hours with 25 µl pre-blocked (10% bovine serum albumin) protein G beads. Beads were washed twice with high-salt FA lysis buffer (0.50 instead of 0.15 M NaCl), twice with ChIP Wash buffer (10 mM Tris-HCl (pH 8.0), 250 mM LiCl, 0.5% NP-40, 0.5% sodium deoxycholate and 1 mM EDTA) and once with TE buffer (10 mM Tris-HCl (pH 8.0) and 1 mM EDTA). Elution was done twice for 15 min at 65 °C while shaking in 100 µl ChIP Elution buffer (50 mM Tris-HCl (pH 7.5), 10 mM EDTA and 1% SDS). Eluates were pooled, and crosslinking was reversed by incubation overnight at 65 °C while shaking. 50 µl chromatin sample for input DNA was taken off after sonication, adjusted to 200 µl with ChIP Elution buffer and treated in parallel for DNA purification. Immunoprecipitated or input chromatin was treated with RNaseA (0.5 mg ml$^{-1}$) for 30 min at 37 °C, then with proteinase K (0.5 mg ml$^{-1}$) for 2 h at 37 °C while shaking. DNA was purified with AMPure beads (1.8 vol), according to the manufacturer's instructions.

Input DNA (1 µl; 1:100 dilution in water), immunoprecipitated DNA (1 µl) and mock-precipitated (no antibody) DNA (1 µl) were analysed in triplicate by quantitative real-time PCR (LightCycler 480 II; Roche) in a 10 µl total volume using the primers given in Supplementary Table 6 (at a final concentration of 3 µM) and 1× Fast SYBR Green Master Mix (Applied Biosystems). ChIP values were calculated from quantitative real-time PCR cycle threshold values as an average of PCR triplicate averages for each of the three amplicons with the Abf1 site divided by the average of PCR triplicate averages for each of the three amplicons without the Abf1 site.

For unknown reasons, some replicates of inviable constructs failed to score in the ChIP assay (that is, they had a ChIP value close to 1; Supplementary Table 1). Nonetheless, if at least one replicate was positive (with a ChIP value of > 10), this confirmed that the respective construct was expressed and imported into the nucleus and could bind specifically to Abf1 sites. ChIP values for inviable versus viable constructs were, on average, lower (Extended Data Fig. 2a). This was probably due to the competition between the untagged wild-type Abf1 and the Flag-tagged inviable construct for binding to Abf1 sites. This competition was unavoidable for inviable constructs, but usually not the case for viable constructs, as these were tested in the absence of the untagged wild-type Abf1. To confirm this, we tested two viable constructs in the presence of untagged wild-type Abf1 to mimic the competition between wild-type Abf1 and the Flag-tagged construct (marked in Extended Data Fig. 2a) and found that they indeed showed ChIP values that matched the expected values from the inviable constructs ($P$ = 0.24; Mann–Whitney $U$-test) in the range of the viable constructs.

### Abf1 anchor-away strains and growth conditions for RNA- and ODM-seq analyses

The Abf1 anchor-away strain ABF1-aa-V5 clone A9 was obtained from W. de Jonge and F. Holstege[42]. The strain was transformed with the same pRS315-based plasmids encoding wild-type Abf1, select Abf1 constructs or the empty pRS315 plasmid (no Abf1 control), as was used for the 5-FOA plasmid shuffle assays. At least three independent clones were isolated and stored as glycerol cultures after transformation and used for biologically independent replicate experiments. Anchor-away strains were grown to log phase (~OD$_{600}$ = 0.4; Genesys 20; Thermo Fisher Scientific) in YNB − leucine media, incubated for 75 min (or 60 min for one RNA-seq replicate each for wild-type Abf1, Abf1 IDR2, FUS$^{1-163}$12E and FUS$^{1-163}$ + Y/M) with 8 µg ml$^{-1}$ rapamycin

(Santa Cruz Biotechnology) and crosslinked for 5 min (for ODM-seq) with 1% formaldehyde at room temperature. Formaldehyde crosslinking was quenched for 15 min with 250 mM glycine at room temperature.

### RNA-seq

Total RNA was prepared after Abf1 anchor-away induction with an RNeasy kit (Qiagen) according to the manufacturer's protocol. Briefly, four OD$_{600}$ units (Genesys 20; Thermo Fisher Scientific) of cells were washed with phosphate-buffered saline (50 mM phosphate buffer (pH 7) and 150 mM NaCl) and flash-frozen in liquid nitrogen. The cell wall was broken up by incubation with beta-mercaptoethanol and zirconia bead beating (6 × 3 min at 30 Hz with 3 min breaks; Precellys; Bertin Technologies). The cell lysate was cleared by centrifugation, mixed with ethanol and bound to a column provided in the kit. We also employed the optional DNase I digestion on the column. RNA was washed on the resin, eluted and digested using DNase I (Qiagen) in solution. After ethanol precipitation, the total RNA preparation was converted into sequencing libraries with the NEBNext Ultra II directional RNA library prep kit for Illumina.

### Cloning, expression, purification and characterization of M.SssI DNA methyltransferase

**DNA methyltransferase cloning.** The DNA sequence coding for the M.SssI methyltransferase, where the TGA stop codons coding for tryptophan in the original organism were replaced by TGG codons, was PCR amplified from the plasmid pCMV-FLAG-4azf-M.SssI (a gift from K. Ford at Kings College London) and cloned into the NcoI and SalI sites of the pBAD24 plasmid (87399; American Type Culture Collection) to achieve tight transcriptional control by the *E. coli* arabinose-regulated *araBAD* promoter. An in-frame C-terminal 6xHis-tag was introduced via the 3′ primer during the PCR. The resulting expression plasmid is called pBAD24-M.SssI-His6 and was transformed (selection on plates containing Luria–Bertani growth medium supplemented with ampicillin (LB amp) plus 0.2% glucose) into *E. coli* One Shot TOP10 (Thermo Fisher Scientific), yielding the expression strain for M.SssI expression.

A single colony streaked out on LB amp plates containing 0.2% glucose was used to inoculate a 100 ml overnight culture in LB amp with 0.2% glucose at 37 °C and 130 r.p.m. Six 1 l volumes of LB amp media (without glucose) were inoculated with 10 ml each of the overnight culture and incubated at 37 °C and 130 r.p.m. until the OD$_{600}$ reached 0.5 (Genesys 20; Thermo Fisher Scientific). Expression was induced by the addition of 0.0002% arabinose for 4 h at 37 °C and 130 r.p.m. Cells were harvested by centrifugation (4,000 r.p.m.; 30 min; 4 °C; Avanti JXN-26; JLA-8.1000 rotor; Beckman Coulter), resuspended and combined in 75 ml lysis buffer (50 mM Tris-HCl (pH 8) and 300 mM NaCl). Aliquots of 35 ml were snap-frozen in liquid nitrogen and stored at −70 °C.

**DNA methyltransferase purification.** Purification started with one frozen cell pellet aliquot that was thawed at room temperature, topped up to 50 ml with lysis buffer and supplemented with 1 mg ml$^{-1}$ chicken egg-white lysozyme (28260; SERVA) and protease inhibitors (0.7 µg ml$^{-1}$ pepstatin A, 1 µg ml$^{-1}$ leupeptin, 1 µg ml$^{-1}$ aprotinin and 0.2 mM phenylmethylsulfonyl fluoride), incubated on ice for 30 min and then sonicated (six cycles of 10 s bursts and 10 s breaks at 50% peak power; Branson 250D sonifier). Lysed cells were centrifuged for 1 h at 4 °C and 20,000*g* (Optima XPN-80; Ti-45 rotor; Beckman Coulter). During centrifugation, 5 ml slurry of Ni-NTA agarose (745400.25; Macherey-Nagel) was washed twice with 20 ml lysis buffer (1 min; 1,000 r.p.m.; 4 °C; Eppendorf 5810R centrifugation for collecting the agarose) and resuspended in 30 ml lysis buffer and then distributed into three 10 ml aliquots. Next, Ni-NTA beads were collected by centrifugation and the supernatant discarded. After ultracentrifugation, the lysed cells' supernatant was evenly distributed onto the pre-washed Ni-NTA beads and incubated for 1.5 h at 4 °C on a rotating wheel. Beads were collected by centrifugation (2 min; 1,000 r.p.m.; 4 °C; Eppendorf 5810R), the

supernatant was discarded and the beads were resuspended in 20 ml cold lysis buffer with protease inhibitors then transferred to a gravity flow column (Econo-Pac; 7321010; Bio-Rad) at room temperature. The column was then washed by gravity flow with 100 ml cold lysis buffer, 20 ml cold wash buffer (50 mM Tris-HCl (pH 8), 250 mM NaCl and 20 mM imidazole-HCl (pH 6.6)). After the addition of 2 ml cold elution buffer (50 mM Tris-HCl (pH 8), 250 mM NaCl and 300 mM imidazole-HCl (pH 6.6)), the column was closed, lightly vortexed and incubated on ice for 10 min with light vortexing every 2 min. One 2 ml fraction was collected on ice by gravity flow. Additionally, three 1 ml fractions were collected after the addition of 3 ml elution buffer. Fractions of the purification steps were analysed by SDS polyacrylamide gel electrophoresis (SDS-PAGE) (Supplementary Fig. 10a and uncropped Supplementary Fig. 13a).

Fractions from the column enriched for the 42 kDa M.SssI methyltransferase were pooled and directly loaded in the cold room onto a HiLoad 16/600 Superdex 200 pg exclusion column (Cytiva) equilibrated with size exclusion buffer (20 mM Tris-HCl (pH 8), 500 mM NaCl, 0.1 mM EDTA and 0.1 mM dithiothreitol (DTT)). Elution was with size exclusion buffer at a 0.5 ml min$^{-1}$ flow rate, and 2 ml fractions were collected in the cold. Fractions were analysed by SDS-PAGE (Supplementary Fig. 10b and uncropped Supplementary Fig. 13b) and fractions were enriched for M.SssI combined and dialysed against dialysis buffer (10 mM Tris-HCl (pH 8), 0.2 mM EDTA, 1 mM DTT and 50% glycerol), which also concentrated the protein solution. The dialysed and concentrated M.SssI solution was analysed again by SDS-PAGE (Supplementary Fig. 10c and uncropped Supplementary Fig. 13c) and the methyltransferase activity was quantified by restriction enzyme activity assay. Aliquots corresponding to ~440 units were snap-frozen in liquid nitrogen and stored at −80 °C.

**Quantification of CpG-specific DNA methyltransferase activity.**
Plasmid pUC19-ftz (500 ng; containing a fragment of the *Drosophila* ftz locus from sequence TAGTTTCCTAATGAT to GCCGAAGATGATGCT cloned into the pUC19 multiple cloning site) was incubated in a 122 μl final volume with increasing amounts of purified M.SssI solution or commercially acquired M.SssI (M0226; New England Biolabs (NEB)) in NEBuffer 2 (NEB) supplemented with 160 μM *S*-adenosyl-L-methionine for 1 h at 25 °C. The reaction was stopped by the addition of a one-tenth volume of 10× stop buffer (50 mM Tris-HCl (pH 7.5), 4% SDS and 100 mM EDTA) and incubated with 5% proteinase K. After incubation for 1 h overnight at 37 °C, a one-fifth volume of 5 M NaClO₄ was added to the solution, followed by an equal volume of phenol (77607; Sigma–Aldrich), vortexing, an equal volume of chloroform:isoamylalcohol (24:1) and then more vortexing. After this, the phases were separated by centrifugation (Hettich Microliter). The DNA was recovered from the aqueous phase by ethanol precipitation and resuspended in 37 μl rCutSmart Buffer (NEB). The DNA was incubated with 30 units each of BamHI-HF and PvuI-HF (NEB) for 1 h at 37 °C. The complete reaction was loaded onto a 1% agarose gel in 1× Tris-acetate-EDTA (TAE) buffer and electrophoresed for 1.5 h at 100 V. The DNA in the gel was stained with 5 μl Midori Green solution (Nippon Genetics Europe) per 100 ml gel for a few minutes and directly imaged (ChemiDoc; Bio-Rad) (Supplementary Fig. 11a and uncropped Supplementary Fig. 14a,b). BamHI linearized the 5,769 bp pUC-ftz plasmid regardless of CpG methylation. PvuI cut in a CpG-methylation-sensitive way and yielded 2,544-, 2,329- and 896-bp fragments in combination with the BamHI cut if cleavage was complete, but only yielded the 5,769-bp linearized plasmid band if complete CpG methylation blocked PvuI digestion. Comparison of the degree of blocked PvuI digestion between reactions with purified and commercial M.SssI (NEB) allowed us to quantify the specific activity of the purified M.SssI according to the unit definition by NEB.

The M.SssI preparation was checked for contaminating nucleases by incubation with 500 ng supercoiled pUC19-ftz plasmid in either rCutSmart or NEBuffer 2 (NEB) for 2 h or overnight at 25 °C. The reaction was electrophoresed in 1% agarose 1× TAE gel, stained and imaged. The persistence of the supercoiled form demonstrated the lack of contaminating nucleases (Supplementary Fig. 11b and uncropped Supplementary Fig. 14c).

**Absolute nucleosome occupancy mapping by ODM-seq**
Chromatin was prepared from formaldehyde-crosslinked yeast cultures as described[71]. Chromatin corresponding to a 0.1 g wet cell pellet was washed in cold Kladde buffer (20 mM HEPES-NaOH (pH 7.5), 70 mM NaCl, 0.25 mM EDTA (pH 8.0), 0.5 mM EGTA (pH 8.0) and 0.5% (vol/vol) glycerol), resuspended in 870 μl Kladde buffer and incubated with 4,000 U M.SssI methyltransferase at 25 °C. After 90 min, 4,000 U M.SssI were added, together with the plasmid pUC19-ftz, which was included as a spike-in control. After 180 min, one 430 μl aliquot was removed, and plasmid pFMP233 (ref. [119]) was added to the remaining aliquot as a second spike-in control. The reaction was further incubated for 60 min (240 min total) at 25 °C. A similar degree of CpG methylation after 180 versus 240 min (Extended Data Fig. 6a), in combination with ongoing CpG methylation as probed by methylation of the spike-in plasmids (Extended Data Fig. 6b), indicated saturated CpG methylation. After DNA methylation, the reaction was stopped by adding a one-tenth volume of 10× stop buffer. The resulting DNA was purified by proteinase K digestion, phenolization and ethanol precipitation and then resuspended in TE buffer. Nanopore sequencing libraries were prepared and sequenced on a PromethION device by the Laboratory for Functional Genome Analysis at the Gene Center at LMU Munich.

**Cloning, expression and purification of wild-type Abf1 and the constructs FUS$^{1-163}$12E and ΔIDR1/2**
Expression plasmids for wild-type Abf1 and Abf1 variant constructs were constructed in pPROEX as described by Krietenstein et al.[67], but starting from the respective pRS315 plasmids with the respective Abf1 constructs and using the primers for backbone and insert PCR, respectively, which were combined by Gibson cloning using the following sequences: pPROEX_V_rev (AATTTGTCCATggGGATCCATGG); Abf1_I_for (GATCCCATGGACAAATTAGTCGTGAAT); Abf1_I_rev (ACCGCATGCCTCGAGCTActtatcgtcatcgtctttg); and pPROEX_V_for (CTCGAGGCATGCGGTACC).

Wild-type Abf1 and variant proteins were expressed using the BL21 Star (DE3) pLysS strain (Thermo Fisher Scientific). LB amp media (100 ml) was inoculated with a single colony and incubated overnight at 130 r.p.m. and 37 °C. Then, 20 ml of this preculture was used to inoculate 1 l LB amp and incubated at 130 r.p.m. and 37 °C until the OD₆₀₀ was 0.5–0.6 (Genesys 20; Themo Fisher Scientific). Isopropyl β-D-1-thiogalactopyranoside was added to a final concentration of 1 mM and incubated for 4 h at 37 °C and 130 r.p.m. Cells were collected by centrifugation (4,000 r.p.m.; 20 min; 4 °C; Avanti JXN-26; JLA-8.1000 rotor; Beckman Coulter), combined in a 50 ml Greiner tube in 30 ml lysis buffer with protease inhibitors and centrifuged (1 min; 1,000 r.p.m.; 4 °C; Eppendorf 5810R). The pellet was flash-frozen and stored at −70 °C.

For purification in the cold room, the cell pellet was thawed on ice, resuspended in 40 ml lysis buffer with 1 mg ml$^{-1}$ chicken egg-white lysozyme and incubated for 30 min on ice. Cell lysis was performed by sonication on ice (Branson 250D sonifier; six cycles of 10 s bursts and 10 s breaks at 50% peak power). The cell lysate was centrifuged at 45,000 r.p.m. for 1 h at 4 °C (Optima XPN-80; Ti-45 rotor; Beckman Coulter) and the supernatant was collected and transferred to a 50 ml Greiner tube, where it was mixed with pre-washed Ni-NTA resin (see above) equivalent to 2 ml slurry. This mixture was incubated for 1.5 h at 4 °C on a rotating wheel, then washed with 100 ml lysis buffer then 20 ml wash buffer and eluted with 5–10 ml elution buffer. The eluate was dialysed overnight at 4 °C against 1–2 l dialysis buffer II (20 mM HEPES-NaOH (pH 7.5), 70 mM NaCl, 0.1% Tween, 40% glycerol and 1 mM DTT) and concentrated if necessary (Amicon Ultra-4 Ultracells 10 or

30 K; Millipore). The dialysed and maybe concentrated sample, for the wild-type Abf1 and FUS[1–163]12E variants, was loaded onto a 1 ml HiTrap Q FF anion exchange chromatography column (Cytiva) equilibrated in low-salt buffer (50 mM Tris-HCl (pH 8.0), 80 mM NaCl, 10% glycerol, 1 mM DTT and 1 mM EDTA) and washed to baseline with low-salt buffer. Elution was by linear gradient from 0–100% high-salt buffer (50 mM Tris-HCl (pH 8.0), 350 mM NaCl, 10% glycerol, 1 mM DTT and 1 mM EDTA) over 20 column volumes. The dialysed and maybe concentrated ΔIDR1/2 construct was loaded onto a 1 ml HiTrap Heparin HP column (Cytiva) equilibrated with heparin buffer A (25 mM Tris-HCl (pH 7.6), 10% glycerol, 1 mM DTT, 50 mM NaCl and 1 mM EDTA), washed to baseline with heparin buffer A and eluted by linear gradient from 0–100% of heparin buffer B (25 mM Tris-HCl (pH 7.6), 10% glycerol, 1 mM DTT, 1 M NaCl and 1 mM EDTA) over 30 column volumes.

The purified proteins (Supplementary Fig. 12 and uncropped Supplementary Fig. 15) were dialysed overnight against storage buffer (20 mM HEPES-NaOH (pH 7.5), 350 mM NaCl, 0.1% Tween, 40% glycerol and 1 mM DTT), aliquoted, flash-frozen in liquid nitrogen and stored at −80 °C.

## Genome-wide chromatin reconstitution and remodelling assay

Genome-wide chromatin reconstitution, remodelling assay, MNase-seq assay and Illumina sequencing were done as described by Chacin et al.[93] for Fig. 6b and Extended Data Fig. 5a,c, and as described by Oberbeckmann et al.[89] (but without ammonium sulfate in the final buffer) for Extended Data Fig. 5b, with the exception that: (1) Illumina sequencing was in paired-end mode; and (2) for the samples shown in Fig. 6b, recombinant human octamers were used, reconstituted from H2A/H2B dimers and H3/H4 tetramers according to Kunert et al.[120]. Recombinant human histones were obtained from the Histone Source (Colorado State University). The concentration of INO80 was 10 nM unless stated differently in Extended Data Fig. 5b. To verify the robustness of this assay, we repeated the experiment (as seen in Fig. 6b) using *Drosophila* embryo histone octamers (Extended Data Fig. 5c), yielding analogous results. Wild-type and variant Abf1 proteins were used at 30 nM in Fig. 6b and Extended Data Fig. 5b and 20 nM in Extended Data Fig. 5c. We showed that INO80 could not generate nucleosomal arrays phased to Abf1 sites in the absence of Abf1 (Extended Data Fig. 5a).

## Data processing for RNA-seq

Raw sequence reads were preprocessed with a Snakemake pipeline that involved several steps, including quality control, alignment of reads to the *S. cerevisiae* sacCer3 (R64-1-1 build) genome using the STAR aligner[121] and quantification of gene expression via RSEM[122]. Rule-based directives in Snakemake made it easier to perform key preprocessing steps, ensuring the effective management of computational resources and data dependencies. The processed data were imported into RStudio, where a summarized experiment object was created to encapsulate gene expression metrics alongside sample metadata. This structured data object made robust data manipulation and subset selection in R easier. Before creating plots, the transcripts per million data were $\log_2$ transformed to normalize expression levels, improving the statistical reliability of comparisons. The ggplot2 (refs. 123,124) package was utilized to generate boxplots that illustrate the differences in gene expression between wild-type Abf1 and the Abf1 variants (Extended Data Fig. 5f).

## Data processing for occupancy measurement via DNA methylation and high-throughput sequencing (ODM-seq)

Both natural and modified bases were called for Oxford Nanopore sequencing reads, and reads were filtered and demultiplexed with Dorado (version 1.0.0+4a76f8aa; Oxford Nanopore Technologies (ONT) Basecalling Software). The resulting reads were mapped to the *S. cerevisiae* genome (sacCer3 build R64-2-1) with minimap2 (version 2.17), sorted using samtools sort (SAMtools version 1.9) and indexed

with samtools index (SAMtools version 1.15). Methylation information was added to the BAM files using modkit repair (version 0.1.13; ONT).

For computational analysis in R (R versions 4.2.0 and 4.3.1 and tidyverse version 2.0.0), the methylation probabilities were extracted from bam files using modkit pileup (version 0.1.13; ONT). The modkit pileup output corresponds to the averaged methylation probabilities over all reads per genomic coordinate per sample. The genome average methylation was calculated for each sample by averaging the modkit pileup output for 5mC modifications over all coordinates (Extended Data Fig. 10c). As slight batch effects were visible on this level of genome average methylation, we introduced a correction factor. For each Abf1 variant, the genome average methylation was determined by averaging across the four genome average methylation values for each individual replicate (two biological replicates (numbers 1 and 2) with two technical replicates each (180 and 240 min incubation with M.SssI as DNA methylation reached saturation; Extended Data Figs. 6a and 10c)). The resulting sample averages were divided by the analogous average of the four wild-type Abf1 replicates. This quotient was used as a correction factor for each individual replicate.

**+1-aligned composite plots.** The corrected modkit pileup outputs were plotted in composite plots (ggplot2 version 3.4.2) relative to alignment points. Genomic coordinates were converted to new coordinates relative to the alignment points (data.table versions 1.14.8 and 1.17.8; foverlaps function), which were set to zero in each case. Signs of relative coordinates were flipped if the gene corresponding to the alignment point was on the negative strand. Averages of all four replicates per relative coordinate were calculated. Plots were smoothed by plotting the rolling mean over a 50 bp window at the central base pair of each window (data.table; frollmean(., n = 50, align = "center")). The in vivo +1 nucleosome dyad annotations for this were taken from Mahendrawada et al.[74].

**PCA.** PCA was conducted using the corrected modkit pileup outputs, converted to relative coordinates with respect to the +1 nucleosome positions (see the section '+1-aligned plots'). PCA (stats 4.3.1; prcomp(., scale. = F))) compared average occupancies per sample (first averaged over the biological replicates and then over the technical replicates) or per individual replicate in respective regions: downstream flank of +1 nucleosome (0 to 50 bp relative to the in vivo +1 nucleosome dyad); upstream flank of +1 nucleosome (−80 to 0 bp relative to the in vivo +1 nucleosome dyad); and NFR (−145 to −95 bp relative to the in vivo +1 nucleosome dyad). Each +1 nucleosome annotation for each sample or replicate contributed individually to the PCA. In very few cases, the same +1 nucleosome dyad annotation belonged to two different upstream or downstream genes.

**Venn diagrams of binding sites.** First, we compiled a comprehensive list of all sites, annotated either by PWMs or some in vivo mapping technique. PWMs for the prediction of Abf1 binding sites solely from DNA sequence (FIMO (MEME version 5.5.7)) were taken from JASPAR (https://jaspar.elixir.no)[125] and Badis et al.[35]. Previously annotated coordinates according to PWM predictions by others were according to Kubik et al.[69], MacIsaac et al.[126] and Pachkov et al.[127]. The previously annotated in vivo-mapped Abf1 binding sites were either according to Mahendrawada et al.[74] (and kindly shared by L. Mahendrawada and S. Hahn) or published by Zentner et al.[128], Badjatia et al.[129] and Gutin et al.[130]. If binding sites were annotated as regions, each binding site was mapped to the single genomic coordinate at its central position. If binding sites were annotated for replicates, we averaged over the replicates and used the central position of the average.

For each Venn diagram, the compared subset of Abf1 binding sites selected from the comprehensive list was turned into a reference set of sites. For each site, the distance to the closest site with a lower genomic coordinate was measured. If this distance was <20 bp, the site with the

higher genomic coordinate was discarded. From this reference set of sites, pseudo-windows of 1 bp length were constructed.

The Abf1 binding sites selected from the comprehensive list for comparison in each Venn diagram were extended by 75 bp to either side, and overlap with the previously constructed reference set of these sites was determined using foverlaps (data.table 1.17.8). Venn diagrams were constructed from the reference set IDs in each of the displayed categories using ggVennDiagram (of the ggVennDiagram Package) in R[131].

#### Data processing for MNase-seq
Illumina sequencing data were mapped to the *S. cerevisiae* sacCer3 (R64-1-1 build) genome using Bowtie 2 (ref. [132]), excluding reads with multiple matches. The BAM files were converted to BED files and then to RDS files of ranges (start to end), which were imported into RStudio. For comparing samples with different total read numbers, we applied subsampling. To mainly include DNA fragments generated by MNase digestion of canonical nucleosome core particles, we selected only 120- to 170-bp reads. To visualize the MNase-seq patterns, we plotted 50-bp extended dyads. Regions of >200 bp contiguous nulls (no coverage) were omitted among all compared samples for a particular graph. Signs of relative coordinates were flipped if the gene corresponding to the alignment point (the Abf1 site) was on the negative strand.

#### Proteome-wide bioinformatics analysis of disordered regions
Motivated by previous work, the set of aligned syntenic genes from 20 different yeast species was obtained from the Yeast Gene Order Browser, as described previously[5,8,22,25,133]. This dataset contains 5,430 protein sequences. Orthologous sets of sequences were then aligned using Clustal Omega to generate alignments for every *S. cerevisiae* protein[134]. Per-residue sequence conservation was calculated using normalized Jensen–Shannon divergence based on the BLOSUM62 matrix, as was first introduced by Capra and Singh[135–137]. This analysis yields a per-residue conservation score, with values ranging from 0.019–0.948 across the yeast proteome (Fig. 1).

Disorder scores and predicted predicted local distance difference test (pLDDT) scores for analyses were calculated using metapredict (V1), where defined IDRs were identified using the IDR delineation mode[26]. The predicted pLDDT value reflects a prediction of the per-residue pLDDT score—a metric introduced by DeepMind in the context of structural evaluation for AlphaFold2. Metapredict implements a deep learning model to predict the pLDDT score from the sequence alone.

Almost all disorder predictions in this work used metapredict V1, instead of the more accurate metapredict V2 or V3. This was a deliberate decision driven by the underlying training data used for metapredict V2 and V3. Metapredict V1 was trained in a manner that avoids convolving evolutionary information inferred from AlphaFold, whereas V2 and V3 combine AlphaFold-derived information with a set of additional predictions, meaning evolutionary information is (indirectly) encoded, given that AlphaFold's ability to confidently predict structure is in part determined by the availability of large sets of alignable homologous sequences. Using the V1 predictions allows us to legitimately compare disorder and conservation without inadvertently capturing trends encoded in the underlying bidirectional recurrent neural networks. The one exception to this is examining IDRs that are conserved in terms of chemical specificity but not sequence specificity (Fig. 5). For this analysis, we used metapredict V3 first to identify IDRs across the yeast proteome, and then followed up using these IDRs. We made this decision given the accuracy metapredict V3 has for correctly identifying large contiguous IDRs. At the time of writing, metapredict V3 was not published but was publicly available in a branch of metapredict.

pLDDT-defined structured domains are set using a predicted pLDDT threshold of 65 or greater; gaps between contiguous stretches of predicted pLDDT values over 65 that are three residues or smaller are filled in. A minimum region size of five residues is set for convenience.

#### Sequence analysis
All sequence analyses were performed using SHEPHARD (https://shephard.readthedocs.io/). The complete set of sequences, conservation scores, disorder scores, ppLDDT scores and IDR domain boundaries are provided as SHEPHARD-compatible data files (structured, tab-separated data files). IDR sequence analyses (compositional analysis, hydrophobicity, predicted disorder and predicted pLDDT scores) were performed using localCIDER, SPARROW (https://github.com/idptools/sparrow/) and metapredict[26,138–140].

#### Sequence design
IDR designs were generated using GOOSE, a Python package for the rational design of IDRs and IDR variants[141]. Documentation for GOOSE is available at https://goose.readthedocs.io/ and the open-source code is available at https://github.com/idptools/goose.

Sequence designs constrained residue patterning parameters to retain wild type-like residue patterning (Extended Data Fig. 3d–f). Patterning here is quantified in terms of binary patterning using a kappa-like definition for patterning for chemically distinct residue groups, where $\kappa$ is a patterning parameter that measures the degree to which sets of residues intermix[23,138,142]. A high $\kappa$ value means that residues in groups are highly segregated, whereas a low value implies an even intermixing of residues. In the original definition of $\kappa$, three residue groups were considered (positive, negative and neutral), where $\kappa$ quantifies the extent of intermixing between positive and negative residues. $\kappa$ can also be defined as a binary patterning parameter[16,143], where it measures the extent to which residues in one group segregate from all other residues. The other patterning parameters in Extended Data Fig. 3d–f reflect the binary patterning of groups of amino acids noted below each datapoint versus all other residues. For example, GSNHQT reflects the degree of segregation of residues {GSNHQT} from all other residues. While not quantified here, all shuffle and distribution designs ensured that residues were evenly distributed across the context sequence to minimize the possibility of accidentally generating an additional ex nihilo motif.

#### Identification of conserved sub-sequences in the yeast proteome
Conserved sub-sequences were identified via the following procedure. First, hydrophobic (I, L, V, M, Y, F and W) residues in an IDR with a conservation score above 0.65 or higher were identified. These residues are in the 85th (or higher) percentile for conservation across all disordered regions (dashed lines; Extended Data Fig. 1b–d). For each conserved hydrophobic residue (initiator methionine residues are excluded) in an IDR, we identified the residues ±6 of that central conserved IDR (truncating at the N and C termini as necessary). We then calculated the median disorder for this subregion of (up to) 13 amino acids. Finally, having done this for an entire protein, we extracted contiguous subregions where the median disorder score was above some threshold. A lower disorder threshold finds more regions but runs the risk of identifying residues at the boundaries between disordered and folded domains, which may, in fact, be folded. A higher disorder threshold identifies fewer regions but provides higher confidence that the identified regions are bona fide IDR. With the most permissive disorder threshold, we identified 5,173 conserved subregions. With the least permissive disorder threshold, we identified 884 subregions. All sub-sequences, including their position and parent protein, are provided in Supplementary Tables 7–11.

#### Identification of sub-sequences with predicted transient structure
Using a deep learning model trained on ~360,000 protein structures from AlphaFold2, we used the metapredict package to predict per-residue predicted structure across the yeast proteome[26]. Cross-referencing those scores against predicted disordered regions

and excluding sub-sequences directly adjacent to folded domains, we finally filtered for contiguous sub-sequences with a predicted pLDDT score above 40, where the sub-sequence was up to 40 residues long. The complete set of these sub-sequences, average disorder score and average conservation are provided in Supplementary Table 17.

## Identification of IDRs with compositional conservation

Compositional conservation was calculated by taking a set of orthologous aligned IDRs and computing the mean Euclidean distance between fractional amino acid composition across all IDRs. Compositional conservation was only calculated for IDRs ten residues or longer in length (without alignment gaps) and for sequences where ten or more orthologous IDRs were identified. The complete set of IDRs, including median and mean sequence conservation and compositional conservation, is provided in Supplementary Table 16.

## Hydrophobicity and charge scores

Hydrophobicity and charge scores (Fig. 4i) were calculated as compositional weighted metrics.

The hydrophobicity score ($B$) for a sequence was calculated using the following expression;

$$B = \sum_{i}^{\beta} f_i \beta_i \tag{1}$$

Where, for a given protein, $B$ is the binding score, $\beta$ is the set of residues used to calculate the binding score (Y, W, F, M, V, I, L, A and P), $\beta_i$ is the binding score coefficient associated with residue identity $i$ and $f_i$ is the fraction of the sequence made up of residue $i$.

The charge scores ($C$) were calculated analogously:

$$C = \sum_{i}^{\theta} f_i \theta_i \tag{2}$$

Where the residues in set $\theta$ are R, K, E and D and $\theta_i$ is the charge score coefficient. These coefficients are listed in Supplementary Table 16. This definition is an ad hoc analysis used specifically in the context of Abf1 IDR2, and we do not necessarily expect these precise parameters to be transferable over to other systems. That said, the spirit of this analysis may well transfer to other types of IDR-mediated interactions.

## AlphaFold2 model

The AlphaFold2 model was taken directly from the European Bioinformatics Institute AlphaFold2 protein structure prediction database (https://alphafold.ebi.ac.uk/entry/P14164)[144,145].

## Gene Ontology analysis

The Gene Ontology analysis described in Extended Data Fig. 1g, and for proteins with IDRs that are conserved in terms of chemical specificity but not sequence, was performed using PANTHER[146].

## Structure predictions

Helical structural predictions were generated using AlphaFold2 (Fig. 3)[145].

## Coarse-grained simulations

Coarse-grained simulations were performed using the PIMMS simulation engine (https://github.com/holehouse-lab/pimms) to generate the 2D binding landscapes. A 50-bead flexible polymer was defined in which the central six residues were designated as a motif, whereas the flanking residues on either side were defined as the context. A second 50-bead polymer that collapsed due to attractive intramolecular interactions defined a binding partner. Motif:globule (M) and context:globule (C) interactions were systematically titrated to generate a 2D sweep of parameter space (Extended Data Fig. 4a–d). The fraction bound

was calculated for each simulation run at a unique combination of M and C interaction strengths. Binding here was defined as the two molecules in direct interaction and did not distinguish between motif- and context-based binding.

Monte Carlo simulations were run on a cubic lattice of 30 × 30 × 30 sites. The system was evolved via a combination of chain repetition and rigid-body translation moves. Each simulation was run for a total of approximately 30 × 106 individual Monte Carlo moves.

## In silico evolution

In silico evolution (Fig. 6) was performed by iteratively acquiring a set of mutations and then evaluating the fitness of the sequence with respect to some selection function. In our case, mutations were introduced as single-point mutations only (no insertions or deletions) and fitness was evaluated in terms of attractive interactions with a specific binding partner.

Chemical specificity was calculated using FINCHES with the Mpipi-GG forcefield[64,102,147]. In silico evolution used a fitness function whereby the summed attractive interactions across the predicted intermolecular interaction map were computed and variants survived if the summed attractive interactions remained approximately equal to or more attractive than the wild-type-associated value. Mutations were introduced by sequentially making a series of mutations before evaluating fitness, enabling compensatory mutations to appear.

Mutations were introduced through an iterative mutational strategy. First, the amino acid sequence was converted into a nucleotide sequence. Next, the underlying nucleotide sequence was stochastically mutated according to the expected transition or transversion rate for nucleobases in coding sequences. This was done by randomly selecting a nucleotide and having it undergo a transition or transversion event. We typically performed 30–40 events per nucleotide sequence to mutagenize the underlying nucleic acid sequence to a consistent number of amino acid changes. Having mutagenized the underlying nucleic acid sequence, this sequence was converted back into protein space. If a premature stop codon was introduced, the sequence was discarded; otherwise, that sequence was saved as one of a set of possible variants. This procedure was repeated many times to generate an initial library of variants. Finally, each variant was assessed with respect to the underlying fitness function and only those variants that were sufficiently fit were kept. For null models (evolution-absent selection), the same procedure was carried out, except we did not apply a fitness selection filter at the end. The acquisition of all mutations before evaluation by fitness function facilitated the gain of compensatory mutations. It also more accurately mirrors the logistical steps taken for the in vitro error-prone PCR experiment.

The described protocol for in silico evolution was used to create two libraries of 350 sequences; in one library (the real library), every sequence was evaluated and kept based on the fitness function, whereas in the other (the null library), sequences were randomly mutagenized without constraints on which sequences were kept or discarded. We ensured that the numbers of mutations observed in sequences in the real library and null library were always matched and were typically around 28–31 mutations per sequence (that is, affecting 12–15% of the residues). However, it is worth noting that the results did not change if we increased or decreased the number of mutations.

To perform statistical comparisons, we performed a bootstrap-like analysis to investigate per-residue conservation for sequences generated for our two libraries (Fig. 5c). This involved randomly sampling 20 sequences (without replacement) from either the real library or the null library, and for those 20 sequences, calculating the per-residue conservation score. This was then repeated ten times to build a distribution of per-residue conservation scores expected for sequences generated under selection versus sequences generated from the null model. For each residue position, the mean and standard deviation for those distributions are shown in Fig. 5c.

We initially compared the conservation of IDR2 versus the N-terminal IDR of Rad7 as our hypothetical target protein. To establish that the apparent lack of conservation observed with alignment-based methods is a general phenomenon rather than a fluke, we repeated this analysis using eight additional IDRs from proteins with which Abf1 has been proposed to interact. Putative interactors were taken as physical interactors from BioGRID[50]. However, we note that in many cases the interactions were identified in the presence of DNA, such that we cannot distinguish between a scenario in which Abf1 interacts directly with the putative partner and one in which both proteins interact with the same DNA molecule. Rad16 and Rad7 are notable exceptions where protein-only complexes have been identified[47,65]. Regardless, in silico evolution for the eight additional potential partners leads us to the same conclusion (Extended Data Fig. 4h).

For comparing in silico evolution versus the results from error-prone PCR experiments (Fig. 5e), a similar strategy was taken to that described above, except we matched the number of mutations to the average number of mutations seen experimentally (~28) and subsampled groups of 14 sequences to mirror the number of viable sequences identified experimentally.

As a final note, our in silico evolution was far from a faithful reproduction of the putative dynamics that shape mutations in protein-coding sequences. We consider this a toy system enabling us to ask a stylized question: if the only determinants of protein sequence changes were constrained in this way, could conservation be detected through multiple sequence alignments? In reality, partners are co-evolving and post-transcriptional restraints (that is, RNA secondary structure, processing and degradation) will also shape protein-coding sequences, and insertions and deletions will contribute additional noise. We consider our simplified model to be an overly stringent and deliberately simple model for selection—if signatures of conservation are not easily detectable here, the ability to do so in even noisier scenarios seems even less likely.

### Conservation of chemical specificity

To assess IDRs in which chemical specificity was conserved, we first identified those IDRs with a low degree of average sequence conservation. We used an average sequence conservation value of 0.35. To assess chemical specificity, we computed average IDR interactions with a set of 36 previously identified chemically orthogonal dipeptides[64]. For each set of orthologous IDRs, we calculated the variance in mean-field interaction scores with each of those 36 peptides. We then identified those IDRs where, despite very low sequence conservation, the variance among one or more dipeptide interactions was in the bottom 15%. The set of proteins, IDR sequences and dipeptides they were conserved with respect to are included in Supplementary Table 9.

This analysis focused explicitly on large IDRs and is a relatively unsophisticated way to systematically identify signatures of conservation with respect to chemical specificity. This is an area of active and ongoing investigation.

### Flory–Huggins phase diagrams

To generate hypothetical Flory–Huggins phase diagrams, we solved the Flory–Huggins free energy of mixing using the FIREBALL package[148]. This allowed us to calculate volume fraction ($\phi$) and temperature phase diagrams for different chain lengths (Extended Data Fig. 9c). The reduced temperature was calculated as the binodal temperature divided by the critical temperature ($T_c$). To convert volume fraction into molar concentration, we operated under the assumption that

$$\varphi = c \left( \frac{\upsilon_m N_a}{M_m} \right) \quad (3)$$

Where $\phi$ is the volume fraction, $c$ is the mass concentration (in mg ml$^{-1}$), $\upsilon_m$ is the average volume occupied by a single amino acid (140 Å$^2$), $N_a$

is Avogadro's number and $M_m$ is the average mass of an amino acid (110 g mol$^{-1}$). Given that three of the four terms here are known, this expression can be rewritten as

$$c = \varphi \rho_0 \quad (4)$$

Where $\rho_0$ is 1,310.16 mg ml$^{-1}$. Converting volume fraction to mass concentration (in units of mg ml$^{-1}$) involves multiplying the volume fraction by 1,310.16.

Converting from mass concentration to molar concentration involves dividing the mass concentration by the molar mass of the molecule. For the short and long synthetic repetitive FUS sequences, the molar masses were 7,208.9 and 17,422.1 g mol$^{-1}$, respectively. The precise sequences are listed in Supplementary Table 1.

### Data collection

Throughout the methods, data collection and analysis were not performed blind to the conditions of the experiments.

### Reporting summary

Further information on research design is available in the Nature Portfolio Reporting Summary linked to this article.

### Data availability

Sequencing data have been deposited in the NCBI Gene Expression Omnibus (https://www.ncbi.nlm.nih.gov/geo/) under accession numbers GSE314049, GSE314050 and GSE314051, as well as in the Sequence Read Archive (https://www.ncbi.nlm.nih.gov/sra) database under BioProject number PRJNA1377534. All data associated with our figures are provided at https://doi.org/10.5281/zenodo.17770967 (ref. 149). They are also available via our GitHub repository at https://github.com/holehouse-lab/supportingdata/tree/master/2026/Langstein-Skora_2026.

### Code availability

The code used to perform the analyses and generate the figures is provided at https://github.com/holehouse-lab/supportingdata/tree/master/2026/Langstein-Skora_2026. The complete code is provided at https://doi.org/10.5281/zenodo.17770967 (ref. 149). Code for the ODM-seq is deposited at https://github.com/gerland-group/LS_ODM_seq_analysis.

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

## Acknowledgements

We thank R. V. Pappu for his immediate willingness to collaborate once approached by P.K., and for allowing A.S.H. to work independently on this project while completing his postdoctoral work. We thank R. Das for help with jump-starting this project and B. Kornmann for helping M.O.R. to work with data generated by the Kornmann lab at an early stage of the project. We thank W. de Jonge and F. Holstege for sharing their Abf1 anchor-away strain and Abf1 site annotations, as well as for numerous discussions and advice. We are grateful to T. Straub for advice on bioinformatics analyses; S. Kubik and D. Shore for sharing class I–V responder Abf1 site annotations; and S. Hahn and L. Mahendrawada for sharing the annotation of their Abf1 chromatin endogenous cleavage sequencing (ChEC-seq) binding sites. We thank S. Sukenik, D. Moses, B. Schmidt and S. Plassmeyer for critical comments and feedback on the manuscript. We thank A. Keating and the members of the Keating laboratory for helpful discussion. We are grateful to A. Imhof, C. Kurat and T. Schauer for providing input as thesis advisory committee members for I.L.-S., and especially A. Imhof for his suggestion to use the ChIP assay as a control. We thank S. Krebs, A. Graf and H. Blum (Laboratory for Functional Genome Analysis, Gene Center, LMU Munich) for Illumina and Oxford Nanopore sequencing and data handling; and M. Hataichanok (The Histone Source, Colorado State University) for purified recombinant histones. We acknowledge funding from the German Research Foundation (grants KO 2945/3-1 (to P.K.), HO 2489/10-1 (to K.-P.H.), BA 6383/2-1 (to T.B.) and project grants within SFB 1064 (to P.K., K.-P.H. and T.B.)) and European Research Council (ERC Advanced Grant 833613 to K.-P.H.). T.B. acknowledges funding from the Helmholtz Association. F.H. is grateful for the support provided by the TUM Innovation Network RISE, funded through the Excellence Strategy. A.S.H. acknowledges funding from the US National Science Foundation via the Molecular Sciences Software Institute (NSF-1547580; subaward 479590); a CAREER award (number 2338129); and the Water and Life Interface Institute (number 2213983). A.S.H. acknowledges funding from the Longer Life Foundation (Reinsurance Group of America–Washington University Collaboration) and the Human Frontier Science Program (grant RGP0015/2022). N.P. and V.V. acknowledge funding as Amgen scholars in the laboratory of P.K. The funders had no role in study design, data collection and analysis, decision to publish or preparation of the manuscript.

## Author contributions

A.S.H. and P.K. conceived of the idea. A.S.H., P.K., I.L.-S., A.S., F.H., D.S., R.J.E., M.O.R., L.S., T.B., F.K., M.L., F.J.M. and N.P. developed the methodology. I.L.-S., A.S., F.H., D.S., R.J.E., M.O.R., P.K., A.S.H., M.J.G., S.K.P., S.K.R., V.V. and N.P. performed the investigation. A.S.H., I.L.-S., F.H., D.S., W.A. and P.K. visualized the data. P.K., U.G., A.S.H., T.B. and K.-P.H. acquired funding. P.K., A.S.H., U.G. and K.-P.H. performed project administration. P.K., A.S.H., U.G., K.-P.H. and T.B. supervised the project. A.S.H., P.K. and I.L.-S. wrote the original draft of the

manuscript. A.S.H., P.K., F.H., R.J.E. and I.L.-S. reviewed and edited the manuscript.

## Funding

## Competing interests

A.S.H. is on the Scientific Advisory Board for Prose Foods. The remaining authors declare no competing interests.

## Additional information

**Extended data** The data are available at https://doi.org/10.1038/s41556-025-01867-8.

**Correspondence and requests for materials** should be addressed to Philipp Korber or Alex S. Holehouse.

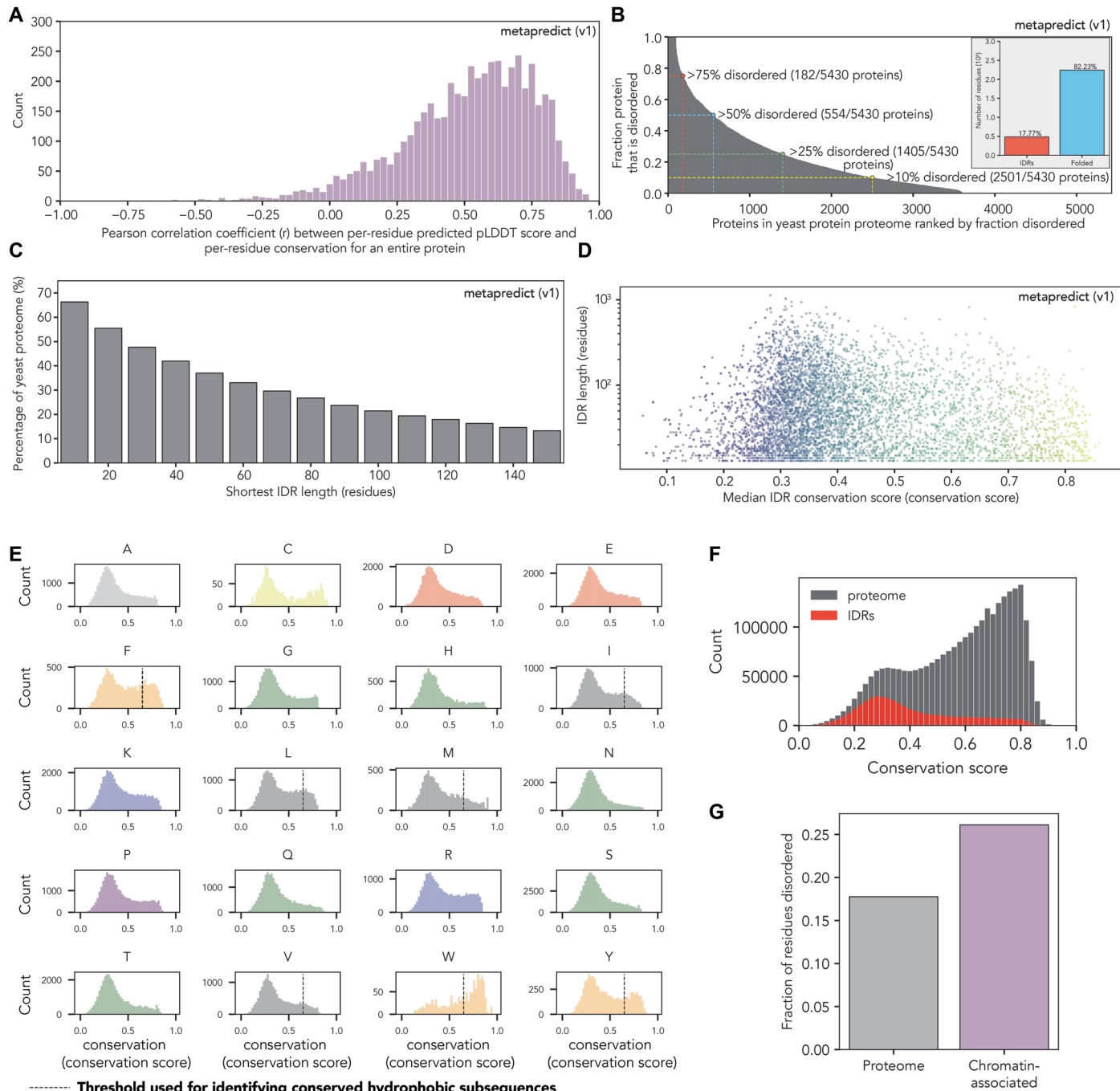

**Extended Data Fig. 1 | See next page for caption.**

**Extended Data Fig. 1 | Informatics analysis across the yeast proteome. a**, Distribution of correlation between per-residue predicted structure and per-residue conservation across the *S. cerevisiae* proteome. Per-residue predicted structure was calculated using the predicted pLDDT, a metric that quantifies the likelihood that AlphaFold2 would correctly predict a structure for the residue in a specific sequence context. This predicted pLDDT was generated using metapredict and correlated (r) with per-residue conservation. **b**, Rank-ordering every yeast protein by fraction disordered illustrates how prevalent disordered regions are predicted to be *in S. cerevisiae*, where almost half of all proteins are 10% disordered or more. In comparison, about 18% of all residues reside in IDRs (inset). We note that these numbers reflect disorder predicted by metapredict V1, which underestimates disorder relative to the more accurate V2 and V3 implementations. **c**, To further convert from fractional disorder into the number of residues, we show the fraction of proteins in the yeast proteome that contain one (or more) IDRs of various lengths. Over 30% of yeast proteins contain one or more IDRs of sixty residues or longer. **d**, Conservation vs. IDR length reveals that, while many IDRs are poorly conserved, a subpopulation of IDRs with a wide range of lengths is relatively highly conserved. Colours track with median IDR conservation (that is, x-axis values). Gene ontology (GO) analysis reveals that the top 200 most conserved IDR-containing proteins are enriched for nucleosomal proteins (for example, histones), ribosomal proteins, and DNA repair proteins. **e**, Histograms of per-residue conservation for each amino acid type within IDRs. We calculated the per-residue conservation for each instance of each amino acid, separated by amino acid type, and histogrammed the values. For hydrophobic and aromatic residues, there tend to be two populations: a large population of poorly conserved residues (conservation scores of 0.3 to 0.4) and a smaller population of highly conserved residues. This is most noticeable for tryptophan (W), tyrosine (Y) and phenylalanine (F). Given that SLiMs are often driven by hydrophobic residues, we defined a threshold conservation score of 0.65 and defined any of the three aromatic (Y/W/F) and four bulky aliphatic (L/I/V/M) residues with a score above this as 'highly conserved' (dashed lines). Based on this delineation, we extracted sub-sequences centred on highly conserved hydrophobic residues as possible signatures for binding motifs (Supplementary Tables 10-14). **f**, Histograms that show the distribution of per-residue conservation scores across the entire proteome and for all IDRs. The per-residue conservation scores for every residue across the whole yeast proteome (black) or within disordered regions (red) were histogrammed. Disordered regions show a monomodal distribution with a peak around 0.3, whereas the whole proteome shows a bimodal distribution, with one peak in line with the disordered regions at around 0.3 and a second peak with substantial skew at around 0.75. **g**, Comparison of disordered residues in chromatin-associated proteins versus all proteins. The fraction of disordered residues in the entire yeast proteome (n = 5430 proteins, left) is compared with chromatin-associated proteins (n = 190, right). Chromatin association is based on annotation with the GO term "chromatin" (GOA = 785). Disordered regions are defined based on metapredict (V1) with standard thresholds.

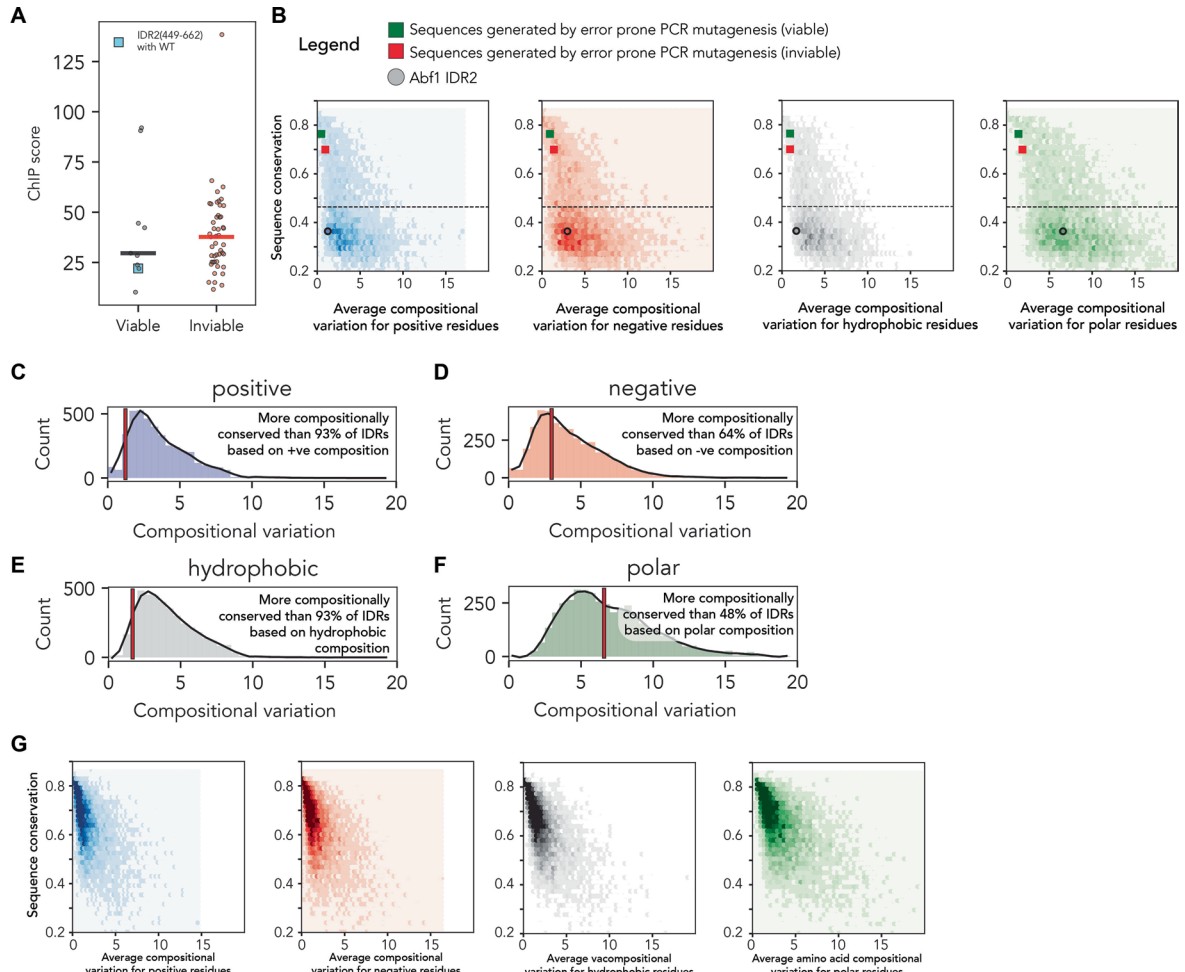

**Extended Data Fig. 2 | Proteome-wide conservation of composition vs. sequence. a**, ChIP scores for viable vs. inviable constructs confirm the ability of variants to bind Abf1-specific genomic loci even in the event of inviability. The distribution of ChIP scores for selected viable and all inviable rationally designed constructs (Supplementary Table 1) is plotted. Horizontal bars mark the median value. Boxed values of viable constructs were measured in the presence of untagged wild-type Abf1 (Methods). These two populations do not differ significantly (Mann–Whitney *U*-test; *P* value = 0.24). **b**, Panels showing median sequence conservation as assessed by alignment (*y* axis for all) compared with the per-residue compositional conservation (*x* axis) for four different compositional groups: negative (E,D), positive (R,K), hydrophobic (I,L,V,M,Y,F,W) and polar (Q,S,N,H,G) residues. In each panel, we also show the compositional conservation and sequence conservation scores for the set of IDRs generated via random mutagenesis (Methods and Fig. 5f), which were either viable (green square) or inviable (red square). In addition, the position of IDR2 is noted (circles). These points are included only to be compared with one another and

are not easily comparable with "natural" proteins. **c–f**, All IDRs that fall below the dashed line in **b** are histogrammed on compositional conservation, with IDR2 from Abf1 shown as a red vertical line on the histogram. **c**, Positive amino acid composition in IDR2 is more conserved than 93% of other IDRs, but the number of positively charged residues is low, indicating that IDR2 is less likely to acquire positive residues. **d**, Negative amino acid composition in IDR2 is more conserved than 64% of other IDRs. **e**, Hydrophobic amino acid composition in IDR2 is more conserved than 93% of other IDRs. **f**, Polar amino acid composition in IDR2 is more conserved than 48% of other IDRs. From this, we naively conclude that charge and hydrophobicity are more constrained in IDR2 than polar amino acid content. **g**, Sequence conservation vs. compositional conservation compared with all other similarly conserved folded domains in the yeast proteome. Folded domains were identified using predicted pLDDT-based sequence analysis, analogous to the identification of disordered regions. Compositional and linear conservation analyses confirm that folded domains are substantially more conserved in both respects.

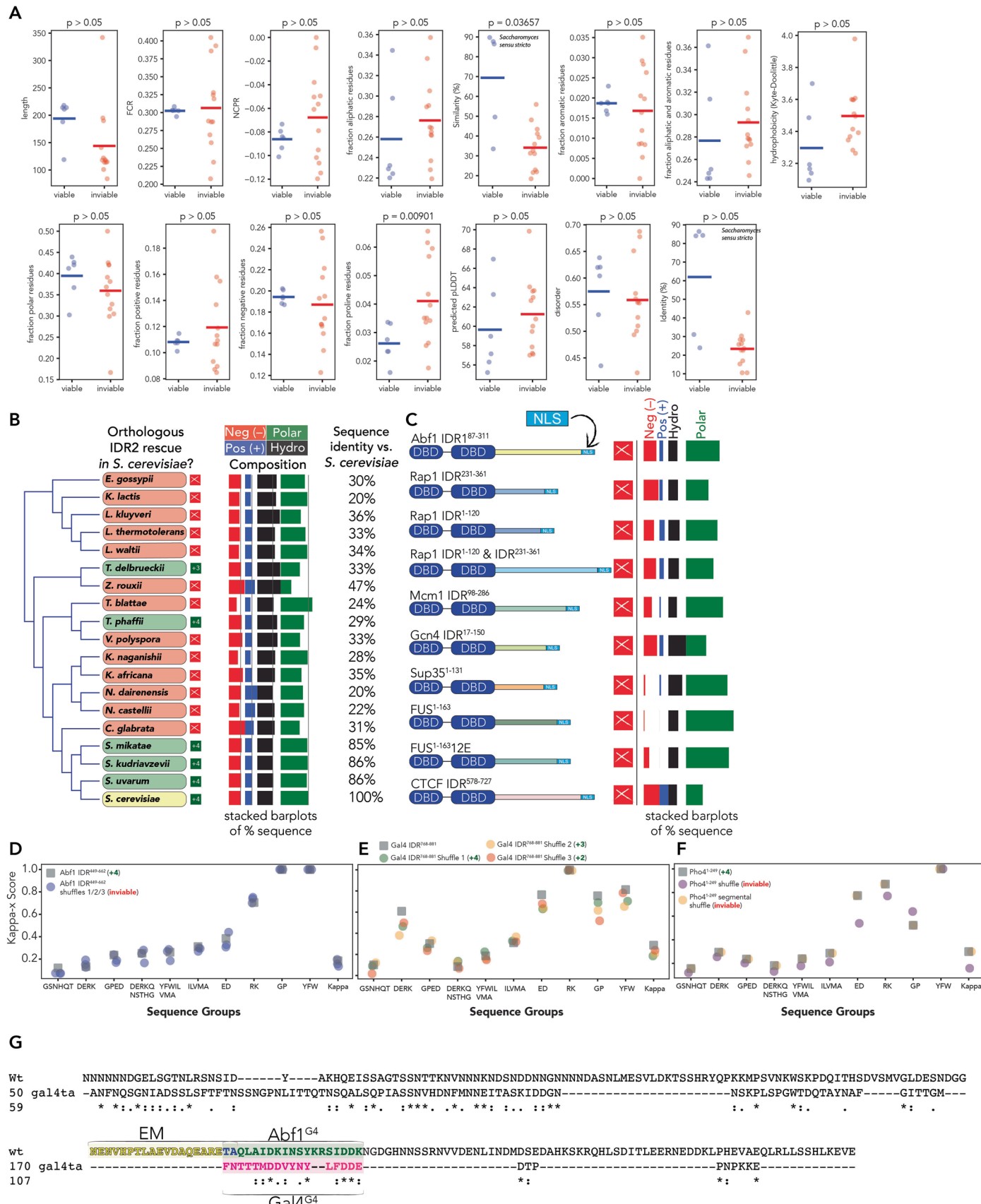

**Extended Data Fig. 3 | See next page for caption.**

**Extended Data Fig. 3 | Analysis of orthologous IDRs, designed variants, and Abf1:Gal4 alignment. a**, Comparison of various sequence features for viable vs. inviable IDR2 orthologues. Across the 13 sequence analyses tested (excluding similarity and identity), only one (fraction of proline) has a Welch's *t*-test *P* value below 0.05 when testing for a statistical difference in viable vs. inviable sequences. In addition to sequence features, we computed sequence similarity and identity to wild type. Including species from *Saccharomyces sensu stricto*, an evolutionarily close species complex that includes *Saccharomyces cerevisiae, Saccharomyces uvarum, Saccharomyces mikatae* and *Saccharomyces kudriavzevii*, sequence identity is not statistically different between viable and inviable orthologues. Although sequence similarity has a *P* value below 0.05 when the *Saccharomyces sensu stricto* species are included, the viable and inviable sequences are indistinguishable in similarity without this species complex. Sequence properties were calculated using localCIDER, SPARROW or metapredict (Methods). NCPR, net charge per residue; FCR, fraction of charged residues. **b**, Evolutionary tree (as in Fig. 2d) showing amino acid composition for each construct as a stacked bar chart. The grey lines reflect the composition of *S. cerevisiae* IDR2. **c**, Comparison of tested IDRs from other proteins (as in Fig. 2e) with the amino acid composition for each construct, shown as a stacked bar chart. **d**, Abf1 IDR2 shuffles (first shown in Fig. 3b) were

designed to largely preserve residue patterning for different residue groups, with small perturbations above and below the wild-type sequence. Regardless of patterning, all shuffles were inviable. **e**, For Gal4$^{768-881}$ shuffle 1/2/3, Gal4$^{768-881}$ was globally shuffled to generate a range of sequence patterns across different residue types. In this case, all sequences were viable, albeit with varying growth scores. Although {GP} patterning followed an approximate trend in which further from wild type showed an increasing growth defect, this trend is not preserved in any of the other designs, and Gal4$^{768-881}$ is relatively glycine and proline deficient (total fraction just 11%). In contrast, for most other shuffles, there is no correlation between residue patterning and growth score. **f**, Shuffles for Pho4$^{1-249}$ were explicitly designed to test residue patterning. We compared a global shuffle with a segmental shuffle, in which local, chemically self-similar subregions were shuffled. Both shuffles are inviable, but patterning analysis for the segmental shuffle matches 1:1 with wild type. **g**, Alignment of Abf1 IDR2 (wild type) and Gal4$^{768-881}$ (Gal4TA). The EM, Abf1$^{G4}$ subregion and Gal4$^{G4}$ subregion are highlighted. Although other regions in the alignment are also reasonable (for example, the N termini of both proteins), our goal here was to determine whether any regions in Gal4 explicitly align with the EM. Although we did not find any such region, we reasoned that testing adjacent to the EM would be most meaningful. The alignment was performed using EMBOSS Needle.

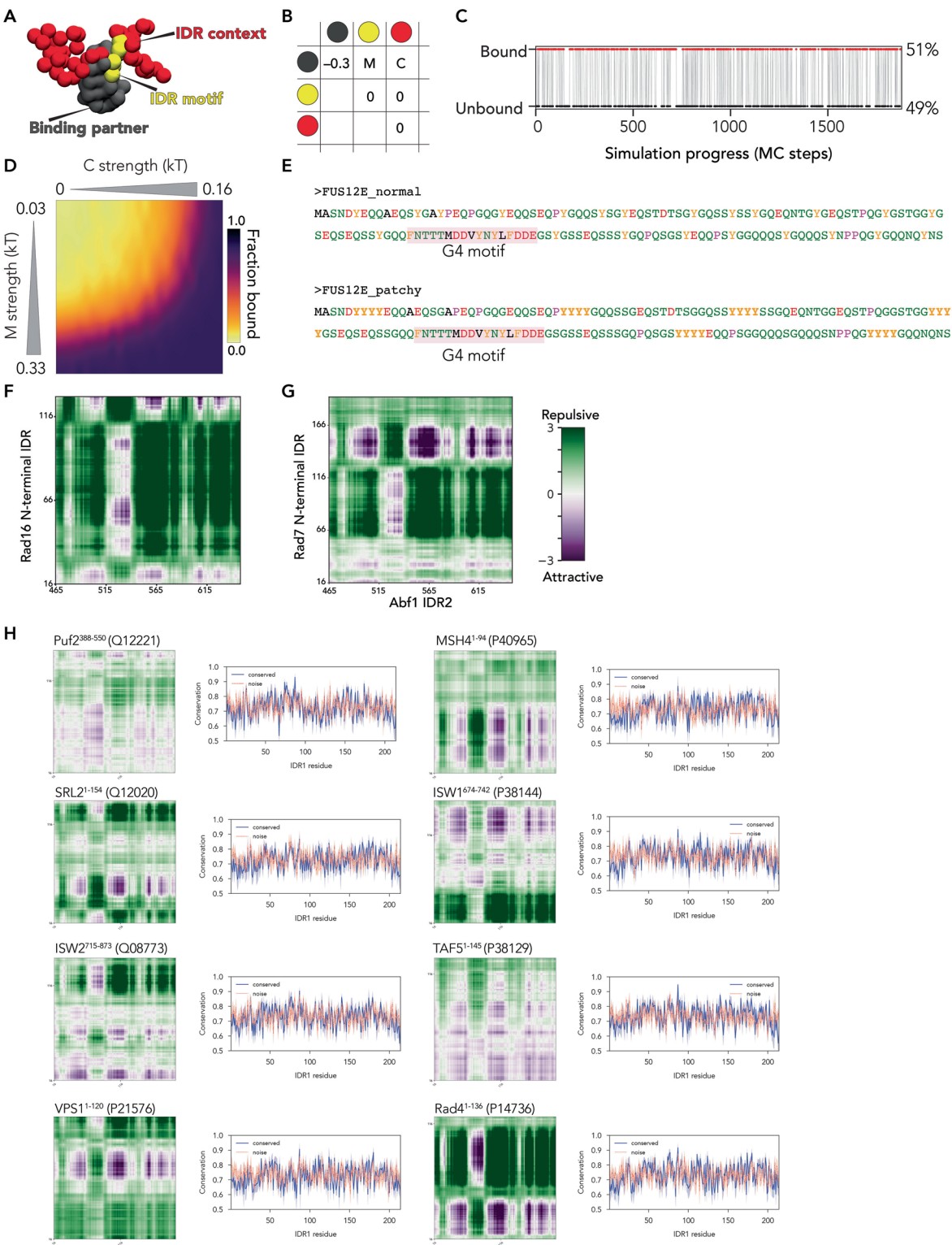

**Extended Data Fig. 4 | See next page for caption.**

**Extended Data Fig. 4 | Chemical specificity through simulations and FINCHES-based analysis. a**, Coarse-grained simulations were performed using a simplified representation, in which an IDR comprises a central binding motif (yellow) with longer N- and C-terminal flanking residues (red). This species binds to a partner represented as a globular polypeptide, shown in grey. **b**, Interaction strength for inter-bead interactions. 'M' defines an interaction strength between the IDR motif and globular binding partner. 'C' defines an interaction between the IDR context and the globular binding partner. Globular intra-bead interactions are strong, whereas motif:globular protein ('M') and context:globular protein ('C') binding strengths are systematically varied. **c**, Example of a single simulation trajectory, revealing an almost 50:50 split of bound and unbound. Simulation progress here is measured in terms of Monte Carlo steps. **d**, Reproduction of Fig. 4b,d. The 2D landscape is generated by systematically varying both C and M parameters, as defined above. **e**, Aromatic residues clustered in the FUS12E patchy sequence. The top sequence shows the normal FUS12E sequence (FUS[1-163]12E) with the embedded G4 motif, whereas the bottom sequence shows the patchy variant. Compositionally, these two sequences are identical, but the patterning of aromatic residues is very different. Specifically, using the inverse-weighted distance (IWD) to compute residue clustering, the aromatic clustering of the wild-type sequence is 0.82, whereas the aromatic IWD clustering in the patchy sequence is 2.49. **f**, Predicted intermolecular interaction map (InterMap) generated using FINCHES to predict specific subregions in Rad16 (UniProt P31244; 1–143) that will interact with IDR2. **g**, Predicted InterMap generated using FINCHES to predict specific subregions in Rad7 (UniProt P06779; 1–207) that will interact with IDR2. **h**, Conservation of chemical specificity yields alignment-based conservation scores that are indistinguishable from random noise regardless of the interaction partner. As described in Fig. 5b, in silico evolution of Abf1 IDR generates conserved interaction profiles with other IDRs taken from potential interaction partners.

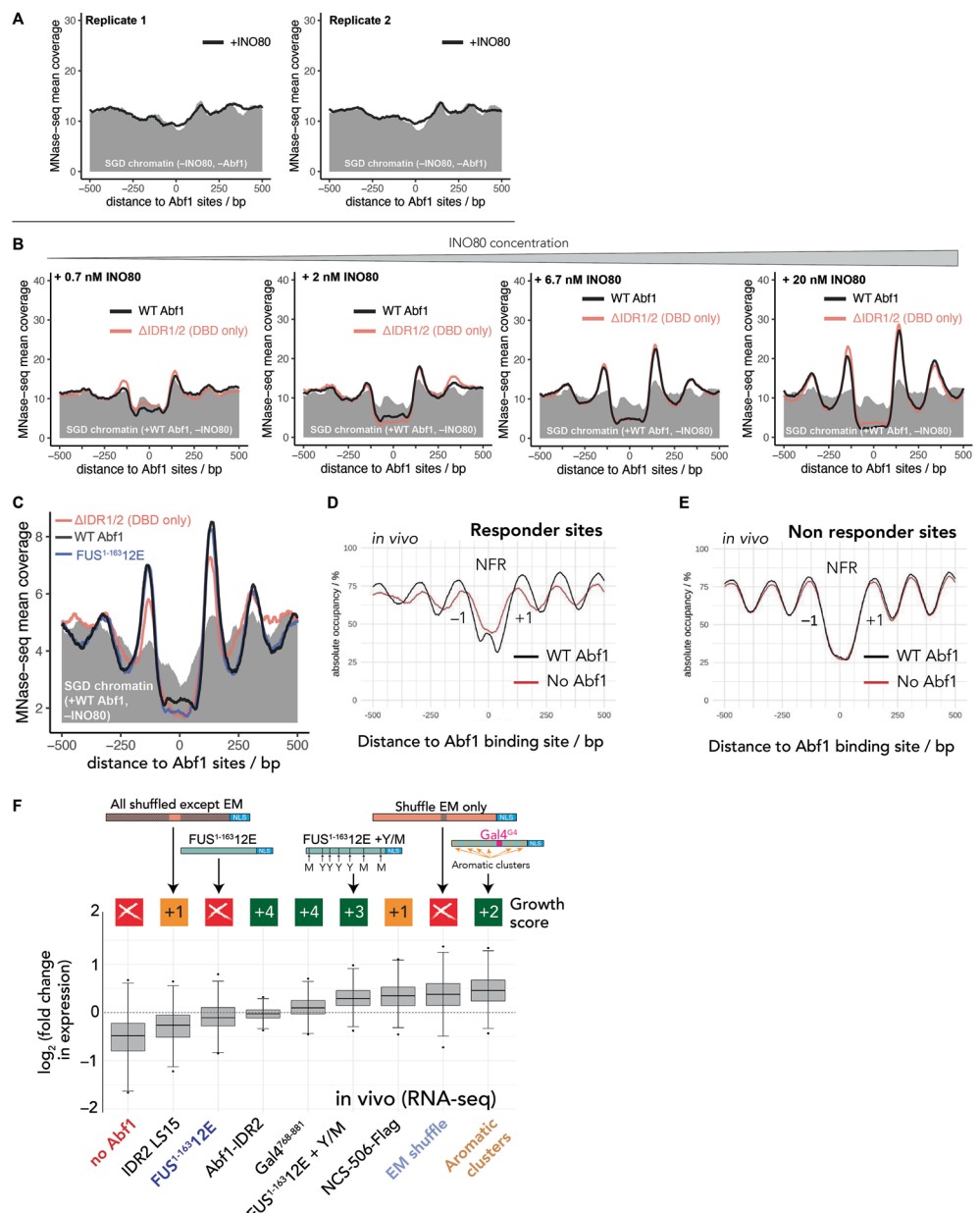

**Extended Data Fig. 5 | MNase-seq, ODM-seq and RNA-seq analysis. a**,
MNase-seq data for whole yeast genome reconstitution show that INO80
could not generate nucleosomal arrays phased to Abf1 sites without Abf1. Two
independent replicates are shown, as in Fig. 6b, but with the indicated conditions.
**b**, The INO80 remodeller concentration was titrated to assess whether Abf1 IDRs
contribute to INO80 recruitment. If this were the case, differences in ± IDR at
low INO80 concentration would be expected, whereas those differences should
vanish in excess INO80. Our results here do not show this behaviour and suggest
that the IDRs of Abf1 are not required for remodeller recruitment. This panel is as
in Fig. 6b, but with the indicated concentrations of INO80. The fact that wild-type
Abf1 vs. ΔIDR1/IDR2 overlapped almost perfectly across different INO80
concentrations indicated that we were not in a regime in the experiment shown in
Fig. 6b where INO80 concentration was so high that IDR-dependent recruitment
to Abf1 was masked. *n* = 1 for each condition **c**, Replicate in vitro reconstitution
experiment as in Fig. 6b, but with some altered conditions (for example,
*Drosophila* histones), as detailed in the Methods. **d**, When Abf1 was depleted
from the nucleus, the NFR around Abf1 sites filled up with nucleosomes, whereas
the flanking nucleosomes (−1, +1) and up- and downstream nucleosome arrays
became disordered. These results are reported as decreased peak-to-trough
ratios and shifts of occupancy peaks towards the NFR. This panel is as in Fig. 6e;
that is, the data are aligned at responder Abf1 sites (classes I and II; *n* = 397)
as identified by Kubik et al. based on MNase-seq data in an analogous Abf1

anchor-away strain and oriented such that a gene is to the right for unidirectional
promoters[69]. **e**, As in **d**, but for non-responder sites (classes III to V; *n* = 478), as
defined by Kubik et al.[69]. **f**, Different Abf1 constructs enhance or suppress RNA
levels for genes (*n* = 279) linked to Abf1 responder sites (**d** and **e**; Methods).
Shown are the changes in total RNA levels in Abf1 anchor-away strains with the
indicated Abf1 constructs left in the nucleus relative to a strain with wild-type
Abf1 left in the nucleus. *n* = 3 for FUS[1–163]12E + Y/M, no Abf1; *n* = 2 for all other
constructs. NCS506 is a random mutagenesis construct (Fig. 5d) with just 13
point mutations. Boxes span the middle two quartiles; whiskers extend to 1.5× the
interquartile range; and horizontal bars denote median log₂[fold change] values
of the Abf1 variant relative to wild-type Abf1. The naive expectation that the
degree of downregulation in RNA-seq data for these genes with Abf1 responder
promoter sites would scale with the degree of viability in Abf1 IDR variants was
not borne out. Instead, there were reproducible but idiosyncratic transcription
responses. In hindsight, such varied outcomes may even be expected given
that total RNA-seq after 75 min of rapamycin treatment should include indirect
effects. In addition, the range of tested IDRs may mediate more or less specific
interactions with various partner proteins, exerting diverse effects depending on
the Abf1 sites, that is, where in the genome these IDRs are recruited via the Abf1
DBD. Different IDRs need not necessarily lead to a loss of function in terms of an
inability to recruit native Abf1 partners, but also a gain of function in recruiting
other nuclear components.

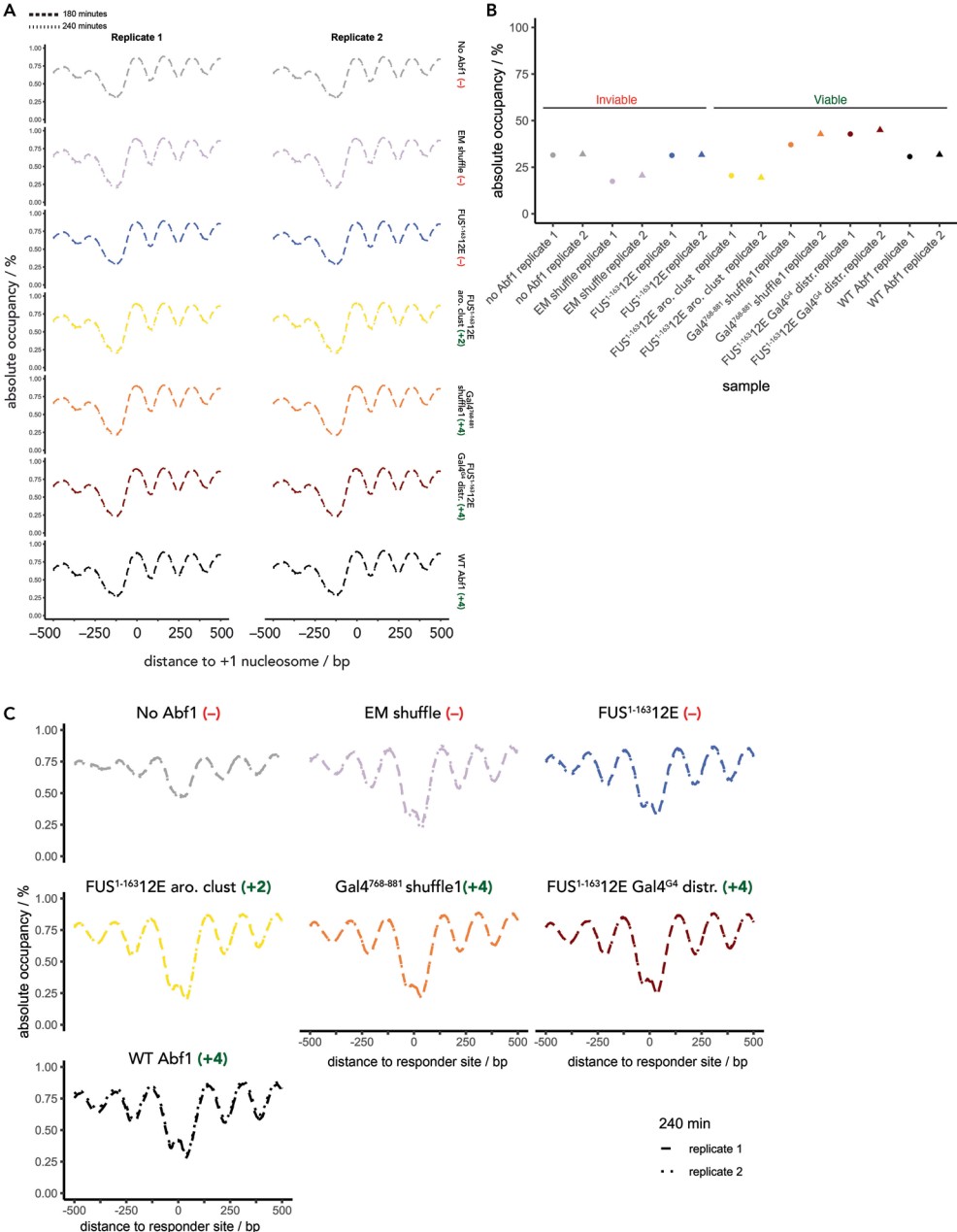

**Extended Data Fig. 6 | ODM-seq analysis. a**, ODM-seq reaches saturated DNA methylation. Incubation of each yeast chromatin preparation with CpG-specific DNA methyltransferase M.SssI for 180 vs. 240 min resulted in the same occupancy patterns of composite plots aligned at all in vivo +1 nucleosomes. Importantly, active methyltransferase activity during the 60 additional minutes of incubation for the 240 min samples was demonstrated by de novo methylation of plasmid pFMP233 spiked in at 180 min (**b**). **b**, Spike-in control for sustained

DNA methylation after 180 min. Average occupancy levels (occupancy = 1 − methylation; that is, low occupancy reflects high DNA methylation levels) are shown for plasmid pFMP233, which was spiked into the indicated samples at 180 min. **c**, Independent biological replicates demonstrate near-perfect reproducibility in ODM-seq occupancy patterns. ODM-seq data for the indicated samples after 240 min incubation with M.SssI are aligned at the responder Abf1 sites, as defined by Kubik et al. (see also Extended Data Fig. 5d)[69].

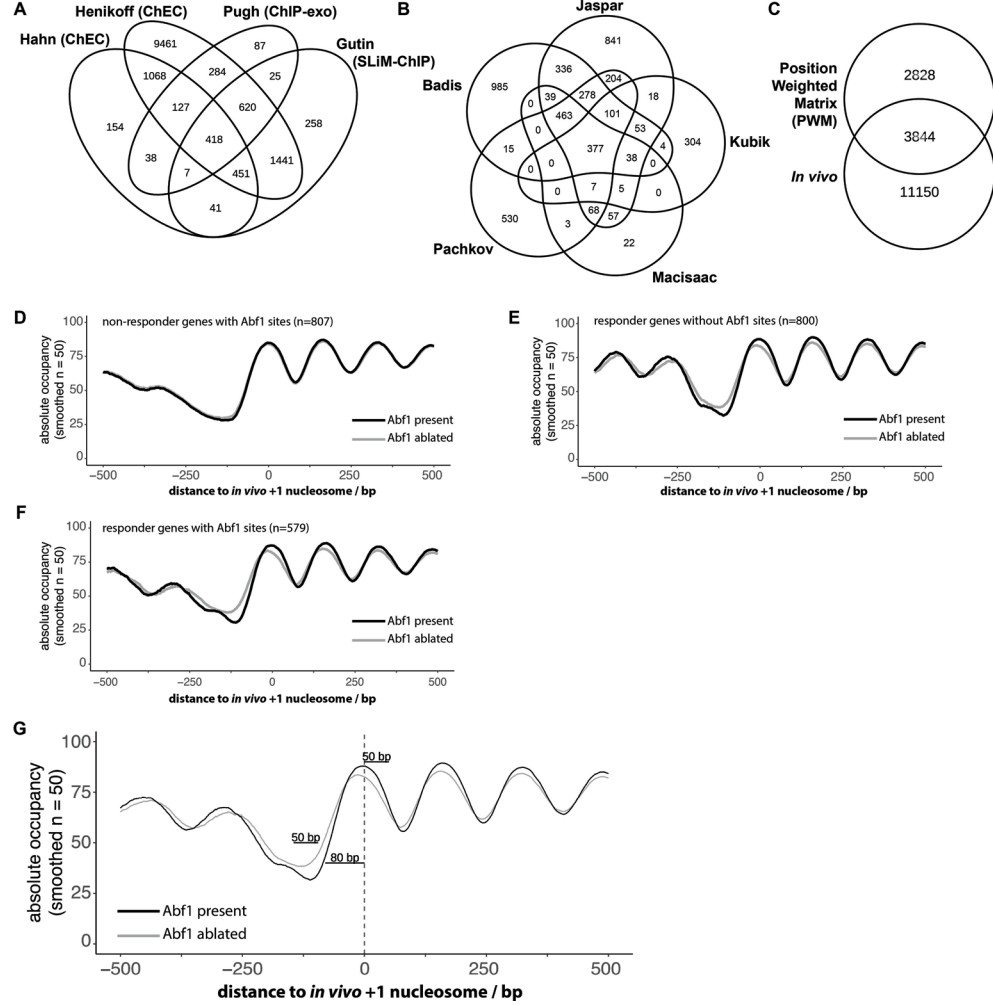

**Extended Data Fig. 7 | Responder overlap and analysis. a**, Venn diagrams comparing previously annotated Abf1 sites from in vivo annotations (Pugh group: Rossi et al. 2021, Hahn group: Mahendrawada et al. 2025, Henikoff group: Zentner et al. 2025, Gutin et al. 2018)[41,74,128,130]. We allow a generous 75 bp window for overlapping sites. **b**, Comparison between Abf1 sites predicted from published PWMs (Badis et al. 2008, Kubik et al. 2018, JASPAR data base: Rauluseviciute et al. 2024, MacIsaac et al. 2006, Pachkov et al. 2013)[35,69,125–127]. Again, we allow a generous 75 bp window for overlapping sites. **c**, Comparison between all in vivo annotated sites, as in **a**, versus all PWM-predicted sites, as in **b**. We recapitulated the previous conclusion that binding sites mapped in vivo need not contain a sequence motif. **d**–**f**, In vivo +1 nucleosome-aligned composite plots of ODM-seq data for wild-type Abf1 (Abf1 present) and no Abf1 (Abf1 ablated), shown for different gene sets, as defined by Mahendrawada et al. 2025 (ref. 74). Abf1 binding sites were called by ChEC-seq signal in the region of −400 bp to +200 bp relative to the transcription start site. Responder genes were called by significant changes in nascent RNA levels after Abf1 ablation. **d**, Comparison for non-responder genes (*n* = 807). **e**, Comparison for responder genes without Abf1 sites (*n* = 800). **f**, Comparison for responder genes with Abf1 sites (*n* = 579). **g**, Schematic showing the three windows (NFR, upstream and downstream flank of in vivo +1 nucleosome position) used for PCA in Fig. 6f. ODM-seq data for wild-type Abf1 (Abf1 present) and no Abf1 (Abf1 ablated) are aligned at in vivo +1 nucleosome positions of responder genes (*n* = 1,378) as defined by Mahendrawada et al. 2025 (see also **d**–**f**)[74].

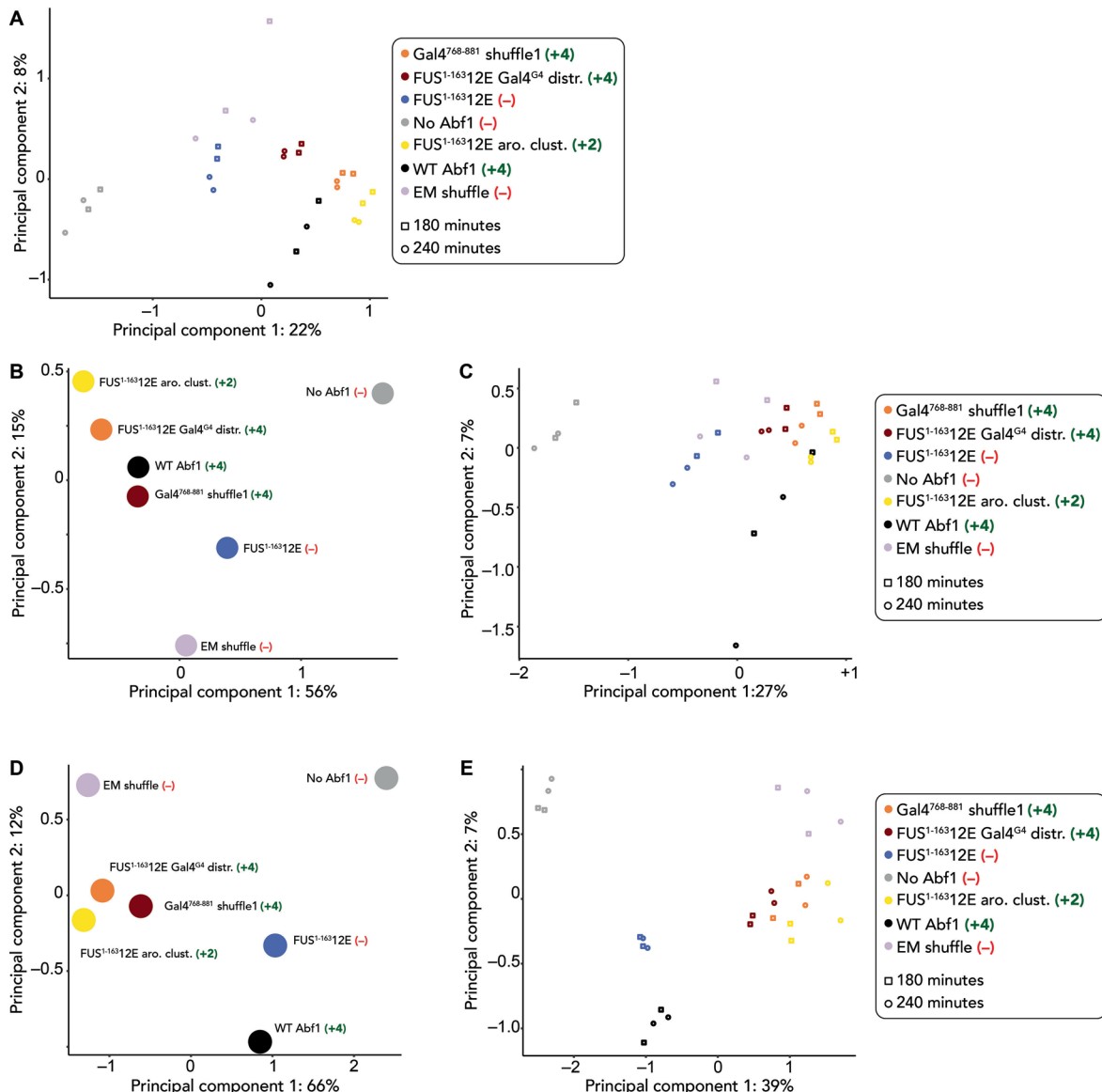

**Extended Data Fig. 8 | ODM-seq PCA analysis. a**, PCA as shown in Fig. 6f, but with individual biological and technical replicates. The 180 and 240 min incubation times with M.SssI were in saturation (Extended Data Fig. 6a) and could therefore serve as technical replicates. **b**, Examining PCA for the downstream flank of the +1 nucleosome. This PCA is as in Fig. 6f, but for the downstream flank of the +1 nucleosome (Extended Data Fig. 7g). **c**, As in **b**, but for the biological and technical replicates individually (see **a**). **d**, PCA as in Fig. 6f, but for the NFR window (Extended Data Fig. 7g). **e**, As in **d**, but for the biological and technical replicates individually (see **a**).

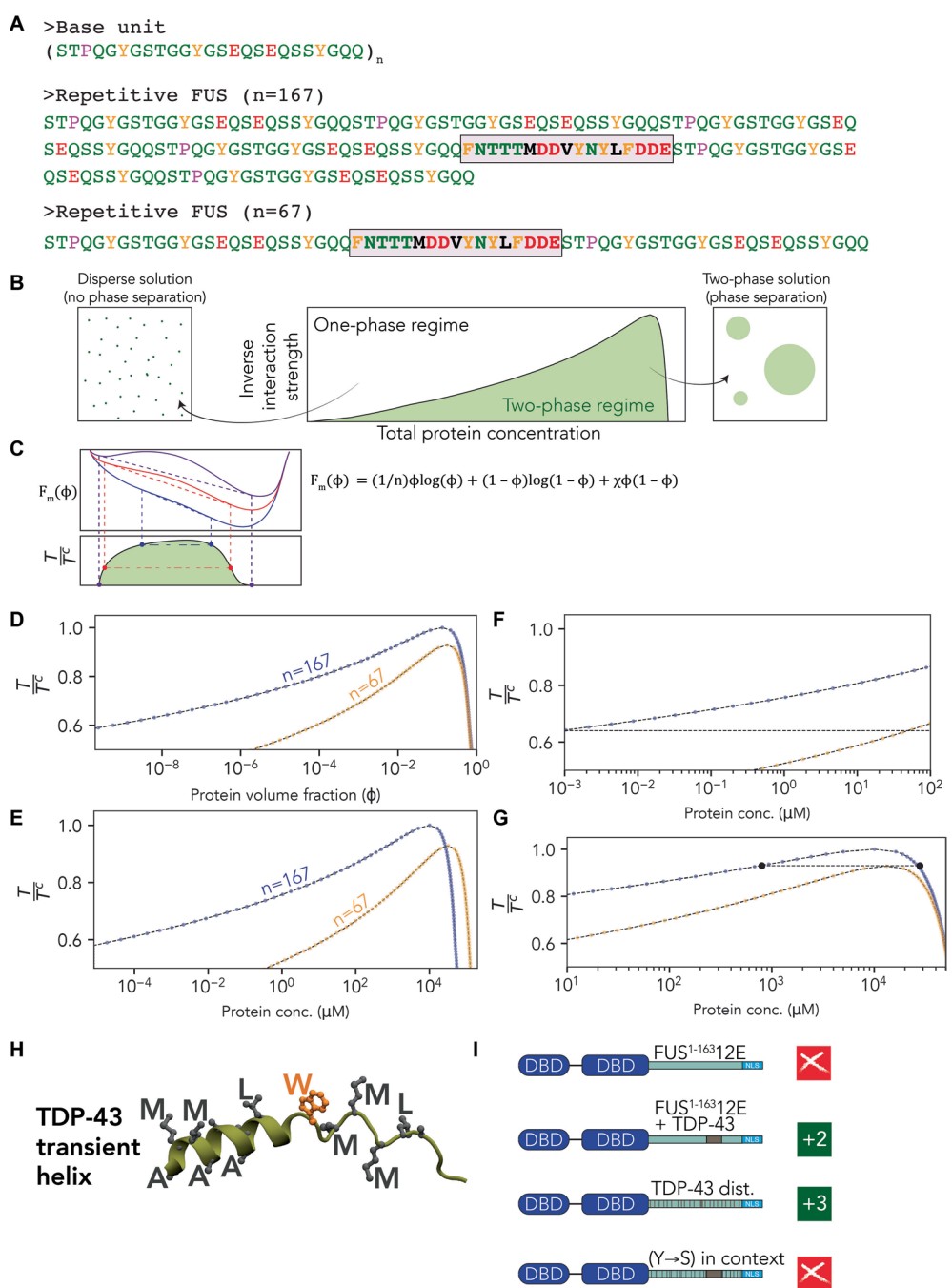

**Extended Data Fig. 9 | See next page for caption.**

**Extended Data Fig. 9 | Phase separation analysis. a**, Intracellular phase transitions have been implicated in the molecular basis for chromatin organization. Although FUS[1–163]12E is, in principle, highly soluble, could our results be interpreted in terms of a phase-transition mode in which Abf1 undergoes homotypic phase separation? We emphasize that the theoretical investigation here focuses on a model in which IDR2/FUS[1–163]12E self-assembles into de novo phase-separated assemblies. It in no way speaks to a model in which Abf1 partitions into extant condensates, where recruitment is defined by the presence/abundance of an interacting partner in those condensates. To investigate this, we used Flory–Huggins theory to predict the expected shift in the saturation concentration ($c_{sat}$) with IDR length. We designed a basic repeating unit (top) and used it to generate $n = 6$ and $n = 2$ synthetic polymer units (middle and bottom). **b**, Flory–Huggins theory allowed us to predict an expected change in saturation concentration for two polymers with identical attractive interactions but of different lengths. As such, in a phase separation model, we can predict how changing the IDR length would shift the saturation concentration. **c**, Constructing phase diagrams requires reconstruction from the free energy of mixing to extract coexisting phases. The expression for the free energy of mixing is shown to the right. **d**, Phase diagram for $n = 67$ and $n = 167$ polymers in reduced temperature ($y$ axis) versus volume fraction ($x$ axis) space. Reduced temperature is an effective parameter for tuning intramolecular interactions up to the critical temperature (reduced temperature = 1.0). As such, the $y$ axis position reflects the normalized intermolecular interaction between polymers. With this in mind, we can convert from volume fraction into molar concentration and ask how, at any arbitrary intermolecular interaction strength, the saturation concentration (in µM) would change in response to a change in IDR length. **e**, Full phase diagram showing a change in molar concentration of the phase diagram expected for the two IDRs of lengths 67 and 167 residues. **f**, Under concentration regimens compatible with cellular conditions, the change from 167 to 67 residues yielded a predicted change in saturation concentration of almost five orders of magnitude. **g**, Even at the top of the phase diagram, where the difference is minimal, we found a concentration difference of almost two orders of magnitude. **h**, Given that the EM appeared to function as a hydrophobic helix, we wondered if we could rationally design an IDR with a hydrophobic helix. The conserved region (CR) from the human RNA-binding protein TDP-43 forms a transient helix. The structural model of this helix shows hydrophobic residues lining up along a consistent interface. **i**, Inserting the 25-residue CR from TDP-43 into the (inviable) FUS[1–163]12E context rendered it viable, albeit with a +2 growth score. This result was initially interpreted as indicating that the TDP-43 helix could function as a bona fide motif. However, redistribution of the helix residues was also viable (with a better growth score of +3), indicating the importance of performing motif-redistribution controls to establish whether an inserted sub-sequence is a bona fide motif or not. We also confirmed that—even in the presence of the TDP-43 CR region—altering the context by removing aromatic residues yielded an inviable construct, mirroring results observed in other systems.

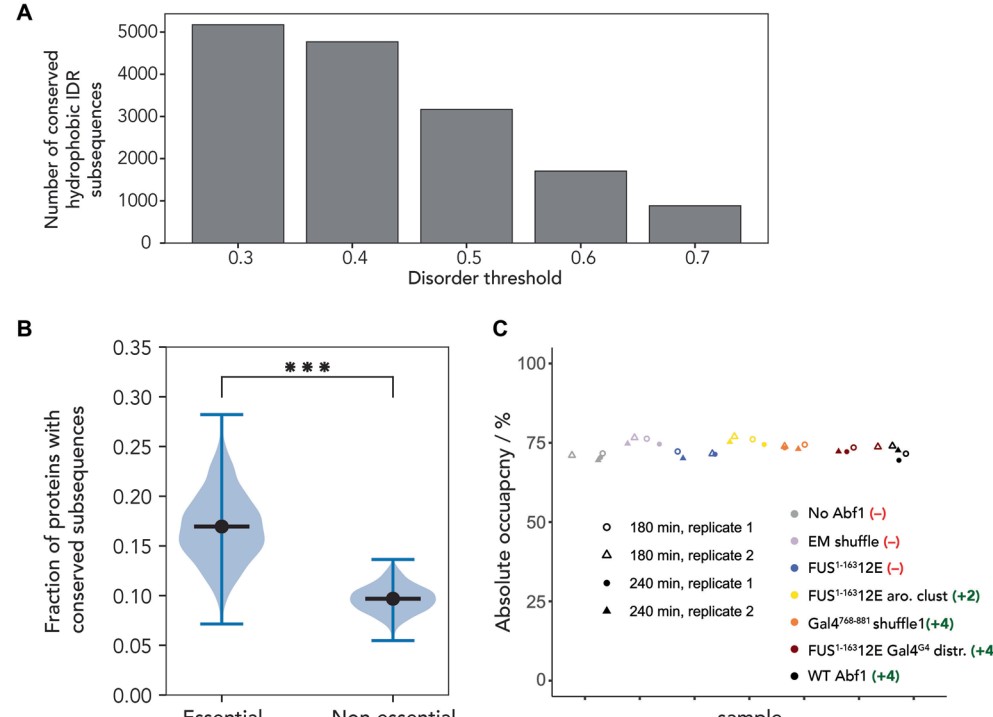

**Extended Data Fig. 10 | Conserved sub-sequences and ODM-seq saturation analysis. a**, Using the conservation threshold of 0.65 (Extended Data Fig. 1b–d), we identified sub-sequences centred on a ±6-residue window containing conserved hydrophobic residues to identify conserved subregions. Hydrophobic residues here were defined as I, L, M, V, Y, F, W. Initiator methionines were discarded from the analysis. Depending on the stringency of the disorder score for the resulting sub-sequence, between 5,173 and 884 conserved sub-sequences were identified across IDRs in the yeast genome. All sub-sequences at all disorder thresholds are reported in Supplementary Tables 10-14. **b**, Assessment of the number of conserved hydrophobic sub-sequences in essential versus

non-essential yeast proteins. On average, there are almost twice as many conserved sub-sequences in essential proteins as in non-essential proteins. The distribution was generated by bootstrapping (the black line is the overall mean) with a $P$ value below the detectable threshold using an independent $t$-test with unequal variance. The analysis here used the most stringent disorder threshold (0.7; see Extended Data Fig. 1f), but analogous results were obtained regardless of the threshold. **c**, Genome-averaged methylation reached saturation for each sample. This panel is as in Extended Data Fig. 6b, but the genome-wide average occupancy of the indicated samples and conditions is plotted.

# Reporting Summary

## Statistics

For all statistical analyses, confirm that the following items are present in the figure legend, table legend, main text, or Methods section.

| n/a | Confirmed | |
|---|---|---|
| ☐ | ☒ | The exact sample size (*n*) for each experimental group/condition, given as a discrete number and unit of measurement |
| ☐ | ☒ | A statement on whether measurements were taken from distinct samples or whether the same sample was measured repeatedly |
| ☐ | ☒ | The statistical test(s) used AND whether they are one- or two-sided<br>*Only common tests should be described solely by name; describe more complex techniques in the Methods section.* |
| ☒ | ☐ | A description of all covariates tested |
| ☒ | ☐ | A description of any assumptions or corrections, such as tests of normality and adjustment for multiple comparisons |
| ☐ | ☒ | A full description of the statistical parameters including central tendency (e.g. means) or other basic estimates (e.g. regression coefficient) AND variation (e.g. standard deviation) or associated estimates of uncertainty (e.g. confidence intervals) |
| ☐ | ☒ | For null hypothesis testing, the test statistic (e.g. *F*, *t*, *r*) with confidence intervals, effect sizes, degrees of freedom and *P* value noted<br>*Give P values as exact values whenever suitable.* |
| ☒ | ☐ | For Bayesian analysis, information on the choice of priors and Markov chain Monte Carlo settings |
| ☒ | ☐ | For hierarchical and complex designs, identification of the appropriate level for tests and full reporting of outcomes |
| ☒ | ☐ | Estimates of effect sizes (e.g. Cohen's *d*, Pearson's *r*), indicating how they were calculated |

*Our web collection on statistics for biologists contains articles on many of the points above.*

## Software and code

Policy information about availability of computer code

| Data collection | Software used here was as follows: metapredict (https://github.com/idptools/metapredict) FINCHES (https://github.com/idptools/finches), SHEPHARD (https://github.com/holehouse-lab/shephard) protfasta (https://github.com/holehouse-lab/protfasta), SPARROW (https://github.com/idptools/sparrow) |
|---|---|
| Data analysis | All code and data for all figures is provided at https://github.com/holehouse-lab/supportingdata/tree/master/2026/Langstein-Skora_2026 and also at https://github.com/gerland-group/LS_ODM_seq_analysis |

For manuscripts utilizing custom algorithms or software that are central to the research but not yet described in published literature, software must be made available to editors and reviewers. We strongly encourage code deposition in a community repository (e.g. GitHub). See the Nature Portfolio guidelines for submitting code & software for further information.

## Data

Policy information about availability of data

All manuscripts must include a data availability statement. This statement should provide the following information, where applicable:
- Accession codes, unique identifiers, or web links for publicly available datasets
- A description of any restrictions on data availability
- For clinical datasets or third party data, please ensure that the statement adheres to our policy

Sequencing data are deposited at NCBI Gene Expression Omnibus (GEO; https://www.ncbi.nlm.nih.gov/geo/) under accession number GSE314049,  GSE314050, and

## Research involving human participants, their data, or biological material

Policy information about studies with [human participants or human data](). See also policy information about [sex, gender (identity/presentation), and sexual orientation]() and [race, ethnicity and racism]().

| | |
|---|---|
| Reporting on sex and gender | N/A |
| Reporting on race, ethnicity, or other socially relevant groupings | N/A |
| Population characteristics | N/A |
| Recruitment | N/A |
| Ethics oversight | N/A |

Note that full information on the approval of the study protocol must also be provided in the manuscript.

# Field-specific reporting

Please select the one below that is the best fit for your research. If you are not sure, read the appropriate sections before making your selection.

☒ Life sciences          ☐ Behavioural & social sciences          ☐ Ecological, evolutionary & environmental sciences

For a reference copy of the document with all sections, see [nature.com/documents/nr-reporting-summary-flat.pdf]()

# Life sciences study design

All studies must disclose on these points even when the disclosure is negative.

| | |
|---|---|
| Sample size | No specific sample sizes were chosen, other than ensuring we had at least 2 independent transformants for every strain, as is standard. |
| Data exclusions | No data were excluded. |
| Replication | At least 2 independent transforms were tested for every yeast strain. All data are reported in supplementary figures 1 and 2 and all replication attempts were successful and reported |
| Randomization | No randomization was done in this study. |
| Blinding | No blinding was done as this was not relevant. |

# Reporting for specific materials, systems and methods

We require information from authors about some types of materials, experimental systems and methods used in many studies. Here, indicate whether each material, system or method listed is relevant to your study. If you are not sure if a list item applies to your research, read the appropriate section before selecting a response.

### Materials & experimental systems

| n/a | Involved in the study |
|---|---|
| ☒ ☐ | Antibodies |
| ☒ ☐ | Eukaryotic cell lines |
| ☒ ☐ | Palaeontology and archaeology |
| ☒ ☐ | Animals and other organisms |
| ☒ ☐ | Clinical data |
| ☒ ☐ | Dual use research of concern |
| ☒ ☐ | Plants |

### Methods

| n/a | Involved in the study |
|---|---|
| ☒ ☐ | ChIP-seq |
| ☒ ☐ | Flow cytometry |
| ☒ ☐ | MRI-based neuroimaging |

## Plants

| | |
|---|---|
| Seed stocks | N/A |
| Novel plant genotypes | N/A |
| Authentication | N/A |

