## [Peer Review File · Nature Cell Biology]

Sequence and chemical specificity define the functional landscape of intrinsically disordered regions

Corresponding Author: Professor Alex Holehouse

Version 0:

Decision Letter:

Dear Professor Holehouse, dear Alex,

Thank you for your interest in submitting your work to Nature Cell Biology.

I have discussed the information you provided with my colleagues, and we think that the study sounds interesting and could be appropriate for this journal. However, given the limited information provided, we would like to evaluate the complete manuscript before deciding whether to formally review it.

Please use this link to submit the complete manuscript:

Link Redacted

Please feel free to contact me if you have any questions.

Kind regards,

Daryl

Daryl Jason Verzosa David, PhD

Senior Editor, Nature Cell Biology
Advisory Editor, npj Biological Physics and Mechanics
Nature Portfolio

Heidelberger Platz 3, 14197 Berlin, Germany
Email: daryl.david@nature.com
ORCID: <https://orcid.org/0000-0002-9253-4805>

Version 1:

Decision Letter:

*Please delete the link to your author homepage if you wish to forward this email to co-authors.

Dear Professor Holehouse,

I am sorry once again for the delay. Your manuscript, "Sequence- and chemical specificity define the functional landscape of intrinsically disordered regions", has now been seen by 3 referees, who are experts in structural biology and biomolecular condensation (referee 1); transcriptional regulation and transcriptional condensates (referee 2); and computational analyses of regulatory networks including in yeast (referee 3). As you will see from their comments (attached below) they find this work of potential interest, but have raised substantial concerns, which in our view would need to be addressed with considerable revisions before we can consider publication in Nature Cell Biology.

Nature Cell Biology editors discuss the referee reports in detail within the editorial team, including the chief editor, to identify

key referee points that should be addressed with priority, and requests that are overruled as being beyond the scope of the current study. To guide the scope of the revisions, I have listed these points below. We are committed to providing a fair and constructive peer-review process, so please feel free to contact me if you would like to discuss any of the referee comments further.

In particular, it would be essential to:

- A) Appropriately define motif vs. chemical specificity and whether or not they can be disentangled (all Reviewers).
- B) Assess with further analyses the IDR characteristics, as motifs/characteristics may have positioning effects (Reviewer #1).
- C) Assess other IDR mutants of Abf1 and their effects on Abf1 function (Reviewer #2), and effects on Abf1 interactions (Reviewer #2).
- D) All other referee concerns pertaining to strengthening existing data, providing controls, methodological details, clarifications and textual changes, should also be addressed.
- E) Finally please pay close attention to our guidelines on statistical and methodological reporting (listed below) as failure to do so may delay the reconsideration of the revised manuscript. In particular please provide:

We would be happy to consider a revised manuscript that would satisfactorily address these points, unless a similar paper is published elsewhere, or is accepted for publication in Nature Cell Biology in the meantime.

- ensure that it conforms to our format instructions and publication policies (see below and <https://www.nature.com/nature/for-authors>).

- provide a point-by-point rebuttal to the full referee reports verbatim, as provided at the end of this letter.

- provide the completed Reporting Summary (found here <https://www.nature.com/documents/nr-reporting-summary.pdf>). This is essential for reconsideration of the manuscript will be available to editors and referees in the event of peer review. For more information see <http://www.nature.com/authors/policies/availability.html> or contact me.

Nature Cell Biology is committed to improving transparency in authorship. As part of our efforts in this direction, we are now requesting that all authors identified as 'corresponding author' on published papers create and link their Open Researcher and Contributor Identifier (ORCID) with their account on the Manuscript Tracking System (MTS), prior to acceptance. ORCID helps the scientific community achieve unambiguous attribution of all scholarly contributions. You can create and link your ORCID from the home page of the MTS by clicking on 'Modify my Springer Nature account'. For more information please visit www.springernature.com/orcid.

This journal strongly supports public availability of data. Please place the data used in your paper into a public data repository, or alternatively, present the data as Supplementary Information. If data can only be shared on request, please explain why in your Data Availability Statement, and also in the correspondence with your editor. Please note that for some data types, deposition in a public repository is mandatory - more information on our data deposition policies and available repositories appears below.

Link Redacted

We would like to receive a revised submission within six months.

We hope that you will find our referees' comments, and editorial guidance helpful. Please do not hesitate to contact me if there is anything you would like to discuss.

Best wishes,

Daryl

Daryl Jason Verzosa David, PhD

Senior Editor, Nature Cell Biology
Advisory Editor, npj Biological Physics and Mechanics
Nature Portfolio

Heidelberger Platz 3, 14197 Berlin, Germany
Email: daryl.david@nature.com
ORCID: <https://orcid.org/0000-0002-9253-4805>

Reviewers' Comments:

Reviewer #1 (Remarks to the Author):

The manuscript describes a rich computational and experimental investigation into functional sequence properties within intrinsically disordered regions (IDRs), including their evolutionary conservation. The focus is on an IDR of an essential yeast protein, Abf1, and related proteins, but additional work is performed across the yeast proteome and with certain human proteins containing significant IDRs. As previously reported by others for IDRs in general (see doi: 10.1186/gb-2011-12-2-r14, for example), the authors found poor sequence alignment-based conservation for yeast IDRs, including the Abf1 IDR2 that they demonstrated to be essential. They found that composition, specifically of charged and hydrophobic residues, which they term "chemistry", is more conserved (also previously found in general by others). However, they found that orthologous sequences, even with similar composition, generally can not substitute for the yeast Abf1 IDR2. In parallel and throughout the work, the authors used rational design of IDR sequences to test hypotheses (with yeast viability as a readout) regarding the link between sequence and function. They identified a short region in the IDR2 as the "essential motif", based on the inability to shuffle this sequence and retain viability. The "chemistry" around the motif, termed "context", was found to be critically important, negating a simplistic definition of a motif and leading the authors to propose that sequence-specific motifs and chemistry are two axes of a binding landscape, performing coarse-grain simulations to illustrate this concept. The authors went on to reveal that the motif could be removed as long as other changes to the chemistry were made to compensate, and to state that: "... chemically specific interactions are sufficient and necessary. In contrast, sequence-specific interactions are, in fact, essential only in a small window of chemical contexts." This view is counter to a traditional understanding of protein-protein interactions, but in line with the perspective of chemical group-driven biomolecular condensation. Then the authors performed *in silico* and *in vitro* evolutionary selection experiments constrained for binding and found that both yielded sequences that could not be aligned, but had more conserved chemistry. These results led to the statement: "... our results demonstrate that IDRs can conserve chemical specificity despite appearing unconserved based on alignment-based analysis". Further experimental studies were performed to understand the biological role of Abf1 and its IDRs.

Overall, the work is a fascinating and extensive study of the inter-relationship between classical binding motifs and distributed "chemistry" within IDR sequences, with functional implications. The text is generally clear and the figures are good tools to understand the work. The methods used are varied (and seem to be done well, although this reviewer lacks expertise to judge them all), including bioinformatics, IDR design, simulations and experiment. While some of the highlighted conclusions are expected (given previous work on conservation and functional correlates of sequence properties such as composition in the absence of ability to be aligned - which should be better acknowledged), the experiments are creative and provide very clear demonstrations of the concepts. Importantly, these concepts are extremely significant for understanding biological function and the evolution of protein sequences. The way that experimental data have been used to underscore the concepts emerging from the *in silico* analyses makes this work highly valuable for demonstrating non-standard views. The paper should therefore be of interest to a broad audience.

Specific points to be addressed:

(1) p7: "We assumed that the conservation of composition seen for IDR2 would explain functional conservation, and anticipated that orthologous IDRs with similar compositions would support viability in *S. cerevisiae*. To test this, we took eighteen Abf1 orthologs, identified their IDRs corresponding to *S. cerevisiae* IDR2449-662 from the full-length proteins (Fig. 2C), and replaced the *S. cerevisiae* IDR2449-662 in our test plasmid with each of these orthologous IDR2s." Since IDRs are not positionally conserved, the idea of defining a fragment and then positionally aligning to another species to define that fragment is not conceptually logical. This reviewer would not expect such an approach to yield functional rescue. What about this approach for the full-length IDR2 and full length IDRs from each of these species?

(2) p8: "All global shuffles were inviable, demonstrating that IDR2-like composition is insufficient for viability, implicating linear sequence-specific regions that must be essential." and "Our global shuffle variants imply IDR2 contains a SLiM." Composition is not the only sequence feature that is not linear sequence-specific, with residue patterning being key. Shuffling will disturb patterning that may be critical regardless of the presence of a "linear sequence-specific region", ie SLiM. Change "implying" to "suggesting the possibility of" and "imply" to "suggest".

(3) p14: "What if Gal(G4) and the other sequences identified in Fig. 3K were not bona fide motifs but instead altered the IDR context, albeit very locally, without being an actual motif?" Since the experiments used to test this showed viability with a redistributed "motif", it appears that the "motif" is not actually a "bona fide motif". This needs to be clearly stated.

(4) p11 and following: The narrative seems to be written to raise a straw-man hypothesis and take it down, leading to confusion about the definition of a SLiM or motif and whether in fact the EM is actually a SLiM or motif at all. It would be more clear to write the narrative from the perspective understood at the end of the text. Or, at least, if the "motifs" are not really motifs, this needs to be clearly stated.

(5) p15: Again, the text states in various places that if a global shuffle is inviable that this implies the presence of a motif, which is not the case as patterning can be affected by shuffling, and local "chemistry" may also be affected.

(6) p16: "Taken together, these results suggest multiple sequence alignments may be largely blind to conservation of chemical specificity" and "our results demonstrate that IDRs can conserve chemical specificity despite appearing unconserved based on alignment-based analysis." and p17 "In short, we conclude that the conservation of chemical specificity is widespread across the yeast proteome."

This has been demonstrated before, if "chemical specificity" can be understood as a combination of compositional bias, charge and patterning and other sequence features that do not rely on positional alignment; references should be given including doi: 10.1073/pnas.161478711 and doi: 10.7554/eLife.46883.

(7) p23: "Secondly, functionally important features can relocate across an IDR-containing protein. In this model, the specific location of a binding motif in the protein may be relatively unimportant, such that motifs can be lost from one region and emerge in another. An intriguing prediction from this model is that we should expect motifs to rapidly appear and disappear from a given IDR, a prediction supported and compatible with previous work on ex nihilo motif evolution (121)." This is an essential point. The motif can appear and disappear from an IDR (or from a fragment of an IDR) but not as easily from the protein as a whole (or from the IDR as a whole). The work should have been done with all IDRs from the full length protein, and definitely not with an IDR fragment, as noted in point (1). Obviously redoing the work is not feasible, but the issue should be acknowledged more clearly.

Reviewer #1 (Remarks on code availability):

The github has not been updated in the past 3 months (with a note that it would be updated upon submission, but that has clearly not happened). The github should have scripts for analysis of the data. It currently only has supplementary data files.

Reviewer #2 (Remarks to the Author):

IDRs are known to function through short linear motifs (SLiMs), which engage in sequence-specific binding, as well as through distributed multivalent interactions that driven by sequence-encoded chemical properties. This study investigates the interplay between these two modes of interactions by focusing on the IDR of the essential chromatin-binding protein Abf1 in *S. cerevisiae*. Using viability as the functional readout, the authors demonstrate that the IDR2 within Abf1 is essential and contains a crucial SLiM. They also identified two sequence motifs from Abf1 and Gal4 that can rescue Abf1 IDR2 function. Interestingly, these motifs do not function as a motif but instead modify the local chemical context, suggesting that a binding motif essential in the wild-type IDR can be removed if compensatory changes to the sequence chemistry are made. Lastly, they assessed the impact of IDR2 mutants on chromatin and transcription. Surprisingly, IDR mutants exhibited variable effects on chromatin organization and gene expression, which do not clearly correlate with observed viability phenotypes.

Overall, through comprehensive mutagenesis and in silico modeling, the authors thoroughly characterized the sequence and chemical determinants of Abf1 IDR2 that contribute to its essential function in viability. Based on these data, the authors propose that IDR-mediated interactions and function are driven by sequence and chemical specificity. This part of the work is well-executed, of high quality, and the insights are significant. However, the major concern I have is that the mechanistic link between IDR variants, Abf1 chromatin function, and viability remains unclear and somewhat confusing. Additional experiments and analysis are needed to support the conclusion that sequence, and chemical specificity-driven IDR interactions are critical to the molecular function of this protein. Depending on the results, the main claims may need modifications and clarifications.

Major comments

The link between IDR variants, Abf1 chromatin function, and viability remains unclear. I have the following comments regarding this part of the study.

1. The key conclusion is that sequence and chemical specificity define IDR function. As a chromatin-binding protein, Abf1 is presumed to function through chromatin. However, the authors found that IDR mutants have diverse effects on transcription and nucleosome regulation that do not seem to correlate with viability (Fig. 6). How to interpret these data? What molecular functions of Abf1 might underlie the link between IDR sequence/chemical features and viability?
2. The selection of IDR mutants for Figure. 6 is not well-justified. While WT Abf1, no Abf1, and EM shuffle conditions are highly relevant, the rationale for other mutants is unclear other than the viability phenotypes. Selecting mutants that directly test key claims from the first part of the study, for example, SLiM sufficiency (e.g. FUS-12E vs FUS-12E+EM), context dependence (Fig. 3K), motif functioning via context modification (e.g. Gal4-G4 insertion vs distribution) would provide insights into how sequence/chemistry features regulate molecular function (nucleosome phasing, gene expression etc).
3. The authors speculate that chromatin function differences between IDR mutants could arise from different protein interactions. It is crucial to test a few IDR variants to evaluate changes in the interactome, particularly whether SLiM-mediated interactions can be compensated by chemically specific interactions – a key claim of this study.
4. Using ODM-seq, the authors identified a more comprehensive list of Abf1 responder sites and subsets responding differently to Abf1 loss and various IDR mutants (Fig. 6D-G). A more detailed analysis to classify these subsets, examine overlaps, and correlate these changes with gene expression would help to better understand how IDR mutants regulate chromatin function.
5. Based on the current and potentially new data, IDR mutants with similar sequence or chemistry features and viability outcomes could function very differently on chromatin. It also remains to be determined whether SLiM mediated molecular interactions and functions can be compensated by sequence-encoded chemistry properties. This suggests that the conclusion, “sequence- and chemical specificity define the function of IDR”, may require clarification or modification depending on the specific functions being referenced.

Reviewer #3 (Remarks to the Author):

Overall, this is a very interesting study about the sequence-function relationships in IDRs. The authors have chosen an powerful model system where an IDR is essential in yeast, so they can get an easy readout of function.

The authors present compelling evidence for the 'chemical specificity' being critical for IDR function, rather than the order of amino acids in the IDR. Although similar results have been found in other studies, to my knowledge, this is by far the most complete set of experiments assembled to provide evidence for this hypothesis in one IDR.

In my view, however, there are two major issues with the presentation or 'framing' of the evidence presented in this paper. The authors spend much of the introduction and results creating a dichotomy between what they claim are 'orthogonal' aspects of IDR function, namely 'motifs' and 'chemical specificity'. They claim to demonstrate a new finding that these two aspects are interchangeable in their IDR. The two major issues with this are

1. as I understand the evidence presented, they never actually reach the true standard in the field for definition of a new motif (they never identify the binding partner), so the simplest explanation for their observations is that they only have chemical specificity in their IDR.
2. they neglect previous discussion in the field that 'motifs' and 'chemical specificity' can be interchangeable. They claim "These two modes of interaction are generally considered to drive orthogonal types of molecular recognition – that is, loss of a SLiM cannot be compensated for by changing the degree of multivalency outside of the SLiM (16, 22)" and later "Our results imply that sequence context and the presence/absence of SLiMs can compensate for and buffer one another, providing a key missing piece in our understanding of the sequence constraints on IDR evolution." However, for example, many years ago Jim Ferrell's group (PMID: 22078888) argued for the evolutionary interchangeability of protein phosphorylation (determined by SLiMs) and charge. Hence, at least in that case, the connection between what the authors claim are 'orthogonal' is well appreciated. The authors are overstating the novelty of this "missing piece" of understanding.

Hence, the 'story' of this paper should be greatly simplified. As I understand it, the authors present several figures of data

supporting the idea that they are studying a SLiM because: 1. identification of a predicted helix forming region that is essential for function, and can be knocked out with proline mutations 2. sufficiency experiment by adding that segment into a totally unrelated human IDR

Unfortunately, however, they later realize that they can 'distribute' the residues from the SLiM, so that in fact it is not clear if they should be considered SLiM at all, but rather 'context' properties or 'chemical specificity'.

These observations highlight that what they considered very convincing experiments to demonstrate SLiM-ness are not enough. To really define a new motif, it is usually expected that researchers also identify the (folded?) binding partner that actually recognizes it. I think the authors should refer back to the classical literature on defining motifs where this was appreciated. For example, in the 1980s and 1990s it was not considered sufficient to identify even an example of a known motif without demonstrating that it specifically interacted with the expected partner. A new motif could only be defined once the binding partner was known. Hence, the authors have in some ways rediscovered rigorous molecular biology.

In the current paper, the authors present this as a long narrative of discovery about SLiMs, but in the end, I don't think that we have evidence for a bona fide motif in this paper. Hence, I think most of this story is unnecessary and is likely to confuse readers who will think there is a bona fide motif in Abf1. (Just because the authors may have been confused and thought they had discovered a motif, is not a reason to confuse the reader about this.)

For example, the authors write:

"Indeed, even though this design was wholly artificial and even though wild-type IDR2 requires a bona fide motif, this design was viable with near wildtype-like growth (Fig. 4F). This result illustrates how an IDR with an essential SLiM can be rescued using a rationally-designed IDR that is lacking a SLiM if instead compensatory chemical groups are presented."

"Here, we find that even for an IDR that depends on a bona fide motif, alternative sequences with rationally interpretable chemical features can uphold function without any motif. "

However, the authors have not identified the binding partner for what they claim is a "bona fide motif". Hence they have not really reached the standard in the field. To demonstrate a new SLiM, studies need to show evidence of motif-dependent direct, physical interaction with a specific binding partner. Figures 4A,B and D shows the authors conceptual model of binding of a SLiM in an IDR to a folded partner. But as far as I can tell, there is no evidence for this mechanism presented in the paper, and the evidence presented is more simply explained without the SLiM. The most parsimonious explanation for their results seems to be that there is no SLiM at all, but rather distributed properties in their IDRs. As they finally report:

"In sum, we found we can score each of our variants based on a hydrophobicity score (enhanced by aliphatic and aromatic residues, reduced by proline) and a charge score (reflecting the net charge of the sequence) (Fig. 4J). In this empirical chemical space, viable and inviable constructs predominantly fall on either side of a dividing line, with Pho41-249 a notable exception (fig. S21)."

The authors need to greatly reduce or remove the discussion of 'bona fide motifs' in their paper, and make the paper about what they actually found: chemical specificity.

Methods should be written concisely, but should contain all elements necessary to allow interpretation and replication of the results. As a guideline, Methods sections typically do not exceed 3,000 words. The Methods should be divided into subsections listing reagents and techniques. When citing previous methods, accurate references should be provided and any alterations should be noted. Information must be provided about: antibody dilutions, company names, catalogue numbers and clone numbers for monoclonal antibodies; sequences of RNAi and cDNA probes/primers or company names and catalogue numbers if reagents are commercial; cell line names, sources and information on cell line identity and authentication. Animal studies and experiments involving human subjects must be reported in detail, identifying the committees approving the protocols. For studies involving human subjects/samples, a statement must be included confirming that informed consent was obtained. Statistical analyses and information on the reproducibility of experimental results should be provided in a section titled "Statistics and Reproducibility".

All Nature Cell Biology manuscripts submitted on or after March 21 2016 must include a Data availability statement as a separate section after Methods but before references, under the heading "Data Availability". For Springer Nature policies on data availability see <http://www.nature.com/authors/policies/availability.html>; for more information on this particular policy see <http://www.nature.com/authors/policies/data/data-availability-statements-data-citations.pdf>. The Data availability statement should include:

- Accession codes for primary datasets (generated during the study under consideration and designated as "primary accessions") and secondary datasets (published datasets reanalysed during the study under consideration, designated as "referenced accessions"). For primary accessions data should be made public to coincide with publication of the manuscript. A list of data types for which submission to community-endorsed public repositories is mandated (including sequence, structure, microarray, deep sequencing data) can be found here <http://www.nature.com/authors/policies/availability.html#data>.
- Unique identifiers (accession codes, DOIs or other unique persistent identifier) and hyperlinks for datasets deposited in an approved repository, but for which data deposition is not mandated (see here for details <http://www.nature.com/sdata/data-policies/repositories>).
- At a minimum, please include a statement confirming that all relevant data are available from the authors, and/or are included with the manuscript (e.g. as source data or supplementary information), listing which data are included (e.g. by figure panels and data types) and mentioning any restrictions on availability.
- If a dataset has a Digital Object Identifier (DOI) as its unique identifier, we strongly encourage including this in the

Reference list and citing the dataset in the Methods.

We recommend that you upload the step-by-step protocols used in this manuscript to protocols.io. More details can be found at <https://www.protocols.io/help/publish-articles>.

All imaging data should be accompanied by scale bars, which should be defined in the legend.

Cropped images of gels/blots are acceptable, but need to be accompanied by size markers, and to retain visible background signal within the linear range (i.e. should not be saturated). The boundaries of panels with low background have to be demarked with black lines. Splicing of panels should only be considered if unavoidable, and must be clearly marked on the figure, and noted in the legend with a statement on whether the samples were obtained and processed simultaneously. Quantitative comparisons between samples on different gels/blots are discouraged; if this is unavoidable, it should only be performed for samples derived from the same experiment with gels/blots were processed in parallel, which needs to be stated in the legend.

SUPPLEMENTARY INFORMATION – Supplementary information is material directly relevant to the conclusion of a paper, but which cannot be included in the printed version in order to keep the manuscript concise and accessible to the general

reader. Supplementary information is an integral part of a Nature Cell Biology publication and should be prepared and presented with as much care as the main display item, but it must not include non-essential data or text, which may be removed at the editor's discretion. All supplementary material is fully peer-reviewed and published online as part of the HTML version of the manuscript. Supplementary Figures and Supplementary Notes are appended at the end of the main PDF of the published manuscript.

The total number of Supplementary Figures (not including the "unprocessed scans" Supplementary Figure) should not exceed the number of main display items (figures and/or tables (see our Guide to Authors and March 2012 editorial <http://www.nature.com/ncb/authors/submit/index.html#supinfo>; <http://www.nature.com/ncb/journal/v14/n3/index.html#ed>). No restrictions apply to Supplementary Tables or Videos, but we advise authors to be selective in including supplemental data.

GUIDELINES FOR EXPERIMENTAL AND STATISTICAL REPORTING

REPORTING REQUIREMENTS – We are trying to improve the quality of methods and statistics reporting in our papers. To that end, we are now asking authors to complete a reporting summary that collects information on experimental design and reagents. The Reporting Summary can be found here <https://www.nature.com/documents/nr-reporting-summary.pdf>. If you would like to reference the guidance text as you complete the template, please access these flattened versions at <http://www.nature.com/authors/policies/availability.html>.

----- Please don't hesitate to contact NCB@nature.com should you have queries about any of the above requirements ----

Version 2:

Decision Letter:

Our ref: NCB-A55690B

12th September 2025

Dear Dr. Holehouse,

Thank you for submitting your revised manuscript "Sequence- and chemical specificity define the functional landscape of intrinsically disordered regions" (NCB-A55690B). It has now been seen by the original referees and their comments are below. The reviewers find that the paper has improved in revision, and therefore we'll be happy in principle to publish it in Nature Cell Biology, pending minor revisions to satisfy the referees' final requests and to comply with our editorial and formatting guidelines.

Thank you again for your interest in Nature Cell Biology. Please do not hesitate to contact me if you have any questions.

Sincerely,
Daryl

Daryl Jason Verzosa David, PhD

Senior Editor, Nature Cell Biology
Advisory Editor, npj Biological Physics and Mechanics
Nature Portfolio

Heidelberger Platz 3, 14197 Berlin, Germany
Email: daryl.david@nature.com
ORCID: <https://orcid.org/0000-0002-9253-4805>

Reviewer #1 (Remarks to the Author):

The authors have addressed the reviewer comments by extensive new data and textual revisions. The body of work is extremely valuable for understanding the relationship between IDR sequence and function, as well as for clarifying commonly used terms such as "conservation" and "motif". The authors have provided a context for the development of the ideas presented, giving appropriate references, and have summarized the limitations of the work in a very reasonable manner. The paper will be of significant interest to a broad readership and should be highly impactful.

Reviewer #1 (Remarks on code availability):

The GitHub is still not updated, but the authors provide a good rationale.

Reviewer #2 (Remarks to the Author):

The authors provide new data showing that invariable IDR variants correlate with effects on +1 nucleosome repositioning at essential genes, thereby linking Abf1's molecular function on chromatin to viability. They appropriately acknowledge the study's limitations and discuss key open questions regarding the role of IDRs in chromatin regulation for future investigation. Overall, I find the study improved and support its publication.

Reviewer #3 (Remarks to the Author):

Overall, the authors have greatly improved their manuscript. The authors have not really addressed my comments. In the introduction they have clarified the claim that chemical specificity and motifs are orthogonal, but only in the discussion do they mention that there is previous work that has explored the connection. For example, in 2019 (PMID: 31134302) it was reported that charge contributes to the binding of PIP-boxes. Thanks to the revisions, I now appreciate that the authors are trying to say something more general here, i.e., a motif can be completely replaced by multivalent properties, but I still believe they should introduce their work in the scholarly context. Similarly, the authors still make statements like this:

"Conventionally speaking, the ability³²³ to insert a short (<20 residue) sequence into a non-functional IDR context and confer function is³²⁴ interpreted as a simple and unambiguous demonstration of a bona fide modular motif (e.g., a³²⁵ SLiM). Given that sequence-specific motifs are often defined by three characteristics – an inability³²⁶ to tolerate shuffling (Fig. 3C), sensitivity to point mutations (Fig. 3G), and autonomous modular³²⁷ activity (Fig. 3I), our results confirm that the EM is a bona fide motif."

Can the authors at least provide some references for their understanding of the 'conventional' evidence for bona fide motifs?

These are not the unambiguous demonstrations of motifs that I am familiar with in the literature. In my experience is that what is considered unambiguous demonstration of motifs are structural studies of the binding partner and the linear motif (as shown as a cartoon by the authors in figure 4A), along with experiments in cells and in vitro where the effects of mutations on measured interactions are predicted by the structures.

Version 3:

Decision Letter:

Dear Dr Holehouse,

I am pleased to inform you that your manuscript, "Sequence and chemical specificity define the functional landscape of intrinsically disordered regions", has now been accepted for publication in *Nature Cell Biology*.

Over the next few weeks, your paper will be copyedited to ensure that it conforms to *Nature Cell Biology* style. Once your paper is typeset, you will receive an email with a link to choose the appropriate publishing options for your paper and our Author Services team will be in touch regarding any additional information that may be required.

Publication is conditional on the manuscript not being published elsewhere and on there being no announcement of this work to any media outlet until the online publication date in *Nature Cell Biology*.

Please note that *Nature Cell Biology* is a Transformative Journal (TJ). Authors may publish their research with us through the traditional subscription access route or make their paper immediately open access through payment of an article-processing charge (APC). Authors will not be required to make a final decision about access to their article until it has been accepted. <https://www.springernature.com/gp/open-research/transformative-journals> Find out more about Transformative Journals

Authors may need to take specific actions to achieve compliance with funder and institutional open access mandates. If your research is supported by a funder that requires immediate open access (e.g. according to <https://www.springernature.com/gp/open-science/plan-s-compliance> Plan S principles or the <https://www.springernature.com/gp/open-science/us-federal-agency-compliance> NIH public access policy) then you should select the gold OA route, and we will direct you to the compliant route where possible. Because authors warrant under our subscription licensing terms that they haven't committed to licensing any version of their article under a licence inconsistent with the terms of our agreement – including the applicable embargo period – publication under the subscription model isn't suitable for authors whose funders require no embargo.

If you have not already done so, we strongly recommend that you upload the step-by-step protocols used in this manuscript to protocols.io (<https://protocols.io>), an open online resource that allows researchers to share their detailed experimental know-how. All uploaded protocols are made freely available and are assigned DOIs for ease of citation. Protocols and Nature Portfolio journal papers in which they are used can be linked to one another, and this link is clearly and prominently visible in the online versions of both. Authors who performed the specific experiments can act as primary authors for the Protocol as they will be best placed to share the methodology details, but the Corresponding Author of the present research paper should be included as one of the authors. By uploading your Protocols onto protocols.io, you are enabling researchers to more readily reproduce or adapt the methodology you use, as well as increasing the visibility of your protocols and papers. You can also establish a dedicated workspace to collect your lab Protocols. Further information can be found at <https://www.protocols.io/help/publish-articles>.

Nature Cell Biology encourages authors presenting evidence for cell, biological, molecular, and genetic interactions to consider communicating these findings using Biofactoid (<https://biofactoid.org/>). This tool helps users share a searchable representation of interactions (e.g. binding, gene expression, post-translational modification) between genes, gene products, or chemicals. Information added to Biofactoid, with author attribution, is shared on social media and public databases, such as Pathway Commons, where it can be discovered and analyzed in the context of a large and growing corpus of knowledge.

With kind regards,

Daryl

Daryl Jason Verzosa David, PhD

Senior Editor, Nature Cell Biology
Advisory Editor, npj Biological Physics and Mechanics
Nature Portfolio

Heidelberger Platz 3, 14197 Berlin, Germany
Email: daryl.david@nature.com
ORCID: <https://orcid.org/0000-0002-9253-4805>

** Visit the Springer Nature Editorial and Publishing website at http://editorial-jobs.springernature.com?utm_source=ejp_NCB_email&utm_medium=ejp_NCB_email&utm_campaign=ejp_NCB for more information about our career opportunities. If you have any questions please click [here](mailto:editorial.publishing.jobs@springernature.com).

Preamble to all reviewers:

Dear reviewers,

Thank you for your time and effort in providing valuable and constructive feedback on our manuscript. We have done our best to address each of your comments and feedback with new experiments, new analyses, and – in particular – a complete rework of the experiments and data around Fig. 6 (chromatin remodelling assays).

Specifically, we have added the following new data:

- New mutant constructs
 - Sup35¹⁻¹³¹ + EM [Fig. 4C]
 - FUS¹⁻¹⁶³12E + EM shuffle [Fig. 4C]
 - FUS¹⁻¹⁶³12E + Gal4^{G4} distr. II [Fig. 4E]
 - FUS¹⁻¹⁶³12E + Gal4^{G4} shuffle [Fig. 4G]
 - FUS¹⁻¹⁶³12E + Abf1^{G4} distr. [Fig. 4H]
 - Pho4¹⁻²⁴⁹ segmental shuffle [Fig. 4K]
 - NCS-21 hydro->polar [Discussion]
- New patterning analysis
- New ODM-seq data for constructs FUS¹⁻¹⁶³12E + Gal4^{G4} distr. and Gal4⁷⁶⁸⁻⁸⁸¹ shuffle 1
- Complete re-analysis of all ODM-seq data, including new *in vitro* chromatin data, assessment of congruency across all previously reported Abf1 binding studies, and a novel functional definition of Abf1 responder sites. (Fig. 6E, 5F, Fig. S24-S37 and S46).

We have also substantially revised the main text in response to reviewer comments and critiques. We provide a “marked up” version, in which text that has been substantially revised is in red, and a “clean” version, where no such markup is included.

We thank the reviewers collectively for their time and effort, and hope our revised manuscript meets their expectations.

In our response to reviewers:

- Our response text is in blue
- Reviewer text is in black
- Updated manuscript text in red

Reviewer #1 (Remarks to the Author):

The manuscript describes a rich computational and experimental investigation into functional sequence properties within intrinsically disordered regions (IDRs), including their evolutionary conservation. The focus is on an IDR of an essential yeast protein, Abf1, and related proteins, but additional work is performed across the yeast proteome and with certain human proteins containing significant IDRs. As previously reported by others for IDRs in general (see doi: 10.1186/gb-2011-12-2-r14, for example), the authors found poor sequence alignment-based conservation for yeast IDRs, including the Abf1 IDR2 that they demonstrated to be essential. They found that composition, specifically of charged and hydrophobic residues, which they term “chemistry”, is more conserved (also previously found in general by others). However, they found that orthologous sequences, even with similar composition, generally can not substitute for the yeast Abf1 IDR2. In parallel and throughout the work, the authors used rational design of IDR sequences to test hypotheses (with yeast viability as a readout) regarding the link between sequence and function. They identified a short region in the IDR2 as the “essential motif”, based on the inability to shuffle this sequence and retain viability. The “chemistry” around the motif, termed “context”, was found to be critically important, negating a simplistic definition of a motif and leading the authors to propose that sequence-specific motifs and chemistry are two axes of a binding landscape, performing coarse-grain simulations to illustrate this concept. The authors went on to reveal that the motif could be removed as long as other changes to the chemistry were made to compensate, and to state that: “... chemically specific interactions are sufficient and necessary. In contrast, sequence-specific interactions are, in fact, essential only in a small window of chemical contexts.” This view is counter to a traditional understanding of protein-protein interactions, but in line with the perspective of chemical group-driven biomolecular condensation. Then the authors performed *in silico* and *in vitro* evolutionary selection experiments constrained for binding and found that both yielded sequences that could not be aligned, but had more conserved chemistry. These results led to the statement: “... our results demonstrate that IDRs can conserve chemical specificity despite appearing unconserved based on alignment-based analysis”. Further experimental studies were performed to understand the biological role of Abf1 and its IDRs.

Overall, the work is a fascinating and extensive study of the inter-relationship between classical binding motifs and distributed “chemistry” within IDR sequences, with functional implications. The text is generally clear and the figures are good tools to understand the work. The methods used are varied (and seem to be done well, although this reviewer lacks expertise to judge them all), including bioinformatics, IDR design, simulations and experiment. While some of the highlighted conclusions are expected (given previous work on conservation and functional correlates of sequence properties such as composition in the absence of ability to be aligned - which should be better acknowledged), the experiments are creative and provide very clear demonstrations of the concepts. Importantly, these concepts are extremely significant for understanding biological function and the evolution of protein sequences. The way that experimental data have been used to underscore the concepts emerging

from the in silico analyses makes this work highly valuable for demonstrating non-standard views. The paper should therefore be of interest to a broad audience.

We thank the reviewer for their detailed and positive assessment of our work!

Specific points to be addressed:

(1) p7: “We assumed that the conservation of composition seen for IDR2 would explain functional conservation, and anticipated that orthologous IDRs with similar compositions would support viability in *S. cerevisiae*. To test this, we took eighteen Abf1 orthologs, identified their IDRs corresponding to *S. cerevisiae* IDR2449-662 from the full-length proteins (Fig. 2C), and replaced the *S. cerevisiae* IDR2449-662 in our test plasmid with each of these orthologous IDR2s.” Since IDRs are not positionally conserved, the idea of defining a fragment and then positionally aligning to another species to define that fragment is not conceptually logical. This reviewer would not expect such an approach to yield functional rescue.

While we appreciate that often IDRs are not “positionally conserved,” this is not necessarily always the case - moreover, the N- and C-terminal boundaries of IDR are marked by two highlight conserved subregions (the second-half of the DNA binding domain and the C-terminal nuclear localization signal). These markers allow us to align intervening residues, such that while we make no claims regarding the 1:1 mapping between the residues within this region across orthologs, this is akin to taking the IDR between two folded domains (for example), since here (as there) we’re making no assumptions regarding the residues within the IDR, but instead using two conserved regions to define the start and end of an IDR. We have now updated the text to explain this clearly.

What about this approach for the full-length IDR2 and full-length IDRs from each of these species?

This is a good question. We had previously found that full-length *K. lactis* can complement in *S. cerevisiae*, even though IDR2 from *K. lactis* is *not* able to when inserted into the minimal construct used in the paper. In response to this reviewer, we investigated full-length orthologs for several species (*Z. rouxii*, *C. glabrata*, *L. kluyveri*, and *N. castellii*). As a reminder, for all four of these IDR2 in the context of *S. cerevisiae* were inviable. These species are shown on a phylogenetic tree built from sequence identity (right), where the protein name color indicates whether an IDR2 construct was viable (green) or not (red).

Excitingly, of these, *Z. rouxii*, *C. glabrata* (bottom two) are viable (growth +3 and +2, respectively) while *L. kluyveri*, and *N. castellii* were inviable. However, by Western blot, the *L. kluyveri* protein does not appear to be produced, suggesting that this protein may not be tolerated in *S. cerevisiae*. As for *N. castellii*, we confirmed it was expressed, but we obtained a ChIP–score of ca. 1, which indicates that there was no

specificity in binding to Abf1 sites versus non-sites and suggests initially that the lack of viability here may be due to disruption to its DNA binding.

Nevertheless, this is a fascinating set of results, as we now have three examples (*Z. rouxii*, *C. glabrata*, and *K. lactis*) in which IDR2 + the *S. cerevisiae* DBDs (e.g., DBD1-DBD2-IDR2 of foreign yeasts) are *not* viable, whereas the full-length (DBD1-IDR1-DBD2-IDR2) foreign yeast Abf1 orthologs are viable.

As I'm sure the reviewer can appreciate, disentangling the molecular basis for which combination of domains here is essential is clearly a next step. However, we feel doing this goes beyond the scope of the manuscript here, since we would need to test all the combinatorics of DBD1/DBD2 + IDR1+IDR2 to tease apart co-evolution within Abf1.

While we believe this will be highly informative and are actively pursuing it now, we respectfully request that the reviewer appreciate that this project has spanned four years of review/revisions, 11 years of work, and that adding even more experiments for this paper is, at this juncture, not the right strategic move! Nevertheless, we emphasize that this exploration is being actively pursued.

(2) p8: "All global shuffles were inviable, demonstrating that IDR2-like composition is insufficient for viability, implicating linear sequence-specific regions that must be essential." and "Our global shuffle variants imply IDR2 contains a SLIM."

Composition is not the only sequence feature that is not linear sequence-specific, with residue patterning being key. Shuffling will disturb patterning that may be critical regardless of the presence of a "linear sequence-specific region", ie SLIM. Change "implying" to "suggesting the possibility of" and "imply" to "suggest".

The reviewer is quite right! Firstly, we have made the requested changes to the text. Secondly, note that we actually explicitly considered the impact of residue patterning during our design approach. We controlled for this explicitly in our designs, ensuring shuffles and synthetic sequences retained a wildtype-like patterning of different residue groups to try and avoid residue patterning acting as a hidden latent variable. We did not mention this in our original submission, but this was clearly an error. While prior work has highlighted charge patterning as being important, we ensured that the patterning of all residue groups commonly found to influence IDR function was controlled for. We now quantify this explicitly in Fig. S15 and discuss it in the *Results*, and *Methods* sections.

From main text:

As an initial test, we designed three globally shuffled variants of IDR2 in which ~65% of the residues were shuffled across the sequence (**Fig. 3B**). These shuffles retain exactly the same composition, and were designed to retain wildtype-like levels of patterning for different chemical groups (**fig. S15**). All these shuffles were inviable, demonstrating that IDR2-like composition is insufficient for viability, and suggesting the presence of essential sequence-specific subregions.

and from *Methods*:

Sequence designs constrained residue patterning parameters to retain wildtype-like residue patterning (see **fig. S15**). Patterning here is quantified in terms of binary patterning using a kappa-like definition for patterning for chemically distinct residue groups, where kappa (κ) is a patterning parameter that measures the degree to which sets of residues intermix¹⁻³. A high kappa value means residues in groups are highly segregated, while a low value implies an even intermixing of residues. In the original definition of kappa, three residue groups were considered (positive, negative, and neutral), where kappa quantifies the extent of intermixing between positive and negative residues. Kappa can also be defined as a binary patterning parameter^{4,5}, where it measures the extent to which residues in one group segregate from all other residues. The other patterning parameters in **fig. S15** reflects the binary patterning of groups of amino acids noted below each datapoint vs. all other residues. For example, GSNHQT reflects how segregated residues {GSNHQT} are from all other residues. While not quantified here, all shuffle and distribution designs ensured residues were evenly distributed across the context sequence to minimize the possibility of accidentally generating an additional *ex nihilo* motif.

Further, in response to this reviewer, we designed a Pho4¹⁻²⁴⁹ variant design explicitly to shuffle regions while preserving overall sequence patterning locally. Testing this “segmental shuffle” construct yielded an inviable construct, consistent with our “global” shuffle. This new data has been added to **Fig. 4K**.

Moreover, global and segmental shuffles (in which local subregions are internally shuffled) of Pho4¹⁻²⁴⁹ are non-viable (**Fig. 4K**), supporting the idea that there may exist a *bona fide* motif within Pho4¹⁻²⁴⁹ (**fig. S1**).

(K) While Pho4¹⁻²⁴⁹ supports viability, global and segmental shuffle constructs are inviable, suggesting that Pho4¹⁻²⁴⁹ harbors a motif.

(3) p14: “What if Gal(G4) and the other sequences identified in Fig. 3K were not bona fide motifs but instead altered the IDR context, albeit very locally, without being an actual motif? ”

Since the experiments used to test this showed viability with a redistributed “motif”, it appears that the “motif” is not actually a “bona fide motif”. This needs to be clearly stated.

We agree, and have now changed this to clearly state this:

This result demonstrates that these subregions are in fact not *bona fide* motifs, but instead, the introduction of subsequences (Gal4^{G4}, p65, GR) simply altered the sequence chemistry without any reliance on the precise amino acid order. We

reiterate that the EM is a *bona fide* motif: it cannot tolerate point mutations (Fig. 3G), be shuffled (Fig. 3C), or be distributed (Fig. 4C). However, our results here suggest that viability can be achieved either with or without a *bona fide* motif.

(4) p11 and following: The narrative seems to be written to raise a straw-man hypothesis and take it down, leading to confusion about the definition of a SLiM or motif and whether in fact the EM is actually a SLiM or motif at all. It would be more clear to write the narrative from the perspective understood at the end of the text. Or, at least, if the “motifs” are not really motifs, this needs to be clearly stated.

We thank the reviewer for this suggestion and have substantially reworded this section. We do feel the original general flow of the manuscript is effective, and have explicitly had readers tell us they appreciated the way in which the paper was written. We have done our best to more explicitly state when motifs are not real motifs, and re-emphasize that the EM is a *bona fide* motif. While we do not subscribe to a model in which papers must be written in the order they were done, in this work, we felt there was some value in providing a semi-accurate historical re-enactment. Given what we now know, we would probably have done things differently. However, we appreciate that there is a need to ensure clarity concerning conclusions, and we have attempted to reword sections to achieve this. Given the extent to which these changes are integrated through the text we have not pasted specific examples in here, but they are highlighted in the marked-up copy.

(5) p15: Again, the text states in various places that if a global shuffle is inviable that this implies the presence of a motif, which is not the case as patterning can be affected by shuffling, and local “chemistry” may also be affected.

As discussed above, we agree, and (1) have now explicitly discussed this, (2) also clarified that global shuffles had residue patterning control for, and (3) included an additional design in the Pho4¹⁻²⁴⁹ segmental shuffle to test this explicitly. We do very much appreciate the reviewer’s point, however, and recognize this was an important thing to address.

(6) p16: “Taken together, these results suggest multiple sequence alignments may be largely blind to conservation of chemical specificity” and “our results demonstrate that IDRs can conserve chemical specificity despite appearing unconserved based on alignment-based analysis.” and p17 “In short, we conclude that the conservation of chemical specificity is widespread across the yeast proteome.”

This has been demonstrated before, if “chemical specificity” can be understood as a combination of compositional bias, charge and patterning and other sequence features that do not rely on positional alignment; references should be given including doi: 10.1073/pnas.161478711 and doi: 10.7554/eLife.46883.

We agree, but do note that prior work did not refer to this as chemical specificity, but instead as just sequence features, yet how those sequence features relate to molecular function has generally been less explicitly proposed. That said, we wholeheartedly agree with the reviewer and have added text in the results to clarify that others have seen similar results.

We have now revised text to state:

The presence of conserved sequence features (e.g., amino acid composition and patterning parameters) has been reported before in elegant prior work from several groups^{3,6-8}. Here, we suggest conservation operates not at the level of preserving sequence-intrinsic features, but instead conserving complementary chemical interfaces. As such, our work (and others) argues that conservation in yeast IDRs is widespread, albeit following different “rules” than conservation in folded domains.

(7) p23: “Secondly, functionally important features can relocate across an IDR-containing protein. In this model, the specific location of a binding motif in the protein may be relatively unimportant, such that motifs can be lost from one region and emerge in another. An intriguing prediction from this model is that we should expect motifs to rapidly appear and disappear from a given IDR, a prediction supported and compatible with previous work on ex nihilo motif evolution (121).”

This is an essential point. The motif can appear and disappear from an IDR (or from a fragment of an IDR) but not as easily from the protein as a whole (or from the IDR as a whole). The work should have been done with all IDRs from the full length protein, and definitely not with an IDR fragment, as noted in point (1). Obviously redoing the work is not feasible, but the issue should be acknowledged more clearly.

We appreciate the reviewer’s point, but also doing with the full-length protein from the start would likely have led us to the conclusion that the full-length orthologs are functional, implicating the IDRs as being equivalent. As such, we somewhat contend the idea that “the work should have been done with all IDRs from the full-length protein, and definitely not with an IDR fragment”. This is *one* way the work could have been done, but we disagree that it is the way the work *should* have been done. However, we raise this point of motif jumping in the Discussion now as a limitation of the study.

Second, our focus here is on a construct lacking IDR1 as a “minimal system”. However, it bears noting that if motifs can appear and disappear across different IDRs, it is possible that comparing constructs in which IDR1 and IDR2 are replaced by orthologous versions (as opposed to just IDR2) would reveal motifs “jumping” between IDRs. This is discussed below.

Reviewer #1 (Remarks on code availability):

The GitHub has not been updated in the past three months (with a note that it would be updated upon submission, but that has clearly not happened). The GitHub should have scripts for analyzing the data. It currently only has supplementary data files.

We apologize and are in the process of fully updating the GitHub repository (due to the extensive number of revision experiments we are actively doing this now, but this will, of course, be done before publication, in the next couple of weeks). With this submission, we provide a complete .zip drive with all code/and data associated with the study; however, we recognize navigating this 20 GB zip drive is not ideal, and we

are in the process of rebuilding the GitHub repository to ensure everything is clear and easy to access and linked back to the figure numbering in the final manuscript. As the reviewer can perhaps appreciate, a decade of work done across five labs necessitates substantial effort to ensure all the data are combined in a tractable way, but we are actively doing this now.

Reviewer #2 (Remarks to the Author):

IDRs are known to function through short linear motifs (SLiMs), which engage in sequence-specific binding, as well as through distributed multivalent interactions that driven by sequence-encoded chemical properties. This study investigates the interplay between these two modes of interactions by focusing on the IDR of the essential chromatin-binding protein Abf1 in *S. cerevisiae*. Using viability as the functional readout, the authors demonstrate that the IDR2 within Abf1 is essential and contains a crucial SLiM. They also identified two sequence motifs from Abf1 and Gal4 that can rescue Abf1 IDR2 function. Interestingly, these motifs do not function as a motif but instead modify the local chemical context, suggesting that a binding motif essential in the wild-type IDR can be removed if compensatory changes to the sequence chemistry are made. Lastly, they assessed the impact of IDR2 mutants on chromatin and transcription. Surprisingly, IDR mutants exhibited variable effects on chromatin organization and gene expression, which do not clearly correlate with observed viability phenotypes.

Overall, through comprehensive mutagenesis and in silico modeling, the authors thoroughly characterized the sequence and chemical determinants of Abf1 IDR2 that contribute to its essential function in viability. Based on these data, the authors propose that IDR-mediated interactions and function are driven by sequence and chemical specificity. This part of the work is well-executed, of high quality, and the insights are significant.

We appreciate the reviewer's support!

However, the major concern I have is that the mechanistic link between IDR variants, Abf1 chromatin function, and viability remains unclear and somewhat confusing. Additional experiments and analysis are needed to support the conclusion that sequence, and chemical specificity-driven IDR interactions are critical to the molecular function of this protein. Depending on the results, the main claims may need modifications and clarifications.

We have dedicated substantial resources and effort towards addressing this question, and have also, in parallel, revised the manuscript to – we hope – better align with the reviewer's concern.

Major comments

The link between IDR variants, Abf1 chromatin function, and viability remains unclear. I have the following comments regarding this part of the study.

1. The key conclusion is that sequence and chemical specificity define IDR function. As a chromatin-binding protein, Abf1 is presumed to function through chromatin. However, the authors found that IDR mutants have diverse effects on transcription and nucleosome regulation that do not seem to correlate with viability (Fig. 6). How to interpret these data? What molecular functions of Abf1 might underlie the link between IDR sequence/chemical features and viability?

We wholeheartedly agree; this is THE question! We have added new ODM-seq chromatin data for two viable Abf1 variants with wild-type-like growth rate (+4) but “context-only” IDR design (new Fig. 6E, F). Including these new data, we extended our chromatin analyses to investigate the relationship between viability phenotype and the impact of IDR variation on nucleosome organization. We report now in the revised version that inviable IDR variants correlate with effects on +1 nucleosome re-positioning for essential genes (Fig. 6F, fig. S35, S36). As gene expression in yeast is very sensitive to +1 nucleosome positions^{9–13} this links regulation of essential genes to inviability.

New text (and associated analysis) from the associated Results section is pasted below.

Abf1 IDR is required for +1 nucleosome positioning at essential genes

While we did not see a gross difference in genome-wide nucleosome organization between viable vs. inviable constructs, these composite plots of many genes may obscure small effects and/or effects at a few genes. Given the precision of our ODM-seq data, we reasoned that a more nuanced and hypothesis-driven search for changes, especially at essential genes, might reveal differences between viable and inviable Abf1 constructs.

While we had previously focused on (putative) Abf1 sites, a deep analysis indicated this may not be the right approach for two reasons. First, a comprehensive comparison of annotated Abf1 sites in yeast – identified either via *in vivo* mapping by various techniques or via mapping consensus DNA sequence motifs (position weight matrices, PWMs) – revealed a striking incongruency (fig. S32A,B), even with a generous 75 bp window for nominating “overlapping” sites. This analysis also recapitulated the prior conclusion that binding sites mapped *in vivo* need not contain a DNA sequence motif (fig. S32C)^{14,15}. Second, we were intrigued by recent work in yeast demonstrating that not all genes that change their expression in response to transcription factor ablation required a binding site for the ablated transcription factor in their promoters¹⁴. The authors provide strong arguments that this was not solely due to indirect effects. To further explore this surprising conclusion in our own data, we plotted our ODM-seq data of WT vs. no Abf1 at genes annotated in that study as (i) only having Abf1 binding sites in their promoters, (ii) only responding to Abf1 ablation, or (iii) being both responders and possessing Abf1 binding sites in their promoters. This analysis clearly showed that transcription response, but not necessarily Abf1 sites, correlated with altered nucleosome organization (fig. S33). This confirmed, on the previously unassessed level of nucleosome organization, that a response to Abf1 ablation need not be linked to an Abf1 site. It also confirmed the known correlation between alterations in nucleosome organization at the NFR/+1 nucleosome with gene expression changes^{10–13}. Therefore, we searched for changes in nucleosome organization at NFRs/+1 nucleosomes in our ODM-seq data of Abf1 variants.

As gleaned from the composite plots comparing WT vs. no Abf1 (fig. S33), we looked in our ODM-seq data for changes in three regions (fig. S34): i) the NFR region, which

may reflect differential binding of nucleosomes or transcription machinery (e.g., preinitiation complex), ii) the upstream flank of the +1 nucleosome, which usually contains the transcription start site^{15,16} and where even small upstream shifts of nucleosome positions may impact transcription regulation^{9–13}, and iii) the downstream flank of the +1 nucleosome, which also reflects shifted positions. As we were particularly interested in possible misregulation of essential genes, we performed principal component analyses (PCAs) of ODM-seq data for WT Abf1, no Abf1, and Abf1 variants in these three regions of essential genes. Strikingly, PCA clearly separated inviable from viable conditions for the upstream flank of the +1 nucleosomes of essential genes, both for the average of replicates (**Fig. 6F**) as well as for individual replicates (**fig. S35**). This was similarly the case for the downstream flank of the +1 nucleosomes (**fig. S36**), but not for the NFR region (**fig. S37**). We conclude that proper +1 nucleosome positioning, as a proxy for proper gene regulation, at essential genes depends on Abf1 IDR in a way that is not compatible with the inviable variants or the absence of Abf1.

Finally, we note that changes in Abf1 IDR-dependent +1 nucleosome positioning *in vivo* are not at odds with Abf1 IDR-independent nucleosome barrier function. +1 nucleosome positioning *in vivo* depends on varying combinations of barriers, like Abf1, with several remodelers and possibly transcription machinery^{17–19}. We speculate that the interplay with the latter may depend on Abf1 IDR, while we showed that the pure barrier function does not.

All this said, we do concede that the precise mechanism of the IDR function in nucleosome positioning remains unresolved. Notably, we explicitly discuss this in our new expanded limitations section:

Third, our functional data (**Fig. 6**) clearly show Abf1 IDR is important for +1 nucleosome positioning *in vivo*, especially at essential genes, but by a so far unknown mechanism beyond a pure nucleosome barrier function. We speculate that this mechanism may be linked to a role of GRFs in torsional insulation²⁰ or 3D chromatin organization²¹, which may also explain effects at genes without Abf1 sites, for which we cannot distinguish direct from indirect effects.

As we also note in our *Discussion*, an important step towards answering this question will be to identify the IDR interactome and how it changes for IDR variants in comparison to WT Abf1. A recent genome-wide survey of the interactomes for all yeast proteins by immunoprecipitation coupled to mass spectrometry from the Mann group²² failed to identify ANY meaningful interactors for Abf1. Given our work here, this is not actually unexpected and is consistent with a model in which Abf1 interacts transiently and dynamically with many partners via its IDR in a manner that is poorly suited to CO-IP MS.

Indeed, we are currently awaiting news on proposals being reviewed to further advance this project, particularly by employing *in vivo* proximity labeling, which may reveal the transient Abf1 IDR interactors. We have recently obtained the first high-resolution proximity labelling data, which are still preliminary but very promising and are beyond the scope of this paper. In addition, we need to address how IDR

variants impact Abf1 binding, given the intriguing results from the Barkai and Jonas groups (see recent review²³). Our proposed (and hopefully soon funded) project also entails ChEC-seq experiments to investigate how protein-DNA interactions are altered on a cellular scale as a function of different variants. These techniques can also be coupled with full-length ortholog data (see comments to reviewer 1).

In summary, while we succeeded in linking IDR inviability to the +1 nucleosome organization and, by extension, to the regulation of essential genes, we are about to enter the next phase of this project, where we will elucidate the molecular mechanism of IDR function. We also note that we are – we hope – sufficiently open about the potential limitations here:

2. The selection of IDR mutants for Figure. 6 is not well-justified. While WT Abf1, no Abf1, and EM shuffle conditions are highly relevant, the rationale for other mutants is unclear other than the viability phenotypes. Selecting mutants that directly test key claims from the first part of the study, for example, SLiM sufficiency (e.g. FUS-12E vs FUS-12E+EM), context dependence (Fig. 3K), motif functioning via context modification (e.g. Gal4-G4 insertion vs distribution) would provide insights into how sequence/chemistry features regulate molecular function (nucleosome phasing, gene expression etc).

We agree with the Reviewer that it is especially important in the context of our study to compare IDRs of the “context + motif” versus “context-only” types. The former is represented by WT Abf1, while the latter is now represented by two newly included Abf1 variants: FUS¹⁻¹⁶³12E + Gal^{G4} distr. I, Gal4⁷⁶⁸⁻⁸⁸¹ shuffle 1. We chose these as they have wild-type-like growth rates (+4) and are for sure “context-only”, which we tested in both cases via additional sequence shuffles (Gal4⁷⁶⁸⁻⁸⁸¹ shuffle 2 and 3 and newly added FUS¹⁻¹⁶³12E + Gal^{G4} distr. II). Indeed, having now three viable versus three inviable conditions in our chromatin data helped to delineate the role of the IDR in +1 nucleosome positioning at essential genes.

3. The authors speculate that chromatin function differences between IDR mutants could arise from different protein interactions. It is crucial to test a few IDR variants to evaluate changes in the interactome, particularly whether SLiM-mediated interactions can be compensated by chemically specific interactions – a key claim of this study.

We agree, as mentioned above, we are actively doing this, but we do not have the data yet (nor the funding to go in this direction further) that would allow us to pinpoint IDR interactors and respective changes depending on IDR variation. We are at least one year away from completing these studies, and given the body of data in this manuscript already, delaying its publication further (first submission was November 2021) to wait for more data is not compatible with our funding or trainee timelines. We say all this simply to emphasize that *we agree* - exploring how mutations alter the interactome is key, but we strongly feel this is best suited to our follow-up work, rather than in this study. We have more explicitly addressed this limitation in the Discussion now.

4. Using ODM-seq, the authors identified a more comprehensive list of Abf1 responder sites and subsets responding differently to Abf1 loss and various IDR mutants (Fig. 6D-G). A more detailed analysis to classify these subsets, examine overlaps, and correlate these changes with gene expression would help to better understand how IDR mutants regulate chromatin function.

Yes, we agree – this is a fantastic suggestion and underlies much of our revisions to the chromatin remodelling section. As reported now in our revised version and as prompted by the very recent publication from the Hahn group¹⁴, we changed our approach from searching for “responder sites” towards searching for “responder genes”.

The Hahn group published that responder genes need not harbour Abf1 sites in their promoters and argue that this is not solely due to indirect effects. Further, it is the regulation of genes that matters most in the end, rather than some statistically significant but potentially irrelevant chromatin alterations at some sites in the genome. We also note that Abf1 site annotations are highly inconsistent (new suppl. Fig. S32, reproduced below), which is an interesting analysis in its own right.

Fig. S32 Venn diagrams comparing previously annotated Abf1 sites in the *S. cerevisiae* genome. (A) Comparison between in vivo annotated sites^{14,15,24,25}. (B) Comparison between Abf1 sites predicted from published PWMs^{10,26–29}. (C) Comparison between all in vivo annotated sites as in panel A versus all PWM-predicted sites as in panel B. We recapitulated the prior conclusion that binding sites mapped in vivo need not contain a sequence^{14,15} motif.

As it is a bit tangential to our main topic, we do not wish to explore this point more extensively here. Nonetheless, we note that this touches upon a very intriguing point in the field of transcription factor mechanisms, that mapping of binding sites, although necessary, may be much less informative than monitoring functional responses to alterations/ablation of a transcription factor. Therefore, we focused on alterations in +1 nucleosome positioning as consequences of altered Abf1 function and as a good proxy for altered gene regulation.

We concede that it would be preferable to have matching RNA data. However, we note that our previous total RNA-seq data are of limited value, as we discuss in our manuscript, due to many indirect and secondary effects. Instead, we would need nascent RNA data. Importantly, such data need to be obtained under exactly matched conditions (strains, media, etc.) as the chromatin data to allow stringent correlations. However, we are not set up to generate these, and establishing this approach is again part of our above-mentioned grant application, but beyond our current options.

5. Based on the current and potentially new data, IDR mutants with similar sequence or chemistry features and viability outcomes could function very differently on chromatin. It also remains to be determined whether SLiM mediated molecular interactions and functions can be compensated by sequence-encoded chemistry properties. This suggests that the conclusion, “sequence- and chemical specificity define the function of IDR”, may require clarification or modification depending on the specific functions being referenced.

To paraphrase the reviewer’s concern here: “Two sequences may be viable for different molecular reasons”. This is certainly possible, and the reviewer is correct that we should more clearly state in the Discussion that we cannot rule out this possibility. We have updated the text accordingly, and we thank the reviewer because this is actually an important point we should have made explicitly and failed to in our original version. This has now been done.

First, we posit that if two IDRs offer equivalent growth phenotypes when provided in a protein where mutation/deletion of the same IDR renders yeast inviable, then these IDRs are “functionally equivalent”. This definition, however, is empirical and defined by the scope of the assay. Different IDRs may confer viability in different ways by interacting with different sets of partners. As the Abf1 interactome is revealed in future work, this question can be more directly answered.

Nonetheless, our newly added chromatin function data of the two “context-only” Abf1 variants with wild-type-like growth rate +4 (FUS¹⁻¹⁶³12E + Gal^{G4} distr. I, Gal4⁷⁶⁸⁻⁸⁸¹ shuffle 1) at least partially address this point. Even though they did not behave exactly like WT Abf1, at least one of them (FUS¹⁻¹⁶³12E + Gal^{G4} distr. I) clustered very closely to WT in PC1, which reflects most of the variability (**Fig. 6E, fig. S35**). We take this as support that we have at least one “context-only” Abf1 variant, for which we have functional data suggesting a very similar mode of action compared to the “context + motif” WT Abf1. Nonetheless, as is true with the other questions regarding

Abf1 IDR function, the future approaches discussed above will be required to rigorously answer this point.

Reviewer #3 (Remarks to the Author):

Overall, this is a very interesting study about the sequence-function relationships in IDRs. The authors have chosen an powerful model system where an IDR is essential in yeast, so they can get an easy readout of function.

The authors present compelling evidence for the 'chemical specificity' being critical for IDR function, rather than the order of amino acids in the IDR. Although similar results have been found in other studies, to my knowledge, this is by far the most complete set of experiments assembled to provide evidence for this hypothesis in one IDR.

We thank the reviewer for their positive assessment of our work!

In my view, however, there are two major issues with the presentation or 'framing' of the evidence presented in this paper. The authors spend much of the introduction and results creating a dichotomy between what they claim are 'orthogonal' aspects of IDR function, namely 'motifs' and 'chemical specificity'. They claim to demonstrate a new finding that these two aspects are interchangeable in their IDR. The two major issues with this are

1. As I understand the evidence presented, they never actually reach the true standard in the field for definition of a new motif (they never identify the binding partner), so the simplest explanation for their observations is that they only have chemical specificity in their IDR.

We appreciate the reviewer's concern here. However, we would suggest that the identification of a motif with a binding partner would reflect the definition of a specific protein:protein interaction. We are not making such a claim. Instead, we are making the claim that if a subregion of an IDR can tolerate being shuffled/distributed across a 150+ amino acid sequence, it cannot possibly be a motif that enables sequence-specific binding because the resulting binding interface no longer exists. Similarly, if a region is sensitive to individual point mutations, shuffling, and distribution (that is, those perturbations yield a non-functional protein) and can also be transplanted into another non-functional sequence to transform this non-functional protein into a functional protein, this is consistent with molecular interactions mediated by every motif reported to date.

While the reviewer is quite correct that we do not formally know what the binding partner is, we suggest that this is not a prerequisite to claim. Moreover, knowing a partner (e.g., via a Co-IP or *in vitro* binding assay) is absolutely insufficient data to make the claim that a motif exists, given that many studies have identified IDRs that do not need to possess motifs to interact with partners³⁰⁻³⁷. As such, defining a partner is not sufficient and – we would argue – not necessary to demonstrate (or at least very, very strongly suggest) the presence of a motif.

2. They neglect previous discussion in the field that 'motifs' and 'chemical specificity' can be interchangeable. They claim "These two modes of interaction are generally

considered to drive orthogonal types of molecular recognition – that is, loss of a SLiM cannot be compensated for by changing the degree of multivalency outside of the SLiM (16, 22)" and later "Our results imply that sequence context and the presence/absence of SLiMs can compensate for and buffer one another, providing a key missing piece in our understanding of the sequence constraints on IDR evolution." However, for example, many years ago Jim Ferrell's group (PMID: 22078888) argued for the evolutionary interchangeability of protein phosphorylation (determined by SLiMs) and charge. Hence, at least in that case, the connection between what the authors claim are 'orthogonal' is well appreciated. The authors are overstating the novelty of this "missing piece" of understanding.

We apologize if the message of the work was not clear. The Ferrell's lab work is beautiful, but it is not the right comparison with the idea we are proposing. If the Ferrell lab work demonstrated that you could add Asp/Glu residues *anywhere* on a protein and recapitulate interactions mediated by phosphorylation, that would be more apt. However, the Ferrell work (as an example) focuses on the exchange between phosphoresidues and phosphomimetics (not actually, but in an evolutionary sense). Central to the analysis in this work is demonstrating a precise structural basis for how salt bridges mediated between E/D residues (and an R/K) are replaced by salt bridges mediated by pS/pT and the same R/K. The exchange of chemically similar/equivalent residues is, of course, important and frequently seen in motifs. For example, most short linear motifs are defined by regular expressions, where at least a subset of the residues can tolerate a range of different (e.g., P*x*L*x*T motif, L*x*C*x*E motif, where "x" can in principle be any residue). However, if one were to randomly distribute the location of the P*x*L*x*T residues, L*x*C*x*E residues, or the serine or threonine residues from the Ferrel work elsewhere in the protein, the structurally-derived interactions these residues engage in would no longer be accessible. The contrast we make in our work is that for IDRs, such a complete redistribution of these residues can be tolerated if the model of intermolecular interaction is via chemical specificity (as opposed to sequence specificity). These are the orthogonal modes in discussion.

Hence, the 'story' of this paper should be greatly simplified. As I understand it, the authors present several figures of data supporting the idea that they are studying a SLiM because: 1. identification of a predicted helix forming region that is essential for function, and can be knocked out with proline mutations 2. sufficiency experiment by adding that segment into a totally unrelated human IDR

Unfortunately, however, they later realize that they can 'distribute' the residues from the SLiM, so that in fact it is not clear if they should be considered SLiM at all, but rather 'context' properties or 'chemical specificity'.

We apologize to the reviewer for our lack of clarity. The essential motif (EM, identified in **Fig. 3**) is indeed a real motif. It is sensitive to point mutations (**Fig. 3G**), cannot be distributed (**Fig. 4C**) or shuffled (**Fig. 4C**), illustrating how in the *wildtype* sequence that motif is necessary for function. HOWEVER, the *orthogonality* here is that in other rationally designed sequences, we do have (many, many) examples in which that EM motif is not necessary, and in fact, *no motif* is necessary.

These observations highlight that what they considered very convincing experiments to demonstrate SLiM-ness are not enough. To really define a new motif, it is usually expected that researchers also identify the (folded?) binding partner that actually recognizes it. I think the authors should refer back to the classical literature on defining motifs where this was appreciated. For example, in the 1980s and 1990s it was not considered sufficient to identify even an example of a known motif without demonstrating that it specifically interacted with the expected partner. A new motif could only be defined once the binding partner was known. Hence, the authors have in some ways rediscovered rigorous molecular biology.

We appreciate that such a model may have been previously used; however, given we now appreciate that many sequences can interact in the absence of defined motifs, such that defining a specific partner through (say) pull-down experiments or solving structures is not possible, we are left with the need to expand how we think about protein-protein interactions beyond a model in which all interactions can be characterized as binary molecular recognition events that depend on a fixed bound state.

In the current paper, the authors present this as a long narrative of discovery about SLiMs, but in the end, I don't think that we have evidence for a bona fide motif in this paper. Hence, I think most of this story is unnecessary and is likely to confuse readers who will think there is a bona fide motif in Abf1. (Just because the authors may have been confused and thought they had discovered a motif, is not a reason to confuse the reader about this.)

We again apologize for our lack of clarity. In our revised version, we have attempted to more clearly address this: e.g.:

We reiterate that the EM is a *bona fide* motif: it cannot tolerate point mutations (**Fig. 3G**), be shuffled (**Fig. 3C**), or be distributed (**Fig. 4C**). However, our results here suggest that viability can be achieved either with or without a *bona fide* motif.

...

Is the EM the only motif in IDR2? We noted that during our sequential shuffling analysis, we shuffled Abf1^{G4} in the context of IDR2 (**Fig. 3C**, LS-13). While this construct was viable, there was a notable growth defect upon shuffling this 10 amino acid region (+4 to +2), hinting that Abf1^{G4} may be a non-essential motif in the context of IDR2 (where EM is present). To test if Abf1^{G4} is a motif, we tested a construct in which Abf1^{G4} was distributed across FUS¹⁻¹⁶³12E (**Fig. 4H**). While the FUS¹⁻¹⁶³12E + Abf1^{G4} was viable, the variant in which the Abf1^{G4} was distributed was not, suggesting this region may be a *bona fide* motif.

...

With Pho4¹⁻²⁴⁹, we generated two types of shuffle: (i) a global shuffle, where the entire sequence was shuffled, and (ii) a segmental shuffle, in which subregions of ~40 amino acids were locally shuffled, preserving the overall chemical distribution (**Fig. 4K**). The segmental shuffle was designed explicitly to preserve local chemical

composition and sequence patterning (fig. S15). Unlike in Gal4⁷⁶⁸⁻⁸⁸¹, both of these variants were inviable, consistent with a result in which Pho4¹⁻²⁴⁹ possesses at least one *bona fide* motif. While further investigation to isolate the specific subregion is needed (e.g., as shown in Fig. 3C), this result suggests that additional motifs beyond the EM can function in the context of Abf1.

For example, the authors write:

"Indeed, even though this design was wholly artificial and even though wild-type IDR2 requires a *bona fide* motif, this design was viable with near wildtype-like growth (Fig. 4F). This result illustrates how an IDR with an essential SLiM can be rescued using a rationally-designed IDR that is lacking a SLiM if instead compensatory chemical groups are presented."

"Here, we find that even for an IDR that depends on a *bona fide* motif, alternative sequences with rationally interpretable chemical features can uphold function without any motif. "

However, the authors have not identified the binding partner for what they claim is a "bona fide motif". Hence they have not really reached the standard in the field. To demonstrate a new SLiM, studies need to show evidence of motif-dependent direct, physical interaction with a specific binding partner. Figures 4A,B and D shows the authors conceptual model of binding of a SLiM in an IDR to a folded partner. But as far as I can tell, there is no evidence for this mechanism presented in the paper, and the evidence presented is more simply explained without the SLiM. The most parsimonious explanation for their results seems to be that there is no SLiM at all, but rather distributed properties in their IDRs. As they finally report:

"In sum, we found we can score each of our variants based on a hydrophobicity score (enhanced by aliphatic and aromatic residues, reduced by proline) and a charge score (reflecting the net charge of the sequence) (Fig. 4J). In this empirical chemical space, viable and inviable constructs predominantly fall on either side of a dividing line, with Pho41-249 a notable exception (fig. S21). "

The authors need to greatly reduce or remove the discussion of 'bona fide motifs' in their paper, and make the paper about what they actually found: chemical specificity.

We appreciate that our original version lacked clarity for this reviewer. Rather than reiterating the same points again, we hope our responses to early comments go some way to address the reviewer's concern.

At some level, however, we recognize our revisions here may not fully address the reviewer's concerns. If they believe that the only way to identify a motif is to also find its binding partner, the reviewer is right that we do not know what this binding partner is. We emphasize that a big reason for this is that we do not necessarily believe there is a single binding partner, but instead that IDR2 may interact with multiple different partners. As mentioned in our response to reviewer 2, recent large-scale work investigating the interactomes for all yeast proteins by immunoprecipitation coupled

to mass spectrometry from the Mann group failed to identify ANY meaningful interactors for Abf1. We probably shouldn't take this to mean that Abf1 has no interactors, but instead that those interactions are not well-described as fixed, structured bound-states. This is of course consistent with our work here.

As a final note, we fully appreciate that this manuscript is overly complex. This is not ideal. It is a consequence of two related things. First, four years of revisions across seven reviewers, each of whom had distinct concerns and questions. Second, and perhaps more concretely, it reflects the fact that biology IS very complex, and we (in this manuscript) absolutely do not have all the answers. We have tried to be as open and explicit about that as possible in our revised version, and we hope this reviewer can appreciate that while many questions remain, the work here does open the door to new ideas and new ways of thinking about a range of topics, a claim we support by the fact the preprinted version of this manuscript has been cited almost 50 times. We greatly appreciate the reviewer's time and perspective on this, and hope they can appreciate we have done our best to address their concerns while retaining the spirit of what we believe to be an important contribution to the field.

References

1. Das, R. K. & Pappu, R. V. Conformations of intrinsically disordered proteins are influenced by linear sequence distributions of oppositely charged residues. *Proc. Natl. Acad. Sci. U. S. A.* **110**, 13392–13397 (2013).
2. Holehouse, A. S., Das, R. K., Ahad, J. N., Richardson, M. O. G. & Pappu, R. V. CIDER: Resources to analyze sequence-ensemble relationships of intrinsically disordered proteins. *Biophys. J.* **112**, 16–21 (2017).
3. Cohan, M. C., Shinn, M. K., Lalmansingh, J. M. & Pappu, R. V. Uncovering non-random binary patterns within sequences of intrinsically disordered proteins. *J. Mol. Biol.* **434**, 167373 (2022).
4. Martin, E. W., Holehouse, A. S., Grace, C. R., Hughes, A., Pappu, R. V. & Mittag, T. Sequence determinants of the conformational properties of an intrinsically disordered protein prior to and upon multisite phosphorylation. *J. Am. Chem. Soc.* **138**, 15323–15335 (2016).
5. Martin, E. W., Holehouse, A. S., Peran, I., Farag, M., Incicco, J. J., Bremer, A.,

- Grace, C. R., Soranno, A., Pappu, R. V. & Mittag, T. Valence and patterning of aromatic residues determine the phase behavior of prion-like domains. *Science* **367**, 694–699 (2020).
6. Zarin, T., Tsai, C. N., Nguyen Ba, A. N. & Moses, A. M. Selection maintains signaling function of a highly diverged intrinsically disordered region. *Proc. Natl. Acad. Sci. U.S.A.* **114**, E1450–E1459 (2017).
 7. Zarin, T., Strome, B., Peng, G., Pritišanac, I., Forman-Kay, J. D. & Moses, A. M. Identifying molecular features that are associated with biological function of intrinsically disordered protein regions. *Elife* **10**, (2021).
 8. Pritišanac, I., Alderson, T. R., Kolarić, Đ., Zarin, T., Xie, S., Lu, A., Alam, A., Maqsood, A., Youn, J.-Y., Forman-Kay, J. D. & Moses, A. M. A functional map of the human intrinsically disordered proteome. *bioRxiv* 2024.03.15.585291 (2024). doi:10.1101/2024.03.15.585291
 9. Boeger, H., Griesenbeck, J., Strattan, J. S. & Kornberg, R. D. Nucleosomes unfold completely at a transcriptionally active promoter. *Mol. Cell* **11**, 1587–1598 (2003).
 10. Kubik, S., O’Duibhir, E., de Jonge, W. J., Mattarocci, S., Albert, B., Falcone, J.-L., Bruzzone, M. J., Holstege, F. C. P. & Shore, D. Sequence-directed action of RSC remodeler and general regulatory factors modulates +1 nucleosome position to facilitate transcription. *Mol. Cell* **71**, 89–102.e5 (2018).
 11. Challal, D., Barucco, M., Kubik, S., Feuerbach, F., Candelli, T., Geoffroy, H., Benaksas, C., Shore, D. & Libri, D. General Regulatory Factors Control the Fidelity of Transcription by Restricting Non-coding and Ectopic Initiation. *Mol. Cell* **72**, 955–969.e7 (2018).
 12. Shivaswamy, S. & Iyer, V. R. Stress-dependent dynamics of global chromatin remodeling in yeast: dual role for SWI/SNF in the heat shock stress response.

Mol. Cell. Biol. **28**, 2221–2234 (2008).

13. Eustermann, S., Patel, A. B., Hopfner, K.-P., He, Y. & Korber, P. Energy-driven genome regulation by ATP-dependent chromatin remodellers. *Nat. Rev. Mol. Cell Biol.* **25**, 309–332 (2024).
14. Mahendrawada, L., Warfield, L., Donczew, R. & Hahn, S. Low overlap of transcription factor DNA binding and regulatory targets. *Nature* **642**, 796–804 (2025).
15. Rossi, M. J., Kuntala, P. K., Lai, W. K. M., Yamada, N., Badjatia, N., Mittal, C., Kuzu, G., Bocklund, K., Farrell, N. P., Blanda, T. R., Mairose, J. D., Basting, A. V., Mistretta, K. S., Rocco, D. J., Perkinson, E. S., Kellogg, G. D., Mahony, S. & Pugh, B. F. A high-resolution protein architecture of the budding yeast genome. *Nature* **592**, 309–314 (2021).
16. Albert, I., Mavrich, T. N., Tomsho, L. P., Qi, J., Zanton, S. J., Schuster, S. C. & Pugh, B. F. Translational and rotational settings of H2A.Z nucleosomes across the *Saccharomyces cerevisiae* genome. *Nature* **446**, 572–576 (2007).
17. Krietenstein, N., Wal, M., Watanabe, S., Park, B., Peterson, C. L., Pugh, B. F. & Korber, P. Genomic Nucleosome Organization Reconstituted with Pure Proteins. *Cell* **167**, 709–721.e12 (2016).
18. Oberbeckmann, E., Niebauer, V., Watanabe, S., Farnung, L., Moldt, M., Schmid, A., Cramer, P., Peterson, C., Eustermann, S., Hopfner, K. & Korber, P. Ruler elements in chromatin remodelers set nucleosome array spacing and phasing. *Nat. Commun.* **12**, 1–17 (2020).
19. Kubik, S., Bruzzone, M. J., Challal, D., Dreos, R., Mattarocci, S., Bucher, P., Libri, D. & Shore, D. Opposing chromatin remodelers control transcription initiation frequency and start site selection. *Nat. Struct. Mol. Biol.* **26**, 744–754 (2019).

20. Hall, P. M., Mayse, L. A., Bai, L., Smolka, M. B., Pugh, B. F. & Wang, M. D. High-resolution genome-wide mapping of chromatin accessibility and torsional stress. *bioRxiv* 2024.10.11.617876 (2024). doi:10.1101/2024.10.11.617876
21. Oberbeckmann, E., Quililan, K., Cramer, P. & Oudelaar, A. M. In vitro reconstitution of chromatin domains shows a role for nucleosome positioning in 3D genome organization. *Nat. Genet.* **56**, 483–492 (2024).
22. Michaelis, A. C., Brunner, A.-D., Zwiebel, M., Meier, F., Strauss, M. T., Bludau, I. & Mann, M. The social and structural architecture of the yeast protein interactome. *Nature* **624**, 192–200 (2023).
23. Jonas, F., Navon, Y. & Barkai, N. Intrinsically disordered regions as facilitators of the transcription factor target search. *Nat. Rev. Genet.* (2025). doi:10.1038/s41576-025-00816-3
24. Zentner, G. E., Kasinathan, S., Xin, B., Rohs, R. & Henikoff, S. ChEC-seq kinetics discriminates transcription factor binding sites by DNA sequence and shape in vivo. *Nat. Commun.* **6**, 8733 (2015).
25. Gutin, J., Sadeh, R., Bodenheimer, N., Joseph-Strauss, D., Klein-Brill, A., Alajem, A., Ram, O. & Friedman, N. Fine-resolution mapping of TF binding and chromatin interactions. *Cell Rep.* **22**, 2797–2807 (2018).
26. Maclsaac, K. D., Wang, T., Gordon, D. B., Gifford, D. K., Stormo, G. D. & Fraenkel, E. An improved map of conserved regulatory sites for *Saccharomyces cerevisiae*. *BMC Bioinformatics* **7**, 113 (2006).
27. Pachkov, M., Balwiercz, P. J., Arnold, P., Ozonov, E. & van Nimwegen, E. SwissRegulon, a database of genome-wide annotations of regulatory sites: recent updates. *Nucleic Acids Res.* **41**, D214–20 (2013).
28. Badis, G., Chan, E. T., van Bakel, H., Pena-Castillo, L., Tillo, D., Tsui, K., Carlson, C. D., Gossett, A. J., Hasinoff, M. J., Warren, C. L., Gebbia, M.,

- Talukder, S., Yang, A., Mnaimneh, S., Terterov, D., Coburn, D., Li Yeo, A., Yeo, Z. X., Clarke, N. D., Lieb, J. D., Ansari, A. Z., Nislow, C. & Hughes, T. R. A library of yeast transcription factor motifs reveals a widespread function for Rsc3 in targeting nucleosome exclusion at promoters. *Mol. Cell* **32**, 878–887 (2008).
29. Rauluseviciute, I., Riudavets-Puig, R., Blanc-Mathieu, R., Castro-Mondragon, J. A., Ferenc, K., Kumar, V., Lemma, R. B., Lucas, J., Chèneby, J., Baranasic, D., Khan, A., Fornes, O., Gundersen, S., Johansen, M., Hovig, E., Lenhard, B., Sandelin, A., Wasserman, W. W., Parcy, F. & Mathelier, A. JASPAR 2024: 20th anniversary of the open-access database of transcription factor binding profiles. *Nucleic Acids Res.* **52**, D174–D182 (2024).
30. Lyons, H., Pradhan, P., Prakasam, G., Vashishtha, S., Li, X., Eppert, M., Fornero, C., Tcheuyap, V. T., McGlynn, K., Yu, Z., Raju, D. R., Koduru, P. R., Xing, C., Kapur, P., Brugarolas, J. & Sabari, B. R. RNA polymerase II partitioning is a shared feature of diverse oncofusion condensates. *Cell* **0**, (2025).
31. Lyons, H., Veettil, R. T., Pradhan, P., Fornero, C., De La Cruz, N., Ito, K., Eppert, M., Roeder, R. G. & Sabari, B. R. Functional partitioning of transcriptional regulators by patterned charge blocks. *Cell* **186**, 327–345.e28 (2023).
32. Ginell, G. M., Emenecker, R. J., Lotthammer, J. M., Keeley, A. T., Plassmeyer, S. P., Razo, N., Usher, E. T., Pelham, J. F. & Holehouse, A. S. Sequence-based prediction of intermolecular interactions driven by disordered regions. *Science* **388**, eadq8381 (2025).
33. Wang, J., Choi, J.-M., Holehouse, A. S., Lee, H. O., Zhang, X., Jahnelt, M., Maharana, S., Lemaitre, R., Pozniakovsky, A., Drechsel, D., Poser, I., Pappu, R. V., Alberti, S. & Hyman, A. A. A molecular grammar governing the driving forces for phase separation of prion-like RNA binding proteins. *Cell* **174**, 688–699.e16 (2018).

34. Schmidt, H. B. & Görlich, D. Nup98 FG domains from diverse species spontaneously phase-separate into particles with nuclear pore-like permselectivity. *Elife* **4**, (2015).
35. Erijman, A., Kozlowski, L., Sohrabi-Jahromi, S., Fishburn, J., Warfield, L., Schreiber, J., Noble, W. S., Söding, J. & Hahn, S. A high-throughput screen for transcription activation domains reveals their sequence features and permits prediction by deep learning. *Mol. Cell* **78**, 890–902.e6 (2020).
36. Brodsky, S., Jana, T., Mittelman, K., Chapal, M., Kumar, D. K., Carmi, M. & Barkai, N. Intrinsically disordered regions direct transcription factor in vivo binding specificity. *Mol. Cell* **79**, 459–471.e4 (2020).
37. Chong, S. & Mir, M. Towards decoding the sequence-based grammar governing the functions of intrinsically disordered protein regions. *J. Mol. Biol.* **433**, 166724 (2021).

Reviewer #1:

Remarks to the Author:

The authors have addressed the reviewer comments by extensive new data and textual revisions. The body of work is extremely valuable for understanding the relationship between IDR sequence and function, as well as for clarifying commonly use terms such as "conservation" and "motif". The authors have provided a context for the development of the ideas presented, giving appropriate references, and have summarized the limitations of the work in a very reasonable manner. The paper will be of significant interest to a broad readership and should be highly impactful.

Remarks on code availability:

The GitHub is still not updated, but the authors provide a good rationale.

We thank the reviewer for their positive assessment and helpful comments in round 1. We are pleased to say that the GitHub repository is now fully updated. We thank the reviewer for their support!

Reviewer #2:

Remarks to the Author:

The authors provide new data showing that inviable IDR variants correlate with effects on +1 nucleosome repositioning at essential genes, thereby linking Abf1's molecular function on chromatin to viability. They appropriately acknowledge the study's limitations and discuss key open questions regarding the role of IDRs in chromatin regulation for future investigation. Overall, I find the study improved and support its publication.

We thank the reviewer for their positive assessment, helpful comments in round 1, and support of publication!

Reviewer #3:

Remarks to the Author:

Overall, the authors have greatly improved their manuscript. The authors have not really addressed my comments. In the introduction they have clarified the claim that chemical specificity and motifs are orthogonal, but only in the discussion do they mention that there is previous work that has explored the connection. For example, in 2019 (PMID: 31134302) it was reported that charge contributes to the binding of PIP-boxes. Thanks to the revisions, I now appreciate that the authors are trying to say something more general here, i.e., a motif can be completely replaced by multivalent properties, but I still believe they should introduce their work in the scholarly context.

We thank the reviewer for this comment, and have updated the introduction to make this point clearer.

Similarly, the authors still make statements like this:

"Conventionally speaking, the ability to insert a short (<20 residue) sequence into a non-functional IDR context and confer function is interpreted as a simple and unambiguous demonstration of a bona fide modular motif (e.g., SLiM). Given that sequence-specific motifs are often defined by three characteristics – an inability to tolerate shuffling (Fig. 3C), sensitivity to point mutations (Fig. 3G), and autonomous modular activity (Fig. 3I), our results confirm that the EM is a bona fide motif."

Can the authors at least provide some references for their understanding of the 'conventional' evidence for bona fide motifs? These are not the unambiguous demonstrations of motifs that I am familiar with in the literature. In my experience is that what is considered unambiguous demonstration of motifs are structural studies of the binding partner and the linear motif (as shown as a cartoon by the authors in figure 4A), along with experiments in cells and in vitro where the effects of mutations on measured interactions are predicted by the structures.

We thank the reviewer for this question; we have rewritten this section to remove the term "unambiguous" (which we feel is somewhat subjective and possibly not useful) include references to literature from the SLiM field and then a specific primary study in which deep biochemical and biophysical characterization of a motif was performed (including NMR analysis) yet it was not possible to obtain sufficient structural restraints to solve a 3D structure.

We also note that this reviewer's line of inquiry has been extremely well appreciated, while beyond the scope of this work, we are now actively working on asking if – formally – the degree of sequence-specificity encoded by a SLiM directly informs the degree of structure in the bound state. If this were the case, while "structure" here would of course be on a continuum, perhaps the reviewer is correct that we *should* be referring to "sequence-specific motifs" as subregions for which a defined bound-state structure can be resolved, versus referring to binding driven by chemical specificity where a structure (i.e. a representative 3D snapshot of a bound-state complex) cannot be obtained. In any case, we feel extending our already too-long manuscript to explore this idea would not be possible, but we wanted to clarify that far from dismissing the reviewer's concerns here, they have forced us to think hard and critically about the next steps on the biophysical side of this project.

We include our revised text for completeness:

Conventionally speaking, the ability of a short (<20 residue) sequence to confer function when inserted into a non-functional context is taken to demonstrate a *bona fide* motif (e.g., a SLiM)^{1,2}. Given SLiMs can interact without acquiring a defined 3D structure, structural characterization is sufficient but not necessary to nominate a region as a SLiM³.

References

1. Davey, N. E., Van Roey, K., Weatheritt, R. J., Toedt, G., Uyar, B., Altenberg, B., Budd, A., Diella, F., Dinkel, H. & Gibson, T. J. Attributes of short linear motifs. *Mol. Biosyst.* **8**, 268–281 (2012).
2. Kumar, M., Michael, S., Alvarado-Valverde, J., Zeke, A., Lazar, T., Glavina, J., Nagy-Kanta, E., Donagh, J. M., Kalman, Z. E., Pascarelli, S., Palopoli, N., Dobson, L., Suarez, C. F., Van Roey, K., Krystkowiak, I., Griffin, J. E., Nagpal, A., Bhardwaj, R., Diella, F., Mészáros, B., Dean, K., Davey, N. E., Pancsa, R., Chemes, L. B. & Gibson, T. J. ELM-the Eukaryotic Linear Motif resource-2024 update. *Nucleic Acids Res.* **52**, D442–D455 (2024).
3. O’Shea, C., Staby, L., Bendtsen, S. K., Tidemand, F. G., Redsted, A., Willemoës, M., Kragelund, B. B. & Skriver, K. Structures and short linear motif of disordered transcription factor regions provide clues to the interactome of the cellular hub protein radical-induced cell Death1. *J. Biol. Chem.* **292**, 512–527 (2017).